# Protein buffering of aneuploidy is driven by coordinated factors identified through machine learning

Erik Marcel Heller [ID]1, Karen Barthel[1], Markus Räschle [ID]1, Klaske M Schukken[2,3], Jason M Sheltzer [ID]2,3✉ & Zuzana Storchová [ID]1✉

## Abstract

**Aneuploidy, a hallmark of cancer, alters chromosome copy numbers and with that the abundance of hundreds of proteins. Evidence suggests that levels of proteins encoded on affected chromosomes are often buffered toward their abundances observed in diploids. Despite its prevalence, the molecular mechanisms driving this protein dosage compensation remain largely unknown. It is unclear whether all proteins are buffered similarly, what factors determine buffering, and whether dosage compensation varies across different cell lines or tumor types. Moreover, its potential adaptive advantage and therapeutic relevance remain unexplored. We established a novel approach to quantify protein dosage buffering in a gene copy number-dependent manner, showing that dosage compensation is widespread but variable in cancer samples. By developing multifactorial machine learning models, we identify gene dependency, protein complex participation, haploinsufficiency, and mRNA decay as key predictors of buffering. We show that dosage compensation affects oncogenic potential and that higher buffering correlates with reduced proteotoxic stress and increased drug resistance. These findings highlight protein dosage compensation as a crucial regulatory mechanism with therapeutic potential in aneuploid cancers.**

**Keywords** Aneuploidy; Dosage Compensation; Buffering; Machine Learning; Cancer
**Subject Categories** Cancer; Chromatin, Transcription & Genomics

## Introduction

Chromosome and chromosome arm copy number changes, commonly referred to as chromosomal aneuploidy, are a widespread hallmark of cancer (Santaguida and Amon, 2015; Sdeor et al, 2024; Taylor et al, 2018). Aneuploidy alters the expression of many genes located on the affected chromosomes, including tumor suppressors and oncogenes. The altered gene expression disrupts cellular regulation, driving tumorigenesis and promoting cancer progression. This likely explains why certain chromosomes or chromosome arms are recurrently gained or lost across various cancers and why specific aneuploidies are frequently associated with distinct cancer types (Davoli et al, 2013; Shih et al, 2023). For example, trisomy 12 is common in chronic lymphocytic leukemia, while loss of chromosome 13q is prevalent in retinoblastoma. Aneuploidy is often associated with chromosomal instability, and together they facilitate tumor evolution by providing a selective advantage to cancer cells, increasing drug resistance, and metastatic dissemination (Andrade et al, 2023; Lukow and Sheltzer, 2022; Replogle et al, 2020; Shukla et al, 2020).

However, aneuploidy affects the copy number of hundreds of genes, and this deregulated gene expression may have negative effects on cellular proliferation. Evidence from in vitro models of aneuploidy in budding yeast, murine, and human cells revealed that aneuploidy often impairs proliferation, induces cellular senescence, increases genomic instability, and disrupts metabolic homeostasis. Remarkably, aneuploid cells exhibit increased sensitivity to compounds that interfere with protein folding and turnover (Donnelly and Storchová, 2015; Donnelly et al, 2014; Tang et al, 2011; Torres et al, 2010). This suggests that the aberrant gene expression in aneuploid cells overwhelms cellular quality control systems involved in the maintenance of protein homeostasis. Indeed, aneuploid cells often experience an imbalance in the synthesis and degradation of proteins, leading to the accumulation of misfolded proteins, which must be resolved by the proteasome and other protein degradation pathways (Donnelly and Storchová, 2015; Donnelly et al, 2014; Oromendia and Amon, 2014; Oromendia et al, 2012). Moreover, secondary global (i.e., in trans) gene expression changes are often observed in aneuploid cells, as gene expression of multiple genes across the entire genome is altered in response to the stresses induced by aneuploidy (Sheltzer et al, 2012; Stingele et al, 2012).

Transcriptome and proteome analyses of engineered aneuploid cells have provided useful insights into the gene expression changes triggered by aneuploidy. For instance, in budding yeast strains with

[1]Molecular Genetics, Department of Biology, RPTU University Kaiserslautern-Landau, Kaiserslautern, Germany. [2]Department of Surgery, Yale School of Medicine, Yale University, New Haven, CT, USA. [3]Present address: Stanford Cancer Institute, Stanford University School of Medicine, Palo Alto, CA 94304, USA. ✉E-mail: sheltzer@stanford.edu; zuzana.storchova@rptu.de

an extra chromosome, most genes on the aneuploid chromosome show an increased abundance of both mRNA and proteins, reflecting the proportional increase in gene dosage (Dephoure et al, 2014; Torres et al, 2007), but the abundance of ~20% of proteins encoded on extra chromosomes shows dosage compensation, meaning the protein levels are adjusted to levels similar to euploid levels (Fig. 1A; Dephoure et al, 2014). This dosage compensation, also referred to as protein abundance attenuation or buffering, is particularly apparent for subunits of multi-protein complexes and largely depends on proteasomal and lysosomal degradation of the excess proteins, and their sequestration into aggregates (Brennan et al, 2019; Dephoure et al, 2014; Senger and Schaefer, 2021; Taggart et al, 2020). In naturally occurring aneuploid yeast strains, the dosage compensation affects an even larger fraction of proteins and is accompanied by enhanced protein turnover via proteasomal degradation (Muenzner et al, 2024). Analysis of engineered cancerous and non-cancerous human cell lines harboring chromosome gains demonstrated that most genes on gained chromosomes are transcribed and translated proportionally to the chromosome copy number gain. However, protein complex subunits, protein kinases, and several other proteins showed an apparent dosage compensation at the protein level (Stingele et al, 2012; Viganó et al, 2018). Similarly, dosage compensation modulates the abundance of proteins in cells from individuals with trisomy syndromes (Hwang et al, 2021; Liu et al, 2017). Loss of a single chromosome copy is also compensated in engineered monosomic human cells (Chunduri et al, 2021). How dosage changes of chromosomes and single genes shape the proteome in human cancers is poorly understood. Gene amplification often results in increased protein abundance, but in up to ~30% of cases, the abundance changes do not scale linearly (Gonçalves et al, 2017; Mertins et al, 2016; Sousa et al, 2019; Zhang et al, 2014; Zhang et al, 2016). Recently, the Cancer Cell Line Encyclopedia (DepMap CCLE) provided data for hundreds of cell lines with matched DNA copy number, RNA expression, and protein expression measurement levels (DepMap, Broad Institute, 2023; Ghandi et al, 2019; Nusinow et al, 2020). Analysis of this dataset revealed that the mRNA abundances of the majority of genes scale with gene dosage changes, while protein abundance frequently changes less than expected based on the chromosome copy number (Schukken and Sheltzer, 2022). Thus, aneuploidy is subject to dosage compensation at the protein level even in cancer cell lines. The existence of dosage compensation has also been demonstrated in aneuploid ovarian tumors (Schukken and Sheltzer, 2022). However, a comprehensive overview of dosage compensation across multiple in vivo tumor types remains to be established. Previous studies by Schukken and Sheltzer aimed to identify the mechanisms involved in dosage compensation, but the predictive quality of individual factors analyzed in the study was rather low (Schukken and Sheltzer, 2022).

Here, we present a comprehensive analysis of protein dosage compensation in human samples by developing a novel approach that quantifies compensation at single-gene resolution across copy number changes of somatic chromosomes. Exploiting pan-cancer cell line and tumor sample datasets from DepMap CCLE, ProCan, CPTAC, and lab-engineered aneuploid cell lines, our method quantifies dosage compensation for each gene within individual samples using focal gene and chromosome-wide copy number alterations, providing increased granularity. By integrating diverse biochemical and genetic variables from multiple resources into explainable machine learning models, we achieved high predictive accuracy, uncovering critical mRNA- and protein-level features that drive compensation of aneuploid somatic chromosomes. Notably, our findings reveal that cancers with low dosage compensation exhibit more vulnerabilities than those with high compensation—a disparity that can be exploited for targeted therapies. Overall, our results position dosage compensation as a factor contributing to tumorigenesis and cancer resilience.

# Results

## Calculating protein-level attenuation using absolute gene copy numbers in human cell lines and tumor samples

Precise quantification of dosage compensation relies on the analysis of copy number changes across the entire genome, matched to the changes observed on the proteome level. In previous studies, copy number changes were commonly evaluated as gains or losses of entire chromosomes or chromosome arms. However, we frequently observed sub-arm copy number changes in tumor tissues and cancer cells (Fig. EV1A). To increase the granularity of such analyses, we derived a novel indicator, which we term Buffering Ratio (BR), which compares both focal gene and chromosome-wide copy number changes against changes in protein abundance. The BR is defined as the difference between protein abundance log2 fold change (Log2FC) and (gene or chromosome) copy number Log2FC (see "Methods"). This definition allows the use of numeric gene and chromosome arm copy number differences for buffering quantification, thus increasing the precision of the buffering estimate, compared to a priori categorization of chromosome arm copy number alterations (CNAs) into gain, neutral, and loss as previously applied (Schukken and Sheltzer, 2022). The BR allows protein buffering quantification for each gene within each sample (cell line or tumor sample), thus improving analysis granularity compared to methods that average protein buffering per gene across a set of samples. Since the BR is sensitive to noise from both proteomics and copy number data, we calculated confidence scores for the BRs and removed BR measurements with low confidence from subsequent analyses.

Based on the BR and the co-directionality of the protein abundance changes with the gene copy number, we categorized genes into the buffering classes *Scaling*, if the change in protein abundance was proportional to the change in gene copy number; *Buffered,* if the protein abundance change was less than proportional; and *Anti-Scaling*, if these values were anti-proportional (Fig. 1B). *Scaling* and *Buffered* proteins were distinguished using a threshold on the copy number-dependent BR, whereas *Buffered* and *Anti-Scaling* proteins were differentiated using a copy number-independent protein abundance Log2FC threshold. We calculated the BR and buffering classes for each gene and sample using chromosome arm CNAs (Cohen-Sharir et al, 2021) and absolute gene copy numbers (Carter et al, 2012; Data ref: DepMap, Broad Institute, 2023) for the pan-cancer cell line datasets DepMap CCLE containing proteomics data for 375 cell lines, Sanger ProCan with 949 cell lines, and the pan-cancer tumor sample dataset CPTAC with 1026 tumor samples and 523 normal tissue samples. We removed tumor samples with a purity below 40% to control for

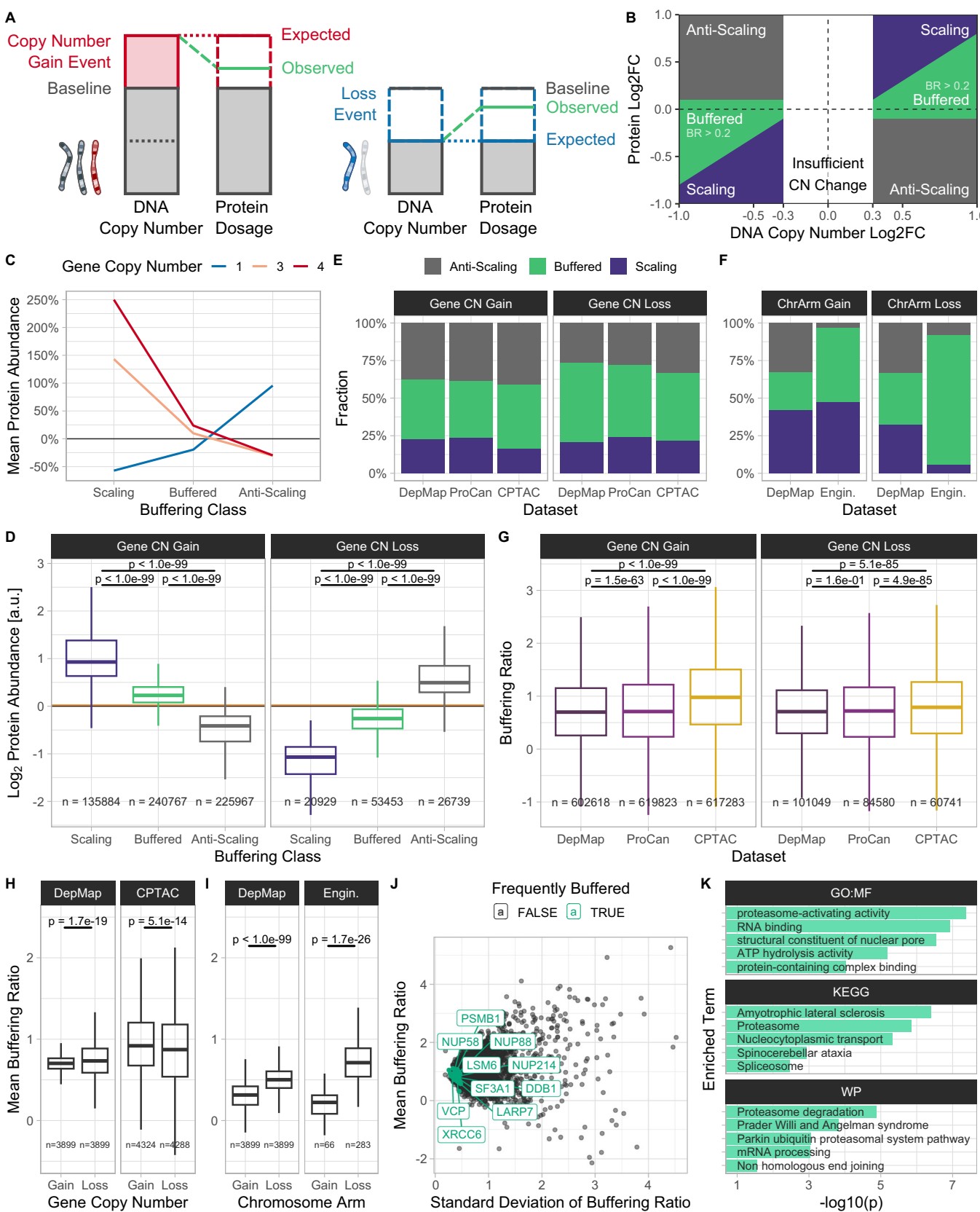

**Figure 1.  Quantification of protein dosage compensation in aneuploid cancer cells.**

(A) Illustration of protein dosage compensation upon gene copy number gain (left) and loss (right). (B) Illustration of the buffering classes *Scaling*, *Buffered*, and *Anti-Scaling*. A threshold on the buffering ratio (BR) was used to distinguish *Buffered* (green) from *Scaling* (indigo) observations, while protein log2 fold-change thresholds were used to distinguish *Buffered* and *Anti-Scaling* (gray) observations. Observations with a DNA copy number log2 fold change between −0.3 and 0.3 were removed (white). (C) Mean change in protein abundance relative to the median disomic protein abundance baseline for proteins in each buffering class in the DepMap CCLE pan-cancer cell line dataset. (D) Difference in protein abundance between the different buffering classes (shown dataset: DepMap). The mean disomic protein abundance baseline is represented as an orange line. (E) Categorical distribution of buffering classes across pan-cancer datasets (DepMap, ProCan, CPTAC) upon gene copy number gain and loss. (F) Categorical distribution of buffering classes across DepMap CCLE and lab-engineered cell lines upon chromosome arm gain and loss. (G) Difference in buffering ratio distribution of all proteins with either gene copy number gain or loss between pan-cancer datasets (DepMap, ProCan, CPTAC). (H) Difference in mean buffering ratio distribution upon gene copy number gain and loss across pan-cancer datasets (DepMap, CPTAC). (I) Difference in mean buffering ratio distribution upon chromosome arm gain and loss across DepMap CCLE and lab-engineered cell lines. (J) Mean and standard deviation of buffering ratio of proteins across samples from the pan-cancer tumor sample dataset CPTAC. Highlighted are 50 genes with the lowest buffering ratio variance among frequently buffered genes across all pan-cancer datasets (DepMap, ProCan, CPTAC). (K) Top 5 enriched terms per selected data source in over-representation analysis (ORA) of highlighted low-variance genes (hypergeometric test using g:Profiler). (D, G–I) Boxes represent the interquartile range (IQR) with the central line indicating the median. The whiskers extend to the data points within 1.5×IQR. *P* values were determined using a two-tailed Wilcoxon rank-sum (Mann–Whitney *U*) test.

tumor purity. Additionally, we analyzed near-diploid p53-deficient human retinal pigment epithelium cell lines (RPE-1) engineered to carry either monosomies or trisomies on single chromosomes (Chunduri et al, 2021; and data from this study). We provide individual buffering ratios and classes upon gene and chromosome copy number gains and losses for all datasets as supplementary data (Dataset EV1).

As expected, proteins classified as Scaling using the BR showed a change in protein abundance proportional to the change in copy number, whereas *Buffered* proteins retained near-disomic levels (Fig. 1C,D). We found that the majority of proteins were either *Buffered* (38–53%) or *Anti-Scaling* (26–41%) upon gene copy number gain and loss across pan-cancer cell line (DepMap, ProCan) and tumor sample datasets (CPTAC; Fig. 1E). CPTAC had the highest fraction of *Buffered* proteins upon gene copy number gain (42%), and DepMap had the highest fraction of *Buffered* proteins upon gene copy number loss (53%). In addition, the fractions of *Buffered* observations were higher upon gene copy number loss than upon gain (gain: 38–42%, loss: 45–53%), whereas *Anti-Scaling* was reduced upon copy number loss (gain: 37–41%, loss: 26–33%). This pattern was preserved in the pan-cancer cell line dataset DepMap CCLE when using chromosome arm CNA data (Fig. 1F), and was also preserved when controlling for whole-genome doubling (WGD; Fig. EV1C). Interestingly, lab-engineered cell lines showed fewer *Anti-Scaling* proteins compared to pan-cancer datasets, and almost all proteins affected by chromosome arm loss were classified as *Buffered* (Fig. 1F; Dataset EV1). Thus, gene dosage buffering is a widespread phenomenon in engineered aneuploid cells.

Comparing BR distributions between datasets showed that tumor samples have significantly increased BR upon gain and loss compared to cell lines, in line with the observed increase in *Buffered* and *Anti-Scaling* proteins (Fig. 1G). We did not observe a significant correlation between tumor purity and BR that could confound these results (Fig. EV1D). Instead, we observed slightly increased immune and stromal cell invasion scores in tumor samples with high average BRs (Fig. EV1D). To understand the changes in buffering in different conditions, we compared the BR distributions between (gene copy number and chromosome arm) gains and losses. We observed that the BR was significantly higher upon loss than upon gain events in lab-engineered cell lines and pan-cancer cell lines, while in tumor samples, the BR was higher upon gene copy number gain (Fig. 1H,I). This might be caused by

the fact that tumor samples harbored increased fractions of genes with strong copy number amplifications (Fig. EV1A). Therefore, tumor samples may also show increased buffering to restore the protein abundance of these genes to near-disomic levels. Interestingly, the BR was lower upon gene copy number loss than upon gain in cell lines that underwent WGD, but was higher upon loss in Non-WGD cell lines (Fig. EV1B). This suggests that buffering might be especially important if only one gene copy is available. We conclude that more protein buffering occurs upon copy number loss than upon gain events in aneuploid cell lines, independent of whether these are chromosomal aberrations or focal gene copy number variations.

Having calculated the buffering ratios per gene and per sample, we asked which genes are frequently and consistently buffered independent of cell states. To do this, we filtered for genes that were classified as *Buffered* in more than 33% of samples in all pan-cancer datasets (DepMap, ProCan, CPTAC) and whose BR had a standard deviation smaller than 2. We then aggregated their BR standard deviation ranks across datasets. In the top 10 frequently buffered genes with low BR variation were NUP88, NUP214, and NUP58, three subunits of the nuclear pore complex; ribonucleoproteins LARP7, SF3A1, and LSM6; DNA repair proteins XRCC6, and DDB1; proteasomal subunit PSMB1; and VCP, a protein that segregates protein molecules from large protein assemblies to facilitate their degradation in the proteasome (Fig. 1J; Dataset EV2). All these proteins are subunits of macromolecular complexes, which were previously suggested to be subject to extensive dosage compensation (Stingele et al, 2012). The only exception is the segregase VCP/p97, which interacts with many proteins but is not a stable member of a macromolecular complex. Importantly, VCP has a key role in protein quality control, which is often challenged in aneuploid cells (Gwon et al, 2021; Kadowaki et al, 2015; Oromendia and Amon, 2014; Oromendia et al, 2012; Turakhiya et al, 2018). Over-representation analysis (ORA) further confirmed that RNA binding and processing, spliceosome, and proteasome genes were significantly enriched among the top 50 frequently buffered genes (Fig. 1K). This confirms the previous notion that subunits of macromolecular complexes need to be maintained at stoichiometric levels.

These results were reproducible, as we observed a 50% mean correlation of gene copy number-derived BRs between the cell line datasets DepMap and ProCan and a 55% mean correlation for the top 50 frequently buffered genes (Spearman; Fig. EV1E,F).

## Average buffering per sample relates to cellular and clinical features of cancer

Next, we asked whether protein dosage compensation differs among different cell lines and tumor types. Using the same datasets as previously, we calculated sample buffering ratios (sample BRs) by calculating the mean BR per sample (cell line or tumor sample). Calculating the z-scores of sample BRs showed significant differences in average buffering between cell lines (Fig. 2A). These results were consistent across cell line datasets, as the calculated sample BRs were highly correlated between DepMap and ProCan. The high correlation of the BRs allowed us to aggregate the sample BR ranks of cell lines across the datasets using the mean normalized rank (MNR, Upadhya and Ryan, 2022) to provide a resource of dataset-independent average buffering estimates for each cell line (Fig. 2B; Dataset EV3). Plotting the mean sample BR per cancer type and dataset using the OncoTree classification (Kundra et al, 2021) showed differences in average buffering between cancer types (Fig. 2C). Among cancer types with low average buffering we observed fibrosarcomas, hepatoblastomas, Erwing sarcomas, neuroblastomas, and myeloid/lymphoid neoplasms, whereas pancreatic neuroendocrine tumors, pleural mesotheliomas, hepatocellular carcinomas, anaplastic thyroid cancers, and invasive breast carcinomas were among the high-buffering cancer types. In particular, pediatric cancers possessed lower sample BRs on average than adult cancers, and high-buffering tumor samples showed slightly lower patient survival probabilities (Fig. EV2C,D). Interestingly, TP53 mutations had a non-uniform effect, but we observed that the sample BR is positively correlated with the aneuploidy score (AS) and average ploidy of cell lines (Figs. 2C,D and EV2A,B,E). Furthermore, separating cell lines into cohorts with or without whole-genome doubling (WGD) showed significant differences in sample BR between these cohorts, suggesting that WGD-positive cells exhibit more protein buffering on average (Wilcoxon rank-sum test, ProCan: $P = 8.0 \times 10^{-44}$). We conclude that WGD-positive cells, highly aneuploid cells, and cells from adult cancers exhibit more protein buffering on average, and that the degree of aneuploidy and the WGD status are potential confounding factors that need to be controlled for when analyzing average cell sample buffering.

We noticed that non-solid cancers like myeloid and lymphoid neoplasms (MNM, LNM) and other cell lines grown in suspension cultures show significantly lower average protein buffering than those with adherent growth patterns (Fig. 2E). Since aneuploidy is a confounding factor of the sample BR, and suspension-grown cell lines frequently show low AS, we controlled for differences in the degree of aneuploidy between these two growth pattern cohorts by removing cell lines from the adherent cohort with a higher aneuploidy score than cell lines grown in suspension. We additionally applied a stratified sampling scheme to the adherent growth cohort to ensure equal distributions of the aneuploidy score between the two growth pattern cohorts. Even in this setting, we observed decreased average buffering in cell lines grown in suspension cultures compared to those with adherent growth patterns (Figs. 2E and EV2F,G). Thus, the average degree of protein buffering differs between cells and cancer types and may be an intrinsic feature affected by cellular physiology and tissue type.

## Unifactorial analyses highlight gene dependency, protein complex participation, and random allelic expression as comparatively strong predictors of protein buffering

To investigate which factors influence protein buffering, we compiled and integrated multiple datasets of 35 factors describing biochemical and genetic properties of each protein. We then assessed the predictive power of each factor individually to determine its effectiveness in predicting whether a protein is buffered or not by calculating the receiver operating characteristic area under the curve (ROC AUC). We calculate the ROC AUCs on BR-based (per-gene, per-sample) buffering classes derived from absolute gene copy numbers and chromosome arm copy numbers (GeneCN, ChrArm) for all pan-cancer datasets (DepMap, ProCan, CPTAC). In addition, we determined buffering classes per gene across all samples within a dataset (ChrArm (avg.)) by a previously described method that applies average Log2FC thresholds for grouping proteins into *Scaling*, *Buffered*, and *Anti-Scaling* using chromosome arm CNAs categorized into gained, neutral, and lost chromosome arms (Schukken and Sheltzer, 2022). This resulted in three separate analysis variants (Gene CN, ChrArm, ChrArm (avg.)), providing multiple perspectives of differing granularity when analyzing factors contributing to protein buffering. We then aggregated the ROC AUC ranks of all analysis variants (Gene CN, ChrArm, ChrArm (avg.)) and all pan-cancer datasets (DepMap, ProCan, CPTAC) using mean normalized ranks (MNR), to summarize which factors are most relevant in predicting protein buffering upon gain and loss events in all samples, additionally separated by their WGD status (Dataset EV4).

The most predictive factors for distinguishing protein buffering from scaling upon gain were the mean dependency score (i.e., essentiality) of a gene across cell lines derived from CRISPR knock-out screens (DepMap, Broad Institute, 2023), the number of protein complexes a protein participates in (Giurgiu et al, 2019), the frequency of random allelic expression (RAE) of a gene compared to bi-allelic expression (Kravitz et al, 2023), the number of ubiquitination and methylation sites on a protein (Hornbeck et al, 2015), the probability of triplosensitivity (i.e., duplication intolerance; Collins et al, 2022) of a gene, and the number of transcription factors (TFs) that target the gene (Badia-i-Mompel et al, 2022; Garcia-Alonso et al, 2019) (Fig. 3A). Upon loss, the most predictive factors were the mean gene dependency score, protein complex participation, the degree of non-exponential decay (NED) of a protein (Δ-score, McShane et al, 2016), RAE, and triplosensitivity (Fig. 3B). This shows that there is only a partial overlap of the predictive power of factors between gains and losses. While the NED Δ-score and the probability of haploinsufficiency (i.e., deletion intolerance, Collins et al, 2022) were more predictive upon loss than upon gain events, the mean mRNA decay rate (Yang et al, 2003) was more important upon gain than loss. We next asked whether we could identify factors whose importance in distinguishing protein buffering from scaling is affected by genome doubling. The number of ubiquitination and methylation sites and the number of protein–protein interactions of a protein (Alanis-Lobato et al, 2017) were more important in Non-WGD samples upon gain, while the mean mRNA decay rate (Yang et al, 2003), haploinsufficiency probability, and number of TFs with activating modes of

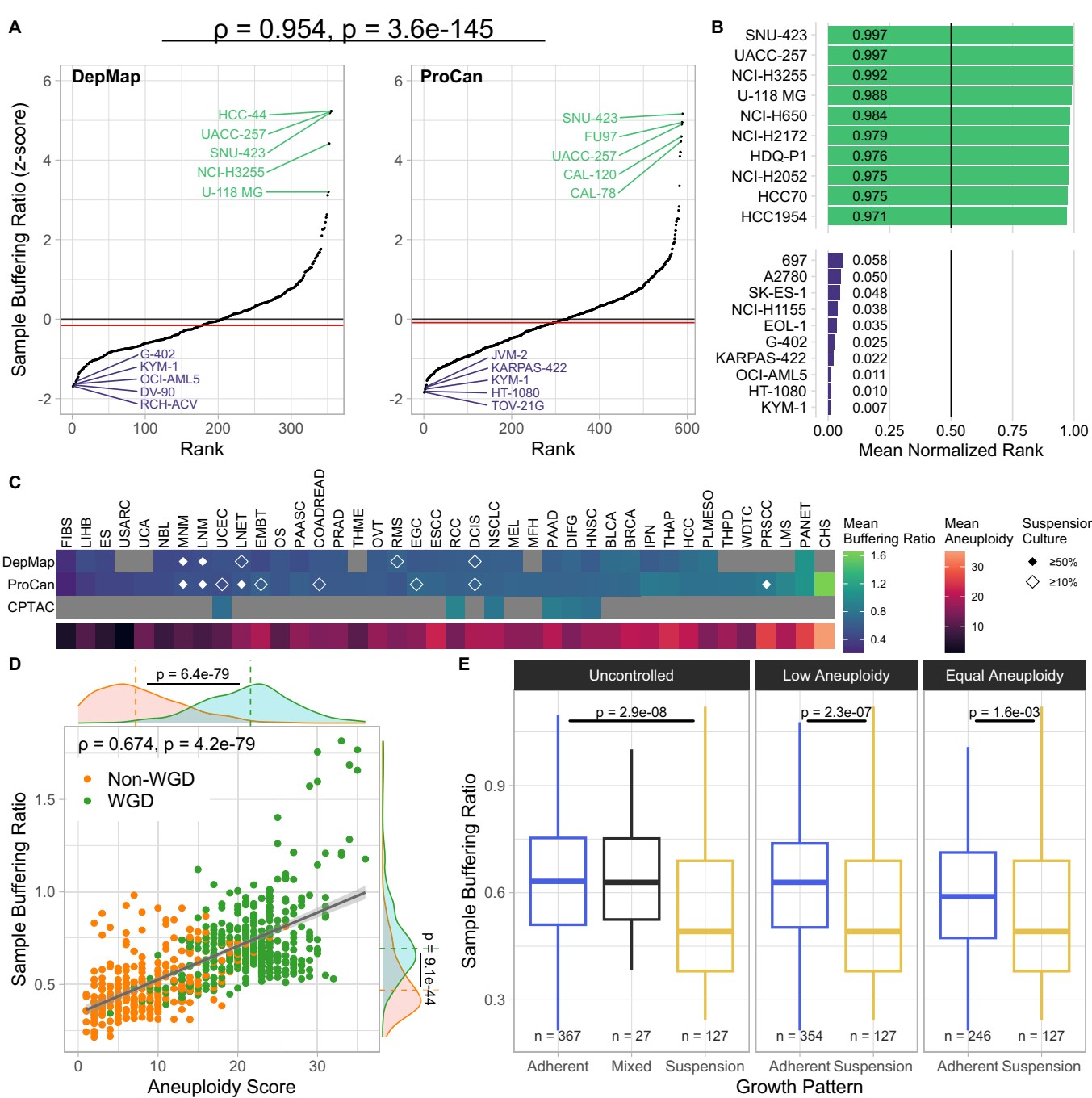

**Figure 2.  Average buffering per sample relates to cellular and clinical features of cancer.**

(A) Z-score of average buffering ratio per sample derived from pan-cancer cell line datasets (DepMap, ProCan). The sample buffering ratio z-scores between samples in DepMap and ProCan are highly correlated (Spearman's $\rho = 0.954$, $P = 3.6 \times 10^{-145}$). (B) Mean normalized ranks (MNR) of sample buffering ratios of cell lines in ProCan and DepMap. (C) Mean sample buffering ratio and aneuploidy score per cancer type (OncoTree code) in pan-cancer datasets. Mean aneuploidy scores of cell lines are added as a reference for the genomic background of each cancer type. Diamond glyphs indicate the fraction of cell lines grown in suspension cultures. (D) Scatter plot showing a positive correlation between sample buffering ratio and aneuploidy score in the ProCan dataset (Spearman's $\rho = 0.674$, $P = 4.2 \times 10^{-79}$). Marginal distributions are separated and compared by whole-genome doubling (WGD) status (Wilcoxon rank-sum test). (E) Difference in sample buffering ratio between cell lines grown in suspension and adherent cell culture types (ProCan). Controlled for aneuploidy by removing cell lines from the adherent cohort with aneuploidy scores above the maximum of the suspension cohort (middle), and by using stratified sampling to ensure an equal distribution of aneuploidy scores (right). Boxes represent the interquartile range (IQR) with the central line indicating the median. The whiskers extend to the data points within 1.5×IQR. P values were determined using a two-tailed Wilcoxon rank-sum (Mann–Whitney U) test.

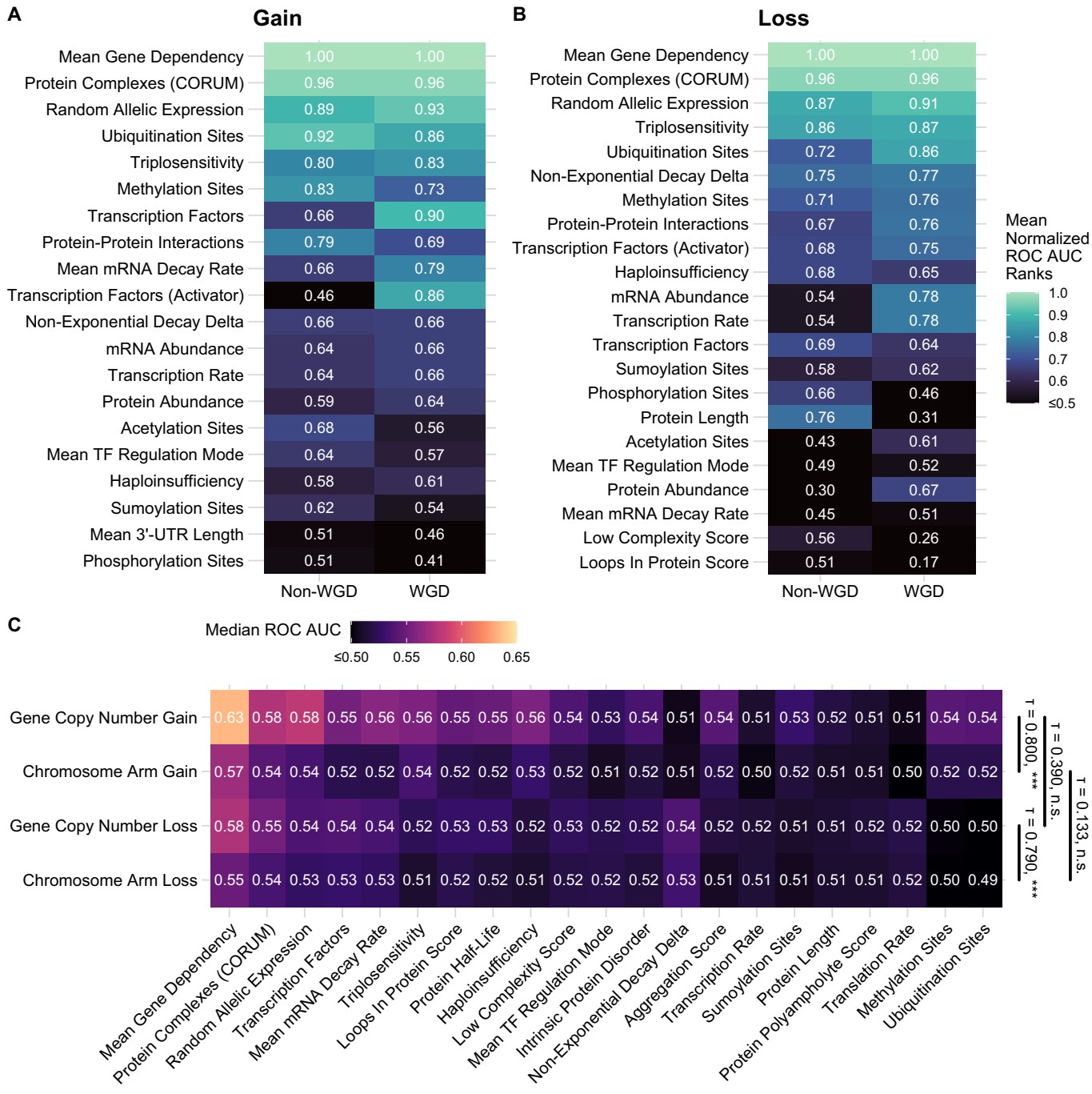

**Figure 3. Unifactorial analyses highlight gene dependency, protein complex participation, and random allelic expression as comparatively strong predictors of protein buffering.**

(A, B) Mean normalized ranks (MNRs) of ROC AUCs of each factor used for classifying whether a gene is *Buffered* or *Scaling* on protein level (*Anti-Scaling* excluded). Heatmaps display selected factors with an MNR of ≥ 0.5 in at least one of the conditions shown. (A) MNRs calculated separately for whole-genome-doubled (WGD) and NonWGD cell lines of the DepMap dataset across ROC AUCs of all analysis variants for gain events. (B) MNRs calculated separately for WGD and NonWGD cell lines of the DepMap dataset across ROC AUCs of all analysis variants for loss events. (C) Median ROC AUCs and rank correlation of factor ROC AUCs (Kendall's τ) between analysis variants by bootstrapping the ProCan dataset ($n = 10,000$). Only factors with a median ROC AUC of 0.51 across analysis variants are shown. Significance levels: *$P < 0.01$, **$P < 0.001$, ***$P < 0.0001$.

action were more important in WGD-positive samples (Fig. 3A). Comparing factors between WGD and Non-WGD samples upon loss events revealed that the number of TFs (activating + repressing), protein length, and number of phosphorylation sites were more important in Non-WGD samples. In contrast, the number of ubiquitination and acetylation sites, mRNA abundance and transcription rate (Hausser et al, 2019), and the number of protein–protein interactions were more important in WGD samples (Fig. 3B).

Next, we determined whether the predictiveness of factors significantly differs between gain and loss events by bootstrapping the buffering classes and calculating ROC AUC rank correlations of factors (see "Methods"; Dataset EV5). To control for differences caused by the choice of the analysis variant, we also evaluated the factor rank differences between buffering classes determined either by using gene copy number data or chromosome arm CNA data. The factor rank correlation was high when comparing gene copy number with chromosome arm gain (or loss) analyses, while there was no significant correlation between gain and loss analyses when using the same copy number source (Fig. 3C). We therefore conclude that protein buffering is at least partially influenced by different factors upon gain than upon loss, independent of whether copy number aberration occurs on gene level or chromosome arm level.

We observed that the maximum achieved ROC AUC using the unifactorial prediction was 0.672, providing a moderate discrimination performance between *Buffered* and *Scaling* proteins (Dataset EV4). Furthermore, the raw values and bootstrapped ROC AUCs of factors such as protein abundance, transcription rate, and mRNA abundance were frequently and strongly pairwise correlated, indicating that these factors carry equivalent information in predicting protein buffering (Fig. EV3B–D). Calculating the correlation between factor value and (gene-level) Buffering Ratio confirmed mean gene dependency score, protein complex participation, triplosensitivity, mRNA decay rate, and ubiquitination sites to be important factors, with higher values being predictive for protein buffering (Fig. EV3E). Higher values of random allelic expression frequency and TF count were more predictive for scaling proteins. However, only a weak correlation was achieved $(\max(|\rho|) = 0.27)$. We conclude that a unifactorial analysis is able to identify factors with higher predictive power relative to each other, but does not enable a strong prediction of protein buffering in total.

## Explainable machine learning uncovers directional relationships between predictive factors and protein buffering

We considered that using all factors simultaneously in a combined model would improve the prediction. To approach this, we trained a series of machine learning models that use a subset of all collected factors simultaneously to predict whether a gene is *Buffered* or *Scaling*, based on the classification scheme established for the unifactorial prediction (Fig. EV4A). Highly correlated factors that contained redundant information were removed to prevent the models from overfitting on an arbitrary subset of factors. We derived training and test datasets from different analysis variants (Gene CN, ChrArm, ChrArm (avg.)), copy number events (gain, loss), and pan-cancer datasets (DepMap, ProCan, CPTAC), and trained the models on each training set (see "Methods").

Evaluating the performance of each model on its associated test set showed that multifactorial models perform better than using individual factors for predicting protein buffering, achieving a strong discrimination performance (Figs. 4A–C and EV4B; Dataset EV6). Moreover, the models trained on gene copy number-derived buffering classes consistently outperformed models using chromosome arm CNA-derived buffering classes. Especially upon gene copy number gain, the models showed a higher performance across datasets compared to copy number loss. This is likely because gene copy numbers provide more fine-grained copy number information, which is lost when using chromosome arm copy numbers. This information can then be used in conjunction with the BR to generate more accurate buffering classes that can be predicted more confidently. In addition, the performance of tumor sample-derived models (CPTAC, Gene CN gain: ROC AUC = 0.846, loss: ROC AUC = 0.811) surpassed the predictive performance of models trained on cell line data (DepMap, Gene CN gain: ROC AUC = 0.773, loss: ROC AUC = 0.684). Next, we evaluated the prediction performance of the models on unseen datasets (out-of-sample prediction). We observed that models trained on cell line datasets perform well on other cell line datasets upon gene copy number gain, but showed a weaker performance on tumor sample data and vice versa (Fig. EV4C; Dataset EV6). Based on these observations, we assume that factors affect protein buffering slightly differently in tumor samples and cell lines.

To determine how each factor contributed to the prediction of each model, we calculated SHapley Additive exPlanation values (SHAP values, Lundberg and Lee, 2017), which quantify the contribution of each feature to a model's prediction by attributing the difference between the actual and baseline predictions to individual features (Fig. EV4D,E; Dataset EV7). As SHAP values can be obtained for each factor and prediction, we calculated SHAP values on a subset of the model's test set using only observations where the model's prediction was correct. Negative SHAP values indicated that for a given protein in a sample, the model is more confident in classifying the observation as Buffered using the corresponding factor value. To quantify the direction of influence of a factor in predicting protein buffering and uncover biologically meaningful trends, we calculated the Spearman correlation between SHAP values and corresponding factor values for each factor. This approach achieved a stronger correlation than by directly correlating BRs and factor values as done in the unifactorial analysis $(\max(|\rho|) = 0.95)$. We observed that frequent participation in macromolecular complexes and a high mean gene dependency were associated with lower SHAP values and were therefore consistently predictive for *Buffered* proteins upon gene copy number gain and loss (Fig. 4D; Dataset EV6). This indicates that the model was more confident in classifying a protein as *Buffered* when it was essential across many samples and involved in numerous protein complexes. In contrast, a high number of TFs binding to a gene, a high fraction of TFs with activating regulatory modes, a high RAE frequency, and a high number of sites known for regulating a protein's function were consistently predictive for *Scaling* proteins upon gene copy number gain and loss. Interestingly, an increased number of ubiquitination sites was more predictive for *Buffered* proteins upon gain than upon loss in DepMap and ProCan, whereas in CPTAC an increased number of ubiquitination sites was more predictive for *Buffered* proteins upon

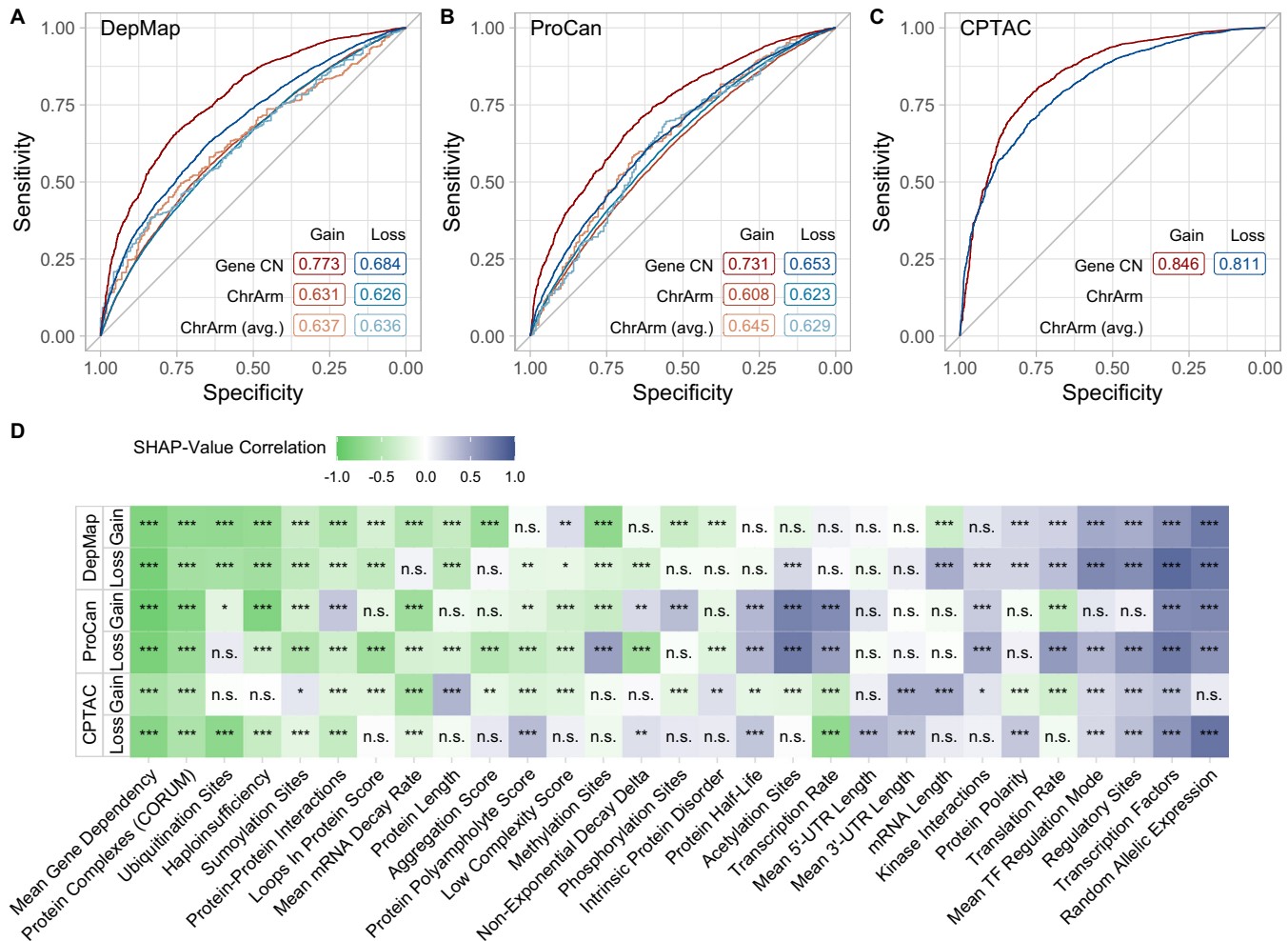

**Figure 4.  Explainable machine learning uncovers directional relationships between predictive factors and protein buffering.**

(A–C) ROC curves and AUCs of multifactorial xgbLinear models trained on predicting *Buffered* and *Scaling* classes for different copy number events, analysis variants and datasets. The ROC curves depict the model's performance on its respective test set (80/20 training/test-split). (D) Heatmap showing the correlation (Spearman's ρ) between SHAP values estimated on a random subset of the respective test set ($n = 300$) and their underlying factor values used for prediction. Negative correlation indicates a stronger contribution of a factor towards predicting observations as *Buffered* if the factor value is high. Correlation is shown for pan-cancer datasets (DepMap, ProCan, CPTAC) upon gene copy number gain and loss with significance levels indicated (Benjamini–Hochberg adjusted $P$ values, *$P_{adj}$ <0.01, **$P_{adj}$ <0.001, ***$P_{adj}$ < 0.0001).

gene copy number loss. We also observed this pattern in the unifactorial correlation analysis (Fig. EV3E).

In conclusion, multifactorial machine learning models outperformed unifactorial analyses in predicting protein buffering. We showed that increased mean gene dependency and protein complex participation contribute to a higher probability of protein buffering, whereas a higher number of activating TF interactions, protein regulatory sites, and higher RAE frequency contribute to a lower probability of protein buffering. These findings are consistent with previous unifactorial analyses (Schukken and Sheltzer, 2022); however, the machine learning approach substantially increases model confidence and enables the prediction of the directionality of each factor (Buffering vs. Scaling). Moreover, our multifactorial approach reveals more nuanced relationships, highlighting context-dependent differences across cancers and cancer cell lines.

## Cancer samples with high average buffering show altered differential expression patterns related to protein folding

Next, we investigated whether high average protein buffering in cell lines and tumor samples is associated with specific gene expression patterns. Using the previously calculated sample BRs, we split the samples into low- and high-buffering groups, and inspected which proteins were differentially expressed in high-buffering samples compared to low-buffering samples (Fig. 5A). We identified the cancer-drivers LMNA (structural protein of the nuclear lamina lamin A), EGFR (transmembrane tyrosine kinase receptor of the epidermal growth factor), RRAS (member of the RAS family of small GTPases), CTNNB1 (β-catenin, a key component of the Wnt signaling pathway), and many others, as significantly upregulated in high-buffering cell lines in both pan-cancer cell line datasets (DepMap, ProCan). Among proteins downregulated in high-

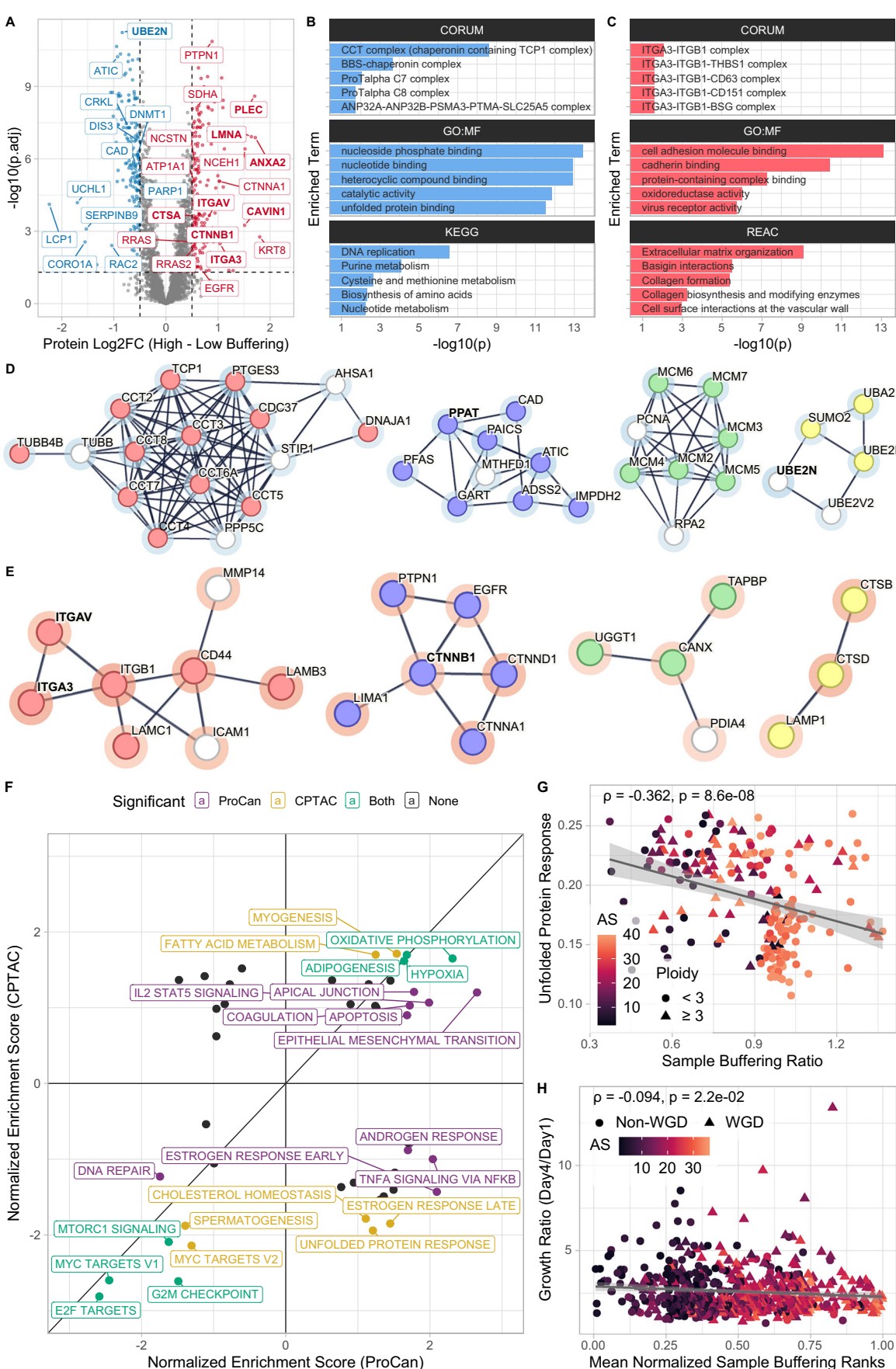

**Figure 5.  Cancer samples with high average buffering show altered differential expression patterns related to protein folding.**

(A) Volcano plot showing a log2 fold change between high (>80% quantile of sample buffering ratios, $n = 118$) and low (<20% quantile, $n = 118$) buffering cell lines in the ProCan dataset plotted against the Benjamini–Hochberg adjusted Student's $t$ test $P$ values. Labeled are genes with significant differential expression ($|Log2FC| > 0.5$, $P_{adj} < 0.05$). Genes with bold labels are significant in all cell lines and control datasets. (B, C) Over-representation analysis (ORA) of genes that are significantly down- (B) and upregulated (C) in highly buffering cell lines (hypergeometric test using g:Profiler). (D) Selected network clusters within the set of significantly downregulated genes in highly buffering cell lines. Highlighted gene sets: *unfolded protein binding* (red, GO:MF), *nucleotide biosynthesis* (blue, Reactome), *MCM complex* (green, GO:CC), and *protein sumoylation* (yellow, GO:BP). (E) Selected network clusters within the set of significantly upregulated genes in highly buffering cell lines. Highlighted gene sets: *ECM receptor interaction* (red, KEGG), *Cadherin binding* (blue, GO:MF), *unfolded protein binding* (green, GO:MF), and *lysosome* (yellow, KEGG). The rings around the dots indicate the log2-fold change (blue: negative, red: positive). (F) 2D enrichment plot comparing the normalized enrichment scores of MSigDB HALLMARK gene sets between CPTAC and ProCan. Significance cutoff of Benjamini–Hochberg adjusted $P$ values: $P_{adj} < 0.05$. (G) Scatter plot showing a negative correlation between sample buffering ratios and single-sample enrichment scores for the *Unfolded Protein Response* HALLMARK gene set using CPTAC tumor sample proteome data (Spearman's $\rho = -0.362$, $P = 8.6 \times 10^{-08}$). Color indicates the aneuploidy score of a tumor sample (see "Methods"). (H) Scatter plot showing a weak negative correlation between mean normalized buffering ratio ranks and the proliferation (day 1-to-day 4 growth ratio) of corresponding cell lines (Spearman's $\rho = -0.094$, $P = 0.022$). Color represents a cell line's aneuploidy score.

buffering cell lines in ProCan, we observed LCP1 (an actin-binding protein linked to hereditary cancer syndrome) and DNA methyltransferase DNMT1. ORA of the downregulated genes in cell lines with high average buffering revealed that terms related to protein folding, such as the chaperonin containing complex CCT, DNA replication, and nucleotide binding were enriched (Fig. 5B). The CCT/TRiC complex is involved in folding 10% of the proteome including actins, tubulins, and cell cycle regulators (Gestaut et al, 2019). Among the set of upregulated proteins, integrin complex partners, cell adhesion proteins, cadherin-binding proteins, and proteins involved in the organization of the extracellular matrix were enriched (Fig. 5C).

We noticed that blood cancers, which are frequently grown in suspension, have particularly low sample BRs. To avoid the confounding effects of the cell growth method, we created an adherent control by calculating the differential expression between high and low-buffering cell lines that were only grown adherently. Using this adherent control, cell adhesion, cell motility, and extracellular matrix organizing proteins were still upregulated, while nucleotide metabolism and biosynthesis proteins remained downregulated in high-buffering cells (Fig. EV5A–C). We furthermore found 27 genes that were commonly deregulated upon high buffering in all cell line and adherent control datasets (Dataset EV8).

STRING was used to visualize how differentially expressed genes in ProCan are involved in known and predicted protein–protein interactions (Szklarczyk et al, 2023; Fig. 5D,E). In addition to the CCT/TRiC complex, we observed that genes involved in nucleotide biosynthesis, protein sumoylation-related genes, and DNA replication factors associated with the MCM complex were downregulated (Fig. 5D). The deregulation of the MCM complex has been linked to increased genomic instability and to aneuploidy (Bochman and Schwacha, 2009; Passerini et al, 2016; Shima et al, 2007). Furthermore, both PCNA and UBE2N were downregulated; these genes play a role in error-free post-replication repair (Hoege et al, 2002; Kim et al, 2024; Moldovan et al, 2007; Motegi et al, 2008; Unk et al, 2008). In agreement with the ORA results, clusters containing integrins and proteins related to extracellular matrix (ECM) receptor interaction were upregulated (Fig. 5E). In particular, the integrins ITGA3, ITGA5, and ITGB1 were upregulated in highly buffering cell lines; their deregulation has been shown to alter cell adhesion and promote cell invasion (Fukushi et al, 2004; Mueller et al, 1999). Cadherin-binding proteins were upregulated as well,

including the oncogene EGFR, which positively regulates cell migration (Ghosh et al, 2010), and Catenin beta-1 (CTNNB1) involved in the regulation of cell adhesion (Brembeck et al, 2006; Kim et al, 2019). We observed that EGFR, among other oncogenes, had a higher BR upon chromosome arm loss, implying that the upregulation of EGFR in highly buffering cells is due to EGFR itself being buffered upon loss (Fig. EV5E–G). EGFR has been shown to frequently scale with chromosome arm gain (Schukken and Sheltzer, 2022), which has been confirmed in our analysis (Fisher's Exact test, $OR = 0.27$, $P = 7.9 \times 10^{-4}$). This suggests that cancer cell lines retain elevated EGFR levels independent of chromosome arm gain or loss. Furthermore, the lysosomal proteins Lysosomal-membrane-associated glycoprotein 1 (LAMP1), Cathepsin D (CTSD), and Cathepsin B (CTSB) were upregulated. CTSD and CTSB overexpression, as well as increased LAMP1 cell-surface expression, are associated with increased metastatic potential and tumor invasion (Jensen et al, 2013; Lee et al, 2024; Rochefort et al, 1990; Ruan et al, 2016; Sarafian et al, 1998). Similar to the downregulated genes, we also found clusters of proteins involved in unfolded protein binding in upregulated genes (UGGT1, CANX, TAPBP), playing a role in the retention and quality control of unfolded and misfolded proteins in the endoplasmic reticulum (Adams et al, 2020; Swanton et al, 2003). This, in combination with the downregulation of the CCT/TRiC complex, suggests an increase in protein folding quality control in cell lines with high average protein buffering, while demand for overall protein folding may decrease.

To further investigate which cancer hallmarks are deregulated in cells with high average buffering, we applied gene set enrichment analysis (GSEA) using the MSigDB hallmark gene sets and compared enrichment scores between tumor samples and cell lines. MYC target, G2M checkpoint, MTORC1 signaling, and E2F target gene sets were significantly downregulated in both high-buffering tumor samples and cell lines, while the DNA repair gene set was only significantly downregulated in cell lines (Fig. 5F). Furthermore, the gene set related to epithelial–mesenchymal transition (EMT) was significantly upregulated in cell lines, matching the upregulation of Cadherin-binding proteins. As we identified aneuploidy as a potential confounding variable for BR-based analyses, we also performed GSEA on differential expression data between high- and low-aneuploid samples. In this comparison, E2F target and MTORC1 signaling pathways were upregulated, while EMT pathways were downregulated in highly aneuploid

tumor samples. This contrast shows that the downregulation of E2F and MTORC1 signaling, together with the upregulation of EMT pathways, was unique to samples with high average buffering and independent of aneuploidy (Fig. EV5H; Dataset EV8). Furthermore, the EMT pathway was upregulated more strongly in high-buffering compared to high-aneuploidy samples.

Highly aneuploid cell lines and primary tumors suffer from increased proteotoxic stress and have an increased Unfolded Protein Response (UPR) (Ippolito et al, 2024; Oromendia and Amon, 2014). We observed that the UPR gene set was significantly downregulated in tumor samples with high average buffering, but not in cell lines, suggesting that tumor samples could use protein buffering to alleviate proteotoxic stress and do so more effectively than cell lines (Fig. 5F). To test this hypothesis, we applied single-sample GSEA (ssGSEA, Barbie et al, 2009) to CPTAC using the UPR hallmark gene set and correlated the single-sample enrichment scores with the sample BR. Indeed, tumor samples that exhibit high average buffering showed a decreased UPR, especially in highly aneuploid WGD-negative samples, implying that protein buffering reduces proteotoxic stress (Fig. 5G; Dataset EV8). Sub-tumor heterogeneity and low tumor purity could potentially confound these results, but the UPR enrichment score and tumor purity were not significantly correlated (Spearman's $\rho = -0.034$, $P = 0.63$; Fig. EV5D), and tumors with a purity above 70% exhibited an increased UPR (UPR > 0.2) less frequently than low-purity tumors ($OR = 0.41$, $P = 0.021$). Furthermore, the correlation between sample BR and UPR enrichment score increased when removing tumor samples with a purity below 50% (Spearman's $\rho = -0.43$, $P = 6.4 \times 10^{-3}$), suggesting that the results are not confounded by tumor samples with low purity.

Therefore, we asked whether cell lines exhibiting high levels of protein buffering could have a proliferative advantage, as they might be able to better mitigate the proteotoxic stress introduced by aneuploidy. Using growth rates from DMSO-treated cell lines in the GDSC drug screen dataset (Iorio et al, 2016) showed that some cell lines proliferated slightly worse the more aneuploid they were (particularly WGD-positive cell lines: Spearman's $\rho = -0.270$, $P = 8.2 \times 10^{-6}$). However, we did not observe a strongly significant correlation between growth rates and average buffering estimates in cell lines (Fig. 5H). We conclude that high buffering alters protein homeostasis in cell lines, affecting DNA repair, protein folding, and cell adhesion pathways. Interestingly, increased buffering did not provide a general proliferative advantage.

## CRISPR knock-out screens reveal vulnerabilities of weakly buffering cells

As high average protein buffering does not appear to provide a general proliferative advantage in aneuploid cell lines, we investigated whether these cell lines show differential dependency on certain genes or if increased buffering mitigates aneuploidy-induced vulnerabilities. We divided the cell lines into groups with high and low average buffering (>80%, <20% sample BR MNR) and compared the dependency scores of genes between these groups. The dependency scores were derived from CRISPR-KO screens and quantify how dependent a cell line is on a particular gene. Cell lines with high average buffering were more dependent on the oncogenes CRKL (a protein kinase possibly activating RAS and JUN) and Hippo signaling pathway transcription coactivator WWTR1. On

the other hand, they are less dependent on the oncogenes MDM4 (a p53 regulator), TAF1 (TATA binding factor); tumor suppressor genes POT1 (telomere-protecting protein), SDHD (subunit of the mitochondrial respiratory chain), and a sensor of oxidative stress KEAP1 (Fig. 6A; Dataset EV9). Furthermore, ITGAV, FERMT2, PTK2, GPR61, and RELA were exclusively essential in high-buffering cells, while CENPI, DROSHA, RNF40, TUBD1, TMX2, RAD1, DPY30, PTBP1, TBCE, MED10, WDR48, TUBE1, and NAMPT were exclusively essential in low-buffering cell lines.

ORA of the genes with significant dependency score differences showed that cell lines with high average buffering were less dependent on DNA damage response, DNA repair, and cellular stress response genes than cell lines with low average buffering (Fig. 6B). Interestingly, high-buffering cells were more dependent on cell adhesion regulation, substrate adhesion-dependent cell spreading, kinase binding, and integrin binding genes (Fig. 6C). As before, we created a control subset of cell lines that were adherently grown, and compared the differential dependency between high and low-buffering cell lines within this subset (Fig. EV6A; Dataset EV9). ORA showed that general transcription initiation factor binding genes were less essential in high-buffering cell lines (Fig. EV6B). Furthermore, genes that remained exclusively essential in low-buffering cell lines after controlling for the growth method were CENPI (Centromere Protein I), RAD1 (RAD1 Checkpoint DNA Exonuclease), TUBD1 (Tubulin Delta 1), and TBCE (Tubulin Folding Protein). This analysis further highlights the physiological differences between high and low-buffering cell lines.

Next, we analyzed whether samples with different degrees of average protein buffering exhibit an altered sensitivity to drug treatment. We collected drug effect scores quantifying the changes in cell viability between drug and DMSO treatment from the PRISM Repurposing dataset (Corsello et al, 2020; Data ref: DepMap, Broad Institute, 2023), where lower values indicate a decreased growth of the cells after drug treatment compared to the control using DMSO. We categorized cell lines as drug-resistant if their median drug effect was higher than 80% of the drug effect scores, while cell lines with a median drug effect score below the 20% percentile were categorized as drug-responsive. As before, we also categorized cell lines as high and low-buffering based on the 80% and 20% percentiles of the sample BR MNR. Remarkably, the median drug effect on cell viability of high-buffering cells is closer to the drug-resistant cell cohort, and the drugs have a significantly lower median effect on the viability of high-buffering cells compared to cell lines with low average protein buffering, although this difference is small (Fig. 6D). To test whether high buffering and drug resistance are frequently co-occurring, we separated all available cell lines by median sample BR MNR and median drug effect. This showed that high-buffering cell lines were more frequently drug-resistant than cell lines with a sample BR MNR below median (Fisher's exact test, $OR = 1.525$, $P = 2.1 \times 10^{-2}$). The increased drug resistance was largely explained by increased aneuploidy, showing that there is only a marginal general effect of buffering on drug sensitivity independent of aneuploidy (Fig. EV6C,D). However, high-buffering samples were still slightly more frequently drug-resistant independent of aneuploidy status (Fig. EV6E; High Aneuploidy: $OR = 1.088$, $P = 0.021$; Low Aneuploidy: $OR = 1.182$, $P = 0.57$). This means that, dependent on the degree of aneuploidy, there is still a small effect on drug sensitivity that can be attributed to buffering. To understand these differences,

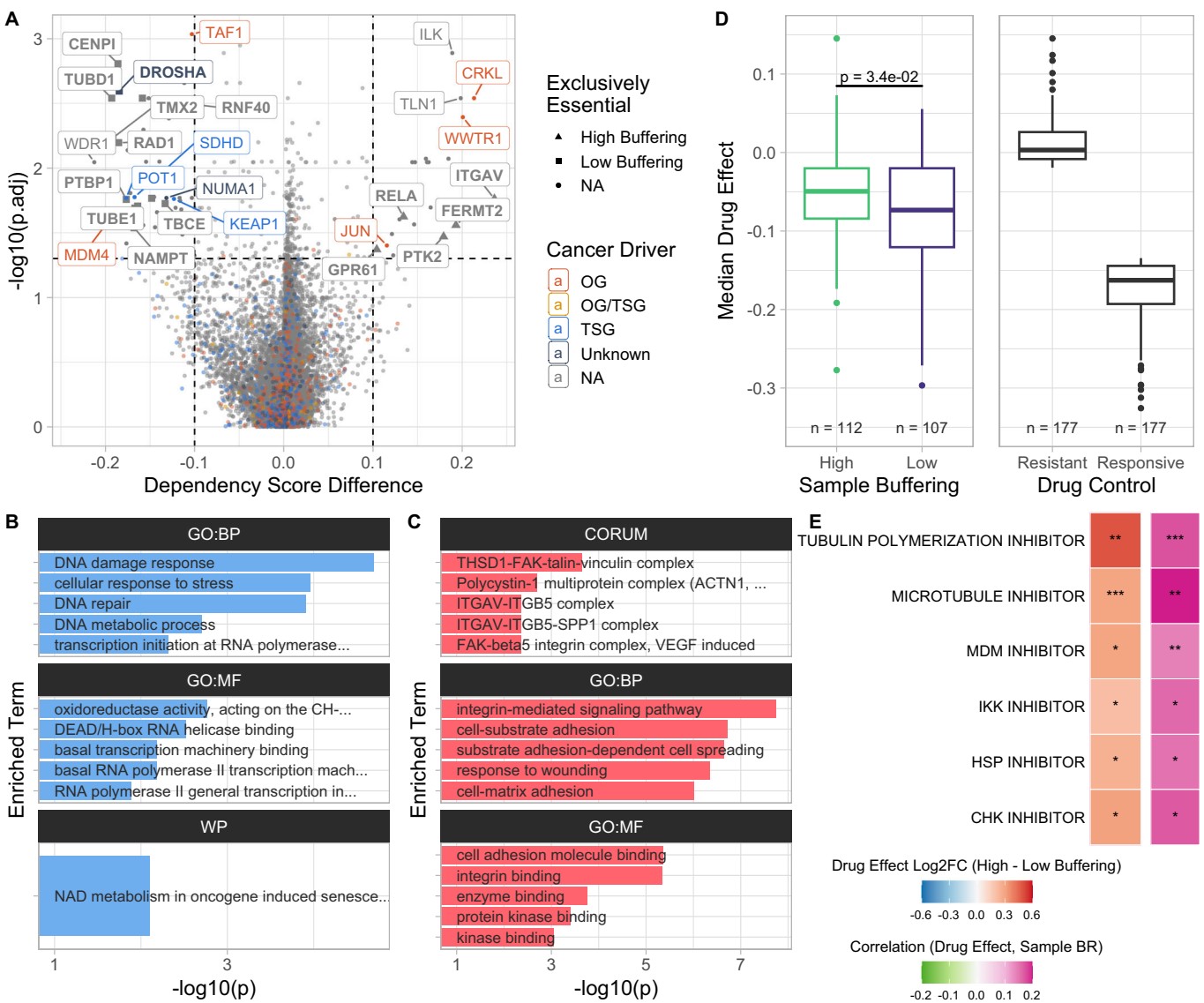

**Figure 6. CRISPR knock-out screens reveal vulnerabilities of weakly buffering cells.**

(A) Volcano plot showing the difference of the CRISPR-KO dependency score between high (>80% quantile of sample BR MNRs, $n = 134$) and low (<20% quantile, $n = 134$) buffering cell lines plotted against Benjamini–Hochberg adjusted Wilcoxon rank-sum test $P$ values. (B, C) Over-representation analysis (ORA) of genes with significantly reduced (B) and increased (C) CRISPR-KO dependency scores in highly buffering cell lines (hypergeometric test using g:Profiler). (D) Median drug effect on cell viability across drugs from the PRISM Repurposing dataset for high (>80% quantile of sample BR MNRs) and low (<20% quantile) buffering cell lines. Cell lines resistant (>80% quantile of median drug effect) and responsive (<20% quantile) to drug treatment are shown as controls. Drug effect describes the Log2FC in cell viability after drug treatment (compared to DMSO; Corsello et al, 2020). Boxes represent the interquartile range (IQR) with the central line indicating the median. The whiskers extend to the data points within 1.5×IQR. $P$ values were determined using a two-tailed Wilcoxon rank-sum (Mann–Whitney $U$) test. (E) Drug mechanisms with significant log2 fold change of mean drug effect between high and low-buffering cell lines (Student's $t$ test, |Log2FC| > 0.2, $P_{adj}$ < 0.05) and significant correlation between sample BR MNRs and drug effect (Spearman's ρ) with significance levels shown (Benjamini–Hochberg adjusted $P$ values, *$P_{adj}$ < 0.05, **$P_{adj}$ < 0.01, ***$P_{adj}$ < 0.001).

we analyzed individual drugs and groups of drugs by mechanism of action. While highly buffering cell lines showed no sensitivity to any group of drugs, the low-buffering cell lines were significantly more sensitive to HSP, CHK, MDM, IKK, microtubule, and tubulin polymerization inhibitors (Fig. 6E). Thus, cells that cannot efficiently buffer protein imbalance arising from aneuploidy are particularly sensitive to drugs that target protein folding, mitotic spindle function, innate immune response, and DNA damage

response. As a control, we analyzed the drug effect difference between high and low-aneuploid cells. Drug mechanisms that were not significant in highly aneuploid cells and were unique to low-buffering cells were CHK and HSP inhibitors (Dataset EV10). In conclusion, while the general increase in drug resistance can be largely explained by aneuploidy, there is an aneuploidy-independent resistance to heat shock protein and checkpoint kinase inhibitors that can be attributed to enhanced buffering.

# Discussion

Abnormal chromosomal content, or aneuploidy, strongly alters protein homeostasis due to aberrant expression of genes located on the affected chromosomes. Yet, a significant fraction of the aberrant expression caused by altered gene dosage is mitigated by adjusting the protein abundance (Chunduri et al, 2021; Dephoure et al, 2014; Stingele et al, 2012). This means protein abundance in aneuploid cells often resembles diploid levels rather than the abundance expected based on the gene dosage.

Previous research indicated that gene dosage buffering is widespread in aneuploid cell lines and that its degree is affected by certain gene and protein features, such as the number of ubiquitylation sites, or participation in macromolecular complexes (Schukken and Sheltzer, 2022). To obtain additional insight into the factors affecting dosage compensation, we established a novel indicator we termed "Buffering Ratio" (BR), where we quantified the protein-level attenuation for each gene within individual samples. This approach was designed to deliver a gene copy number-sensitive quantification of protein abundance changes in aneuploid samples that may be "averaged out" when using protein-average Log2FC cutoffs as in Schukken and Sheltzer (Fig. 1B). While this approach allows us to evaluate the effect of whole-genome doubling, or identify variability of buffering, it also has its limitations. In particular, the BR is sensitive to noise from both proteomics and copy number data. To circumvent these issues, we calculated confidence scores for the BRs and removed BR measurements with low confidence.

Using this approach, we demonstrate that gene dosage compensation occurs not only in aneuploid in vitro cancer cell lines but also extensively and efficiently in aneuploid in vivo tumor samples (Fig. 1). The widespread phenomenon of gene dosage buffering highlights the importance of maintaining protein home-ostasis despite aberrant chromosome and gene copy numbers. While buffering in pan-cancer cell lines and tumor samples is comparable, chromosome-engineered aneuploid human cell lines are buffered to a greater extent. The strong dosage compensation in engineered human cell lines is in contrast with recent observations that natural yeast strains tolerate aneuploidy more effectively than lab-engineered strains (Muenzner et al, 2024). This indicates that aneuploidy in human cells causes high fitness costs and enforces dosage compensation early. Additionally, distinct genomic altera-tions in each sample type may play a role: engineered cell lines exhibit gains and losses of individual chromosomes, whereas cancer samples typically harbor copy number variations affecting multiple chromosomes, as well as a plethora of mutations.

To identify factors that contribute to gene dosage buffering, we applied both unifactorial and multifactorial models to analyze their effects and predictive capacity. While unifactorial models predict factors contributing to buffering with a low predictive power (~0.6 ROC AUC, Fig. 3; Schukken and Sheltzer, 2022), our novel multifactorial ML models performed significantly better, with up to 0.85 ROC AUC (Fig. 4). Using these models, we established gene dependency as the key factor in gene dosage buffering after chromosome or gene gains and losses in all analyzed datasets. This demonstrates that the abundance of essential genes must be maintained at optimal levels independently of the corresponding gene copy number. Additionally, being a member of macromole-cular complexes, as well as having a high number of ubiquitylated

sites, is strongly predictive of buffering, as previously observed in other works (Figs. 3 and 4; Ishikawa et al, 2017; Schukken and Sheltzer, 2022; Stingele et al, 2012). This supports the hypothesis that protein buffering is in parts conveyed by the degradation of non-stochiometric subunits of macromolecular complexes. In contrast, scaling was affected mainly by transcriptional features (transcription factors, bi-allelic expression, etc.) and mRNA turnover. Remarkably, we found a different predictive power of individual factors for buffering protein abundance upon gene/chromosome gain and loss (Fig. 3). This suggests that chromosome gains and losses may affect cellular physiology differently and mitigation of the protein imbalance occurs partly by different mechanisms. Out-of-sample prediction showed that our models performed well across datasets, highlighting the reproducibility of our results. However, the predictiveness differed between ML models trained on cell line datasets compared to tumor sample-trained models. This suggests that tumor samples buffer changes in protein abundance to a different extent than cell lines do.

Our approach of buffering ratio quantification also allowed us to differentiate the consequences of gene copy number changes among cell lines and tumor types. Some cell lines were extremely effective in buffering, while others did not compensate for the changes in copy numbers as much (Fig. 2). Tumor samples also showed this variability. Cell lines grown in suspension showed lower levels of average buffering than cell lines with adherent growth patterns. Interestingly, our analysis suggests that aneu-ploidy is a strong confounding factor of gene dosage buffering, as we observed a strong correlation of the BR with the respective aneuploidy score (AS) of each cell line. Separating aneuploidy and buffering is difficult, as only aneuploid cells present a protein imbalance that can be buffered. The fact that highly aneuploid cells buffer the gene dosage more efficiently than cells with low AS may also suggest that aneuploid cells adaptively rewire their gene expression network to minimize the impact of gene dosage changes. Other possible confounding factors (e.g., p53 status) showed negligible effects. We hypothesize that relatively low levels of buffering in pediatric cancers suggest that these cancers developed quickly and have not adapted to altered protein homeostasis. In contrast, the cancer types with increased buffering suggest an etiology related to environmental and age-related factors. These cells might have gained different aneuploidies in multiple steps and adapted to aneuploidy-induced stresses through buffering, leading to a more balanced proteostasis.

We found that cell lines with highly efficient gene dosage buffering downregulated the chaperonin complex CCT/TRiC, replication factors, and factors required for nucleotide biosynthesis. In contrast, pathways related to integrin signaling, epithelial–mesenchymal transition, extracellular matrix, and cell adhesion were upregulated in highly buffering aneuploid cells, even when controlling for confounding factors. Our analyses point out that protein folding plays a critical role in buffering, and that buffering correlates with DNA and RNA metabolism, as well as with extracellular matrix regulation (Fig. 5). But which of these consequences are due to the efficient buffering, and what is a cellular response to aneuploidy? To disentangle these interlinked phenomena, we performed gene set enrichment analysis (GSEA) for high vs. low-buffering samples and compared the results with analysis in high vs. low-aneuploid samples. This approach clearly shows that E2F and mTORC1 targets are differentially regulated in

samples with a high-buffering ratio, independently of aneuploidy. Tumor samples with higher average buffering also showed a reduced unfolded protein response, particularly in WGD-negative, but highly aneuploid cells. Controlling for the confounding effects of tumor heterogeneity confirmed these results, indicating that buffering alleviates proteotoxic stress introduced by aneuploidy in tumor samples. In contrast, the unfolded protein response was mixed in cell lines, with some protein folding genes upregulated and translation-related genes downregulated, indicating heterogeneity in stress adaptation mechanisms. The upregulation of oncogenes in high-buffering cells suggests that buffering affects the oncogenic potential of cells, in particular by increasing the abundance EGFR upon loss of chromosome arm 7p (Figs. 5A and EV5E–G).

The differences in buffering capacity between tumor samples and cell lines, as well as between cancer types, raised the question of whether efficient buffering influences their growth, survival, and response to treatment. This was suggested by recent observations that naturally occurring adapted aneuploid strains of *Saccharomyces cerevisiae* efficiently buffer the gene dosage by increasing their ubiquitin–proteasome-dependent protein turnover (Muenzner et al, 2024). Intriguingly, we observed no direct improvement of proliferation in efficiently buffering cell lines compared to low-buffering cells. In fact, highly aneuploid cancer cell lines showed decreased proliferation compared to low-aneuploid cells in vitro. Furthermore, we observed that increased buffering is associated with high aneuploidy. We, therefore, propose that buffering mitigates the fitness costs of aneuploidy and thus enables increased tolerance to aneuploidy. It should be noted that this analysis has been performed only for cell lines in a pan-cancer analysis that does not control for genetic and tissue-specific differences. The conditions may be different from in vivo tumor growth. However, there are no data currently available that would allow us to directly test the hypothesis that increased buffering is an adaptive feature and facilitates the proliferation of aneuploid tumors.

Finally, we considered the possibility that the buffering capacity may affect gene dependency or drug sensitivity. Strikingly, we found that high-buffering cells exhibit increased drug resistance and are less dependent on genes related to DNA damage repair and response to cellular stresses, but more dependent on genes involved in cell adhesion. This pattern aligns with their differential expression profiles (Figs. 5 and 6), but the surprising link between cell adhesion and gene dosage buffering remains unclear. We note that increased aneuploidy is likely the dominating cause of increased drug resistance in high-buffering cells (Andrade et al, 2023). In contrast, low-buffering cells are generally more sensitive, in particular to HSP, MDM, CHK, IKK, microtubule, and tubulin polymerization inhibitors. Intriguingly, low-buffering cells remained sensitive to HSP and CHK inhibitors even when controlling for aneuploidy. These findings support the hypothesis that low-buffering cells experience greater proteotoxic stress and that inhibiting chaperones in these cells further exacerbates these stresses, thereby reducing growth. Furthermore, TUBD1 (Tubulin Delta 1), TBCE (tubulin folding protein), and CENPI (Centromere Protein I) participate in the correct formation of the mitotic spindle and are essential in low-buffering cells, suggesting that cells with weakly buffered protein abundance rely on the proper progression of mitosis, as they cannot tolerate increased aneuploidy.

Based on our results, we hypothesize that cells that efficiently buffer changes in protein abundance have an adaptive advantage by altering the adhesion to the extracellular matrix and alleviating aneuploidy-induced proteotoxic stress. Future research should uncover the links between gene dosage buffering and further factors identified in our analysis.

## Methods

**Reagents and tools table**

| Reagent/resource | Reference or source | Identifier or catalog number |
| --- | --- | --- |
| **Experimental models** | | |
| hTERT RPE1 p53−/− | Mardin et al, 2015 | |
| **Recombinant DNA** | | |
| **Antibodies** | | |
| **Oligonucleotides and other sequence-based reagents** | | |
| **Chemicals, enzymes, and other reagents** | | |
| RO3306 | Sigma-Aldrich | SML0569 |
| GSK923295 | Biomol | 1088965-37-0 |
| Reversine | Enzo Life Sciences | LKT-R1885 |
| Gibco™ Penicillin-Streptomycin | Thermo Fisher Scientific | 15140122 |
| Gibco™ Trypsin-EDTA | Thermo Fisher Scientific | 25200056 |
| Gibco™ DMEM, high glucose, GlutaMAX™ Supplement, pyruvate | Thermo Fisher Scientific | 10566016 |
| Gibco™ Fetal Bovine Serum | Thermo Fisher Scientific | A5670701 |
| **Software** | | |
| MaxQuant 2.0.1.0 | Max Planck Institute of Biochemistry, Martinsried, Germany | |
| R 4.3.2 | R Foundation for Statistical Computing, Vienna, Austria, https://www.R-project.org/ | |
| Bioconductor 3.18 | https://doi.org/10.1038/nmeth.3252 | |
| ggplot2 3.5.2 | R package; https://ggplot2.tidyverse.org | |
| ggpubr 0.6.1 | R package; https://CRAN.R-project.org/package=ggpubr | |
| pheatmap 1.0.13 | R package; https://CRAN.R-project.org/package=pheatmap | |
| viridis 0.6.5 | R package; https://doi.org/10.5281/zenodo.4678327 | |
| dplyr 1.1.4 | R package; https://CRAN.R-project.org/package=dplyr | |
| tidyr 1.3.1 | R package; https://CRAN.R-project.org/package=tidyr | |
| stringr 1.5.1 | R package; https://CRAN.R-project.org/package=stringr | |

| Reagent/resource | Reference or source | Identifier or catalog number |
| --- | --- | --- |
| purrr 1.1.0 | R package; https://CRAN.R-project.org/package=purrr | |
| TCGAbiolinks 2.30.4 | R package; https://doi.org/10.1093/nar/gkv1507 | |
| decoupleR 2.8.0 | R package; https://doi.org/10.1093/bioadv/vbac016 | |
| biomaRt 2.58.2 | R package; https://doi.org/10.1093/bioinformatics/bti525 | |
| AnnotationDbi 1.64.1 | R package; https://doi.org/10.18129/B9.bioc.AnnotationDbi | |
| org.Hs.eg.db 3.18.0 | R package; https://doi.org/10.18129/B9.bioc.org.Hs.eg.db | |
| HGNChelper 0.8.15 | R package; https://doi.org/10.12688/f1000research.28033.2 | |
| mskcc.oncotree 0.1.1 | R package; https://CRAN.R-project.org/package=mskcc.oncotree | |
| limma 3.58.1 | R package; https://doi.org/10.1093/nar/gkv007 | |
| GSVA 1.50.5 | R package; https://doi.org/10.1186/1471-2105-14-7 | |
| fgsea 1.28.0 | R package; https://doi.org/10.1101/060012 | |
| gprofiler2 0.2.3 | R package; https://doi.org/10.1093/nar/gkad347 | |
| STRINGdb 2.14.3 | R package; https://doi.org/10.1093/nar/gkac1000 | |
| survival 3.5-7 | R package; https://CRAN.R-project.org/package=survival | |
| survminer 0.5.0 | R package; https://CRAN.R-project.org/package=survminer | |
| msigdbr 25.1.0 | R package; https://CRAN.R-project.org/package=msigdbr | |
| shapr 1.0.4 | R package; https://doi.org/10.21105/joss.02027 | |
| pROC 1.18.5 | R package; https://doi.org/10.1186/1471-2105-12-77 | |
| xgboost 1.7.11.1 | R package; https://CRAN.R-project.org/package=xgboost | |
| randomForest 4.7-1.2 | R package; https://CRAN.R-project.org/package=randomForest | |
| caret 7.0-1 | R package; https://doi.org/10.18637/jss.v028.i05 | |
| arrow 20.0.0.2 | R package; https://CRAN.R-project.org/package=arrow | |
| openxlsx 4.2.8 | R package; https://CRAN.R-project.org/package=openxlsx | |
| **Other** | | |
| Pierce™ High pH Reversed-Phase Peptide Fractionation Kit | Thermo Fisher Scientific | 84868 |
| TMT10plex™ Isobaric Labeling Reagent Set | Thermo Fisher Scientific | 90110 |

| Reagent/resource | Reference or source | Identifier or catalog number |
| --- | --- | --- |
| Mass spectrometer Q ExactiveTM HF | Thermo Fischer Scientific | |
| EASY-nLC 1200 System | Thermo Fischer Scientific | |

## Cell line generation

p53 deficient hTert immortalized human retinal pigment epithelium (RPE1p53−/−; Mardin et al, 2015) cell lines were cultured in DMEM supplemented with GlutaMAX™, 10% fetal bovine serum (FBS; Gibco), and 1% penicillin–streptomycin (Gibco). All cell lines were cultured at 37 °C in a humidified incubator with 5% $CO_2$. To generate aneuploid cell lines, we used an adapted protocol based on a combination of CENP-E and MPS1 inhibition (Soto et al, 2017). Stable cell lines were sequenced to determine the karyotype. The obtained aneuploid cell lines were labeled RM13 (monosomy 13) and Rtr13 (trisomy 13). Cell lines were routinely checked for mycoplasma contamination (MycoStrip detection kit by Invivo-Gen) and authenticated through sequencing.

## MS sample preparation and labeling of peptides by tandem mass tags (TMT)

Peptide preparation and TMT labeling were carried out following the manufacturer's protocol. Briefly, cells were harvested by trypsinization from the plates. In total, $10^6$ cells were washed twice with PBS, pelleted, and stored at −80 °C. Cells were lysed by adding 100 µL lysis buffer (10% SDS in 100 mM triethylammonium bicarbonate (TEAB)) using strong ultrasonication. The lysates were clarified via centrifugation at 16,000×g for 10 min at 4 °C, and protein concentrations were determined using the BCA protein assay kit (Thermo Scientific). In all, 50 µg of protein were reduced using 5 mM Tris 2-carboxyethylphosphine (TCEP) for 1 h at 55 °C and alkylated with 10 mM iodoacetamide in the dark for 30 min at 25 °C. The proteins were then precipitated overnight at −20 °C by adding six volumes of acetone. After precipitation, the proteins were resuspended in 100 mM TEAB (pH 8.5) and digested overnight at 37 °C with sequencing-grade modified trypsin. For TMT labeling, the trypsin-digested peptide samples were labeled with isobaric tags (TMT 10-plex, Thermo Fisher Scientific), with individual tags assigned as follows: RM13_01 (TMT126), RM13_02 (TMT127N), RM13_03 (TMT127C), RPE1_p53−/−_01 (TMT128N), RPE1_p53−/−_01 (TMT1228C), RPE1_p53−/−_03 (TMT129N), Rtr13_01 (TMT129C), Rtr13_02 (TMT130N), Rtr13_03 (TMT130C), and an equal mixture of all cell samples (TMT131). After labeling, the reaction was quenched by adding 5% hydroxylamine and incubation for 15 min at room temperature. All samples were then combined in equal amounts and dried under vacuum. Pooled peptides were fractionated into eight fractions (5, 10, 12.5, 15, 17.5, 20, 22.5, 25, 50% ACN) using the Pierce™ High pH Reversed Phase Peptide Fractionation Kit (Thermo Scientific). Each fraction was dried under vacuum, resuspended in buffer A (0.1% formic acid) containing 0.01% trifluoroacetic acid, and analyzed using nanoflow liquid chromatography coupled to a Q-Exactive HF mass spectrometer (Thermo Fisher Scientific).

**Table 1.  Factors used for the dosage compensation factor analysis grouped by data source.**

| Factors | Preprocessing | Source |
|---|---|---|
| Protein–protein interactions | Counted the number of interactions per protein | HIPPIE, Version:v2.3 (Alanis-Lobato et al, 2017) |
| Protein complexes | Counted the number of complexes a protein participates in | CORUM Version: 210512 (Giurgiu et al, 2019) |
| Mean 3′-UTR Length, Mean 5′-UTR Length | Difference between transcription start and coding region start (5′-UTR), and coding region end and transcription end (3′-UTR), averaged per gene | UCSC Human Genome Browser Track: NCBI RefSeq Assembly: GRCh38/hg38 (Data ref: UCSC Genome Bioinformatics Group, 2023) |
| Phosphorylation, Ubiquitination, Sumoylation, Methylation, Acetylation, and Regulatory Sites; Kinase Interactions | Counted the number of sites/interactions per protein | PhosphoSitePlus®, www.phosphosite.org (Hornbeck et al, 2015) |
| mRNA/Protein Abundance, Transcription/ Translation Rate, Protein/mRNA Length | – | humanRates.ods (Data ref: Hausser, 2019) |
| Intrinsic Protein Disorder, Low Complexity Score, Homology Score, Loops In Protein Score, Protein Polyampholyte Score, Protein Polarity | – | MobiDB Version: 5.0 Release: 2022_07 Proteome: UP000005640 (Piovesan et al, 2021) |
| Non-Exponential Decay Delta | Mean per protein | Supplementary Table 4 in  McShane et al, 2016 |
| Mean mRNA Decay Rate | Mean per gene | Supplementary Table 9 in Yang et al, 2003 |
| Protein Half-Life | Mean per gene | Supplementary Table 2 in Mathieson et al, 2018 |
| Aggregation Score | – | Supplementary Table 2 in Ciryam et al, 2013 |
| Haploinsufficiency/ Triplosensitivity Score | – | Supplementary Table S7 in Collins et al, 2022 |
| Mean Gene Dependency | Mean dependency score per gene | DepMap (DepMap, Broad Institute, 2023) |
| Transcription Factors (Repressor/Activator/ All), Mean TF Regulation Mode | Counted the number of transcription factors (TFs) per gene. TFs with a negative mode of regulation (MOR) were counted as repressors, TFs with positive MOR as activators. Calculated the mean MOR per gene. | Obtained DoRothEA regulons using the decoupleR package (levels A,B,C,D). (Badia-i-Mompel et al, 2022; Garcia-Alonso et al, 2019) |
| Random Allelic Expression | Combined male and female z-score columns of hc-RAE and hc-Biallelic (all tissues), and calculated mean per gene | Supplementary Table S2 in Kravitz et al, 2023 |

Preprocessing describes how datasets have been transformed to be used as factors.

Peptide separation was carried out with an EASY-nLC 1200 ultra-high-pressure liquid chromatography system using chromatography columns (50 cm, 75 mm inner diameter) packed in-house with ReproSil-Pur C18-AQ 1.9μm resin (Dr. Maisch GmbH). Peptides were loaded in buffer A (0.1% formic acid) and eluted with a non-linear gradient of 5-100% buffer B (0.1% formic acid, 80% acetonitrile) at a rate of 250 nL/min over 180 min. Column temperature was maintained at 60 °C. Data acquisition alternated between a full scan (120 K resolution, maximum injection time of 8 ms, AGC target of 3e6) and 15 data-dependent MS/MS scans (60 K resolution, maximum injection time of 100 ms, AGC target of 1e5). The isolation window was 0.7 *m/z*, and normalized collision energy was 32. Dynamic exclusion was set to 30 s, and the fixed first mass was set to 100 *m/z*. Mass spectrometry data were processed using MaxQuant software (version 2.0.1.0). The data were searched against the human reference proteome database (UniProt: UP000005640) with a false discovery rate (FDR) of less than 1% for peptides and proteins. All raw files and MaxQuant output tables have been deposited in the PRIDE repository.

## Data acquisition and preprocessing

We note that different datasets use different types of identifiers to refer to a gene or protein. For all datasets, we mapped different ID types to HGNC gene symbols and UniProt IDs using the biomaRt and AnnotationDbi R-packages, and updated gene symbols with the HGNChelper package (Durinck et al, 2005; Durinck et al, 2017; Hervé Pagès, 2017; Oh et al, 2022).

### Dosage compensation factor datasets

We used a series of data sources to identify the factors affecting dosage compensation. In Table 1, we list the datasets, how the data in the dataset has been processed, and their usage for each factor. Missing values of counted factors (e.g., known PTM sites, protein interactions, protein complex participation) were imputed with 0. The factor datasets were joined by gene symbol and UniProt ID. Genes with more than 25% missing values across features were removed.

### DNA copy number datasets

For the cell line datasets DepMap and ProCan, we obtained absolute gene copy numbers generated with the ABSOLUTE algorithm and chromosome arm copy number alteration (CNA) calls from the DepMap CCLE 23Q2 release (Carter et al, 2012; Cohen-Sharir et al, 2021; Data ref: DepMap, Broad Institute, 2023). The absolute gene copy numbers for the in vivo tumor samples were downloaded from the genomic data commons (GDC) data portal of the National Cancer Institute (NCI) at the National

Institutes of Health (NIH), a part of the U.S. Department of Health and Human Services, using the TCGAbiolinks R package (Antonio Colaprico, 2017; Colaprico et al, 2016; Mounir et al, 2019; Silva et al, 2016). The copy numbers were generated using the AscatNGS algorithm, which infers the absolute copy number by comparing whole-genome sequencing (WGS) data between tumor samples and matched normal samples (Raine et al, 2016). The parameters used for the GDC query were: project = "CPTAC-3", workflow.type = "AscatNGS", data.category = "Copy Number Variation", data.type = "Gene Level Copy Number".

### Proteomics datasets

Pan-cancer cell line proteome and model annotation data were downloaded from the DepMap CCLE project of the Broad Institute (Dephoure et al, 2014; Data ref: DepMap, Broad Institute, 2023; DepMap portal; 375 cell lines) and the Cell Model Passports ProCan project of the Wellcome Sanger Institute (Gonçalves et al, 2022; Cell Model Passports portal; 949 cell lines). Pan-cancer proteome and clinical data for in vivo tumor samples were downloaded from the CPTAC project of the NCI (Li et al, 2023; Proteomic Data Commons; 1026 tumor samples, 523 normal tissue samples; BCM harmonized dataset). The proteomes of engineered cell lines were measured in three biological replicates for each RM13, Rtr13, and the parental control RPE1p53−/− (dataset P0211; Fig. EV7). The dataset of engineered cell lines was created by combining P0211 with previously published data from cell lines: RPE1 p53 KO, RM10;18, RM13, RM19p (Chunduri et al, 2021; PRIDE Accession: PXD018440).

We only used cell lines and tumor samples with proteomics data with matched chromosome arm or gene copy number data. From all pan-cancer proteomics data, we removed measurements from proteins encoded on sex chromosomes. To avoid a disproportional influence of measurement noise, we removed values below 0.1% of protein abundance measurements within each dataset. To ensure the reproducibility of our results across datasets, we used protein–protein reproducibility ranks and removed proteins with reproducibility ranks below the 10% percentile (Upadhya and Ryan, 2022). We normalized the measurements between samples within a dataset with cyclic LOESS using the normalizeBetweenArrays() function of the limma R-package (Ritchie et al, 2015). When analyzing pan-cancer cell lines, we only included genes in our buffering analysis with protein expression data in at least ten cell lines in which the chromosome arm encoding this gene was gained, ten cell lines where this chromosome arm was lost, and ten cell lines in which this chromosome arm had a neutral ploidy (see Schukken and Sheltzer, 2022). We decreased this threshold to three cell lines per chromosome arm CNA group for the dataset of lab-engineered cell lines.

### Aneuploidy score and whole-genome doubling

For cell lines, we obtained aneuploidy scores, whole-genome doubling, and ploidy calls from the DepMap portal (Cohen-Sharir et al, 2021; Data ref: DepMap, Broad Institute, 2023). We estimated the aneuploidy score of tumor samples similar to previous approaches (Bökenkamp et al, 2025; Taylor et al, 2018). Using absolute gene copy numbers obtained for CPTAC, we called gene copy number gain and loss by determining whether the copy number was above/below the rounded average ploidy of the tumor sample. Chromosome arm gain and loss were called by determining whether more than 75% of genes encoded on the chromosome arm had copy number gains/losses. We estimated the aneuploidy score (AS) by counting the number of chromosome arm copy number gain and loss events in each tumor sample.

### Cancer driver, drug screen, CRISPR-KO, and mutation datasets

The list of cancer driver genes was downloaded from the cancer gene list from OncoKB¯; only genes present in at least two sources were included (https://www.oncokb.org/cancer-genes; Chakravarty et al, 2017; Suehnholz et al, 2024). Drug-sensitivity data and compound metadata, like drug targets and mechanisms of action, were obtained from the PRISM Repurposing screens through the DepMap 23Q2 release (DepMap portal; Link to drug sensitivity dataset; Link to compound metadata; Corsello et al, 2020; Data ref: DepMap, Broad Institute, 2023). Gene dependency scores, representing gene dependency probability estimates generated from CRISPR Knock-Out screens using the Chronos algorithm, were obtained through the DepMap 23Q2 release (DepMap portal; Dempster et al, 2019, 2021; Data ref: DepMap, Broad Institute, 2023; Meyers et al, 2017; Pacini et al, 2021). The matrix of damaging mutations obtained through the DepMap 23Q2 release was used to derive the TP53 mutation status (DepMap portal; Data ref: DepMap, Broad Institute, 2023).

## Buffering ratio

We define the buffering ratio (BR) for quantifying the degree of dosage compensation as

$$BR = \text{sign}(c_o - c_b) * \left( \log_2 \frac{c_o}{c_b} - \log_2 \frac{p_o}{p_b} \right).$$

The variable $c_o$ represents the DNA copy number of a gene, and $p_o$ represents its protein abundance observed in a sample. $c_b$ and $p_b$ are estimated baselines for the gene copy number and relevant protein abundance across samples, respectively. Of note, $c_b > 0$ and $p_b > 0$. The "sign" function evaluates the difference $c_o - c_b$ and returns the sign (+1 or −1). The calculation of the baselines is described in the section "Protein dosage compensation analysis".

### Confidence score

The BR is influenced by variability and noise in the proteomics measurement and has, by definition, an increased sensitivity to changes in the protein abundance, specifically when the change in copy number is small. We defined a confidence score for the BR to estimate the uncertainty of the buffering estimate based on these factors:

$$BR_c = \left| \log_2 \frac{c_o}{c_b} \right| * \frac{1}{1 + \widehat{CV}(\mathbf{p_n})}.$$

$\mathbf{p_n}$ is a vector that contains the protein abundances of a gene in all samples within a dataset where the copy number of that gene was neutral, meaning there was neither a gain or a loss of copy number relative to the copy number baseline. $\widehat{CV}(\mathbf{p_n})$ is the sample coefficient of variation (CV) applied to $\mathbf{p_n}$.

## Buffering classes

### BR-based

We defined the scaling direction of protein abundance and copy number as positive if the observed protein abundance and copy number change in the same direction relative to the baseline (formally: $(c_o - c_b) * (p_o - p_b) > 0$), and negative otherwise. A negative scaling direction implies that the protein abundance increased while the copy number decreased relative to the baseline or vice versa. Using protein abundance log2 fold change (Log2FC)-based thresholds, BR-based thresholds, and the scaling direction we classified genes as *Scaling*, *Buffered*, or *Anti-Scaling* (Table 2).

### Log2FC-based

As a control, we classified genes as *Scaling*, *Buffered*, or *Anti-Scaling* using Log2FC-based thresholds only as previously published (Table 3; Schukken and Sheltzer, 2022).

## Protein dosage compensation analysis

We analyzed protein dosage compensation using both absolute gene copy numbers (CNs, ABSOLUTE algorithm) and chromosome arm copy number alterations (CNAs). We defined four different analysis variants (Table 4), each using different copy number data sources and classification methods.

For the analysis variant GeneCN, we calculated the copy number and protein abundance baselines $c_b$ and $p_b$ for each gene as the median absolute gene CN and median protein abundance across all samples in which the corresponding chromosome was disomic. We then calculated BR, the confidence score of the BR, and determined BR-based buffering classes using the protein abundances and

absolute gene CNs of each gene within each sample. For ChrArm, we applied a similar method, but used the rounded ploidy of the sample as the copy number baseline $c_b$ and added the chromosome arm CNA to $c_b$ to obtain the observed copy number $c_o$. ChrArm (Log2FC) uses the same baseline estimation as ChrArm but uses thresholds based on the Log2FC between observed and baseline protein abundance to determine buffering classes. For ChrArm (avg.), we used the method described by Schukken and Sheltzer: For each gene, we separated the samples into "gain", "neutral", and "loss" cohorts based on the chromosome arm CNA for the chromosome that encodes this specific gene. We then calculated the mean protein abundance Log2FC between gain and neutral, and between loss and neutral cohorts, and applied Log2FC-based thresholds to obtain the buffering classes (Schukken and Sheltzer, 2022).

For the tumor sample dataset CPTAC, we approached the GeneCN-based buffering analysis differently, as CPTAC offers proteomes of matched normal tissue samples for some tumors, and offers absolute gene copy number data generated using the AscatNGS algorithm (Raine et al, 2016). Reported protein abundance and absolute copy numbers in tumor samples were used as the observed protein abundance $p_o$ and observed copy number $c_o$. We calculated the protein abundance baseline $p_b$ as the median abundance of each protein within normal samples. As AscatNGS determines the copy numbers of tumor samples in comparison to a panel of normal samples, we assumed the normal tissues to be disomic and set the copy number baseline to $c_b = 2$ for all genes. Tumor samples with no corresponding normal samples or tumor samples with a purity below 40% were removed.

## Rank aggregation

We implemented the mean normalized rank (MNR) to aggregate and unify the estimates (e.g., sample BR and ROC AUCs) from different datasets or analysis variants (Upadhya and Ryan, 2022).

Given $n$ sets $S_1, \dots, S_n (n \in \mathbb{N})$, each comprising exactly $m$ features, so that the size (cardinality) of every set is identical: $|S_1| = \dots = |S_n| = m$. These features can either have present or missing values. For each set $S_i$, with $i \in \{1, \dots, n\}$, we calculated the rank of each feature with a value in $S_i$ and divided the rank by the number of features $m$. We then removed features that have more than $\lfloor n/2 \rfloor$ missing values across all $n$ sets. Finally, we obtained the MNR for each feature $f_j, j \in \{1, \dots, m\}$ by calculating the arithmetic mean of the ranks of the feature $f_j$ across all sets $S_1, \dots, S_n$.

## Buffering ratio variance analysis

To determine which genes were consistently buffered, we analyzed the variance of the BR for each gene. For each gene within the

**Table 2.  Thresholds used for classifying genes as *Scaling*, *Buffered*, and *Anti-Scaling* using the buffering ratio.**

| Buffering class | Scaling direction | BR threshold | Log2FC threshold |
|---|---|---|---|
| Scaling | Positive | BR ≤ 0.2 | – |
| Buffered | Positive | BR > 0.2 | – |
| Buffered | Negative | – | \|Log2FC\| ≤ 0.1 |
| Anti-Scaling | Negative | – | \|Log2FC\| > 0.1 |

**Table 3.  Log2FC thresholds used for classifying genes as *Scaling*, *Buffered*, and *Anti-Scaling*.**

| Buffering class | Chromosome arm gain | Chromosome arm loss |
|---|---|---|
| Scaling | Log2FC ≥ 0.25 | Log2FC ≤ −0.25 |
| Buffered | 0.25 > Log2FC ≥ −0.1 | −0.25 < Log2FC ≤ 0.1 |
| Anti-Scaling | −0.1 > Log2FC | 0.1 < Log2FC |

**Table 4.  Variants of the protein dosage compensation analysis.**

| Analysis variant | Granularity | Copy number data | Buffering classes |
|---|---|---|---|
| GeneCN | per-gene, per-sample | Absolute gene CNs | BR-based |
| ChrArm | per-gene, per-sample | Chromosome arm CNAs | BR-based |
| ChrArm (avg.) | per-gene | Chromosome arm CNAs | Log2FC-based |
| ChrArm (Log2FC) | per-gene, per-sample | Chromosome arm CNAs | Log2FC-based |

DepMap, ProCan, and CPTAC datasets, we calculated the mean and standard deviation of the BR (GeneCN analysis variant), the number of samples with an observable BR, and the fraction of samples where this gene was classified *Buffered*, and performed a one-sided one-sample Wilcoxon signed-rank test to determine whether the BR was significantly above the BR threshold for the *Buffered* class. Within each dataset, we only retained genes with at least 20 calculated BR observations, more than 33% of *Buffered* observations, a *Buffered*-fraction higher than the median *Buffered*-fraction of all genes, a mean BR above the *Buffered* threshold (BR > 0.2), a significant Benjamini–Hochberg adjusted $P$ value ($P_{adj}$ <0.05), and a BR standard deviation below 2. We calculated the mean normalized rank (MNR) of the BR standard deviation for each gene across all datasets and selected 50 genes with the lowest MNR as consistently buffered.

## Sample buffering ratio

We defined the sample buffering ratio (sample BR) to quantify the average degree of buffering exhibited in each sample (aneuploid cell line or tumor sample). We removed all diploid samples and removed BR values with low confidence scores ($BR_c$ <0.3) or low absolute copy number Log2FCs ($|\log_2 c_o/c_b|$ <0.3). Furthermore, we only kept genes that were common across DepMap, ProCan, and CPTAC to ensure comparability of the sample BRs between different proteomics approaches and datasets. Next, we removed all samples that had more than 50 missing gene copy number-derived BR values. With this dataset, the sample BRs were calculated as the mean gene copy number-derived BR (GeneCN analysis variant) across all genes within a sample. We calculated the z-scores by standardizing the sample BRs within each proteomics dataset.

## Analysis of sample properties and cancer types

Cancer type annotations from DepMap, ProCan, and CPTAC were mapped to OncoTree codes with the same hierarchy (level 2) using the *mskcc.oncotree* R-package (Kundra et al, 2021). We controlled for *low aneuploidy* by removing cell lines from the adherent cohort with aneuploidy scores above the maximum of the suspension cohort. To ensure *equal aneuploidy*, we resampled the adherent (target) cohort based on the aneuploidy score distribution in the suspension (reference) cohort. We divided the range of aneuploidy scores into $k = 6$ equal strata by applying the quantile function to the reference cohort aneuploidy scores. We then calculated the fraction of values that fell into each aneuploidy score stratum in the reference and target cohort. For each stratum, we divided the reference sample fraction by the target sample fraction and used this value as the probability for randomly sampling cell lines from the target cohort in the current stratum (without replacement). We then verified that the aneuploidy score difference between the two cohorts is not significantly different after resampling (using $k = 6$ strata) by applying a Wilcoxon rank-sum (Mann–Whitney $U$) test.

## Unifactorial analysis

We filtered for genes in samples with copy number or chromosome arm gain or loss, by comparing observed copy number against baseline copy number (for ChrArm, GeneCN) or by filtering for non-zero chromosome arm CNAs (for ChrArm (avg.)). By removing *Anti-Scaling* observations, we established a binary classification scheme, setting *Buffered* observations as the case (true) and *Scaling* as the control (false) class. To understand how well each factor individually distinguishes whether a gene is buffered or not, the receiver operating characteristic area under the curve (ROC AUC) was calculated for each factor, within each analysis variant, dataset, and copy number condition (gain, loss) using the *pROC* package (Robin et al, 2011).

WGD-controlled results for DepMap and ProCan were generated by separating WGD-positive and WGD-negative samples, determining buffering classes for all analysis variants, and calculating unifactorial ROC AUCs as described above. We then calculated mean normalized ranks of the ROC AUCs for each factor and copy number condition (gain, loss), combining all analysis variants across all datasets, all WGD-positive datasets, and all WGD-negative datasets.

### Bootstrapped analysis

To determine significant differences of factor ROC AUCs between copy number conditions and analysis variants, we applied filtering and binary classification schemes for each dataset, analysis variant, and copy number condition as described above, bootstrapped the resulting datasets ($n = 10,000$ random samples each, with replacement), and calculated ROC AUCs for each factor and bootstrapped dataset. Bootstrapping the ROC AUCs was parallelized using the parallel R-package. We calculated the rank correlation of factors between gain and loss conditions within BR-based analysis variants (ChrArm, GeneCN), and between analysis variants within gain and loss conditions using Kendall's rank correlation coefficient. We removed noisy factors with low predictiveness (median ROC AUC < 0.51) from the rank correlation and excluded factors that were highly correlated with included factors and contained redundant information (protein/mRNA abundance, activating/repressing transcription factors).

### BR-factor correlation analysis

Correlations between Buffering Ratio (GeneCN) and factor values for each factor, copy number event, and pan-cancer dataset (DepMap, ProCan, CPTAC) were obtained using Spearman's rank correlation coefficient; $P$ values were adjusted by the Benjamini–Yekutieli procedure.

## Multifactorial analysis

We trained machine learning models to use all available factors for predicting protein buffering. Before training, we imputed missing values of each factor with its median value and applied min-max normalization to improve convergence during training. For each dataset, analysis variant, and copy number condition, we then filtered for gain or loss and established a binary classification scheme as described in the unifactorial analysis. The observations in the resulting datasets were then shuffled and split into training and test subsets using an 80/20-split. Models were trained on each of the training datasets using the *caret* R-package with threefold cross-validation for hyperparameter tuning (Kuhn, 2008).

Model performance was evaluated by predicting *Buffered/ Scaling* labels in the associated test dataset of the model and calculating the ROC AUC on the *Buffered* class probabilities

returned by the model. We trained *xgbLinear* (extreme gradient boosting using linear classifiers), *rf* (random forest), and *pcaNNet* (artificial neural network with prior PCA feature transformation) model architectures available in *caret*; however, *xgbLinear* produced the highest ROC AUCs in pan-cancer datasets (DepMap, ProCan, CPTAC). Before training, we removed factors that were highly correlated, provided redundant information, and confounded the interpretability of the model (e.g., due to overfitting): "Homology Score", "Protein Abundance", "mRNA Abundance", "Transcription Factors (Repressor)", "Transcription Factors (Activator)", and "Triplosensitivity".

### Feature explanation

We removed observations from a model's test set where the model's prediction was incorrect, and calculated Shapley Additive exPlanation values (SHAP values, Lundberg and Lee, 2017) for a random subset of the remaining observations for each feature of the model (300 *Buffered*, 300 *Scaling* observations). SHAP values were calculated using a variant of the KernelSHAP algorithm that accounts for the statistical dependencies between features and allows categorically distributed features via the *ctree* method of the *shapr* R-package (Aas et al, 2021; Redelmeier et al, 2020). The baseline prediction $\phi_0$, representing the expected model output in the absence of any features, was defined as the fraction of *Buffered* observations in the model's training set. We set the number of sampled feature combinations to 300. Among the trained model architectures, *shapr* only supported calculating SHAP values for *xgbLinear*. To identify which feature values contribute to a higher chance of predicting an observation as *Buffered* or *Scaling*, we calculated the Spearman's correlation coefficient between SHAP values and the associated values of the feature across all observations.

### Out-of-sample evaluation

To evaluate the performance of a model across datasets, analysis variants, and copy number conditions, we evaluated the performance of each model on the test set of every other model using the ROC AUC on the *Buffered* class prediction probabilities as described above.

## Differential expression analysis

Samples were considered low-buffering if their sample BR was below the 20% percentile of sample BRs, and high-buffering if their sample BR was above the 80% percentile. For each gene, we calculated mean protein abundance $\log_2$ fold changes ($\log_2 p_{high} - \log_2 p_{low}$), performed unpaired two-tailed Welch's unequal variances $t$ tests, and adjusted the $P$ values using the Benjamini–Hochberg procedure to determine significant protein abundance differences between high- and low-buffering samples. Genes were considered significantly deregulated if $|Log_2 FC| > 0.5$ and if the adjusted $P$ value was below 0.05. We created an additional adherent control dataset by removing all cell lines with non-adherent growth patterns from the ProCan dataset. We then performed splitting and differential expression analysis on this dataset as described above. Genes were called to be commonly deregulated in high-/low-buffering cell lines if they were significantly up-/downregulated in DepMap, ProCan, and the adherent control.

## Enrichment analysis

Over-representation analysis was performed using the *gprofiler2* R-package. Plots show the five most enriched terms within each selected term dataset (e.g., KEGG pathway, Gene Ontology, CORUM; Kolberg et al, 2023). We performed gene set enrichment analysis (GSEA) for DepMap, ProCan, and CPTAC using the human hallmark gene sets of the molecular signatures database (MSigDB; Liberzon et al, 2015; Subramanian et al, 2005). Normalized enrichment scores have been generated with the *fGSEA* R-package using the log2 fold-change ranked lists of genes from the differential expression analysis (Korotkevich et al, 2016). We generated aneuploid controls for DepMap, ProCan, and CPTAC by performing differential expression analysis on cell lines separated by aneuploidy score (>80%, <20%) and applying GSEA on the Log2FC ranks. Network analysis of significantly deregulated genes has been performed with STRING, using the *STRINGdb* R package (Szklarczyk et al, 2023). We retained edges in the network with a 90% confidence or higher, removed disconnected nodes, and performed clustering using the MCL algorithm with an inflation parameter of 2.

## Proteotoxic stress analysis

To estimate the degree of proteotoxic stress per tumor sample, we performed single-sample GSEA (ssGSEA) using the *GSVA* R-package, the MSigDB human hallmark gene sets, and protein abundance measurements from the CPTAC dataset (Hänzelmann et al, 2013). We then calculated Spearman's rank correlation coefficient between enrichment scores for the *UNFOLDED_PROTEIN_RESPONSE* gene set and the sample BRs of the tumor samples.

## Proliferation analysis

To estimate the proliferation, we obtained the day 4-day 1 growth ratios from DMSO-treated cell lines in the GDSC drug screen dataset, removed ratios with less than three replicates, and used the ratio with the most replicates for each cell line (Cell Model Passports portal; Iorio et al, 2016). Spearman's rank correlation coefficient was calculated between day 4-to-day 1 growth ratios of cell lines, sample BRs calculated for the ProCan and DepMap datasets, and sample BR mean normalized ranks (MNR).

## Gene essentiality analysis

Differential dependency of buffered cell lines on gene functions was determined using dependency scores generated from CRISPR-KO screens (see Data Acquisition and Preprocessing). We separated cell lines into high- and low-buffering cohorts using 80% and 20% percentile cutoffs on the sample BR mean normalized ranks for cell lines, calculated the mean dependency score difference per gene ($d_{high} - d_{low}$), performed two-tailed Wilcoxon rank-sum (Mann–Whitney $U$) tests, and adjusted the $P$ values using the Benjamini–Hochberg procedure. High-/low-buffering cell lines were considered differentially dependent on a gene if the adjusted $P$ value of the gene was below 0.05 and if $|d_{high} - d_{low}| > 0.1$. Genes were called exclusively essential in high-buffering cell lines if the mean dependency score of the gene was significantly higher in high-buffering cell lines, and its mean dependency score was below 0.5 in low-buffering cell lines, and above 0.5 in high-buffering cell

lines. Vice versa, genes were considered exclusively essential in low-buffering cell lines for significantly higher dependency scores in low-buffering cell lines, and a mean dependency score above 0.5 in low-buffering cell lines, and below 0.5 in high-buffering cell lines. Similar to the differential expression analysis, we generated an adherent control dataset by reusing the sample BR MNR dataset, removing cell lines with non-adherent growth patterns, splitting by sample BR MNR, and performing the differential dependency analysis as described above.

We highlight that the differential dependency analysis is unrelated to the definition of the Mean Gene Dependency factor (Table 1) used for factor analysis in Figs. 3 and 4.

### Drug-sensitivity analysis

We calculated the sensitivity of high-buffering cells on individual drugs, drug targets, and drug mechanisms in two ways. First, we calculated the Spearman correlation coefficient of cell line sample BR mean normalized ranks and median drug effect scores from the PRISM Repurposing screens (see section "Data acquisition and preprocessing") per drug, drug target, and drug mechanism of action. Second, we separated cell lines based on their sample BR MNR ($>80\%$, $<20\%$), calculated the mean drug effect score difference between high- and low-buffering cell lines ($e_{high} - e_{low}$), and performed unpaired two-tailed Welch's unequal variances $t$ tests for each drug, drug target, and drug mechanism. Both correlation and differential drug effect $P$ values were adjusted using the Benjamini–Hochberg procedure. Mean drug effect score differences were considered significant if the adjusted $P$ value was below 0.05 and if $|e_{high} - e_{low}| > 0.2$. We selected drugs, mechanisms, and targets as robust hits for visualization if they were significant with matching effects in both correlation-based and differential drug effect analyses.

We controlled for aneuploidy by performing differential drug effect analysis on cell lines split by aneuploidy score instead of sample BR MNR percentiles ($>80\%$, $<20\%$). We called drug sensitivities unique for high/low-buffering cell lines, if both correlation and differential drug effect differences were significant and coincided, and if the drug effect differences between high and low-aneuploid cell lines were non-significant or significant in the opposite direction.

### Statistical analysis

In all boxplots, the box represents the interquartile range with the central line indicating the median. The whiskers extend to the most extreme data points within 1.5× the interquartile range. Two-tailed Wilcoxon rank-sum (Mann–Whitney $U$) tests were performed to determine significant differences in BR-derived quantities (e.g., BR, sample BR) when comparing two conditions. Correlations with BR-derived quantities were calculated using Spearman's rank correlation coefficient. Unless stated otherwise, tests with $P < 0.05$ were considered significant. No blinded experiments were performed.

## Data availability

The datasets and computer code produced in this study are available in the following databases: Mass spectrometry proteomics data of the chromosome-engineered RPE-1 cell lines (P0211): PRIDE PXD060017. Computational study code: GitHub (https://github.com/sheltzer-lab/DosageCompensationFactors).

The source data of this paper are collected in the following database record: biostudies:S-SCDT-10_1038-S44320-026-00187-9.

## Peer review information

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

## Acknowledgements

The proteomics mass spectra of the engineered RPE-1 cell lines (P0211) were measured by the Center for MS Analytics at RPTU Kaiserslautern (https://bio.rptu.de/center-ms-analytics). This project was supported by the grant project from Deutsche Forschungsgemeinschaft (DFG) STO 918/7-2, and by BioComp 4.0 and MLKL (funded by the Research Initiative of Rhineland-Palatinate) to Zuzana Storchová. Relevant research in the Sheltzer Lab was supported by NIH grants R01CA237652 and R01CA276666, and a sponsored research agreement from Ono Pharmaceutical.

## Author contributions

**Erik Marcel Heller**: Conceptualization; Data curation; Software; Formal analysis; Validation; Investigation; Visualization; Methodology; Writing—original draft; Writing—review and editing. **Karen Barthel**: Validation; Investigation; Methodology; Writing—review and editing. **Markus Räschle**: Validation; Investigation; Methodology; Writing—review and editing. **Klaske M Schukken**: Validation; Methodology; Writing—review and editing. **Jason M Sheltzer**: Conceptualization; Supervision; Funding acquisition; Methodology; Writing—review and editing. **Zuzana Storchova**: Conceptualization; Data curation; Supervision; Funding acquisition; Writing—original draft; Project administration; Writing—review and editing.

Source data underlying figure panels in this paper may have individual authorship assigned. Where available, figure panel/source data authorship is listed in the following database record: biostudies:S-SCDT-10_1038-S44320-026-00187-9.

## Funding

## Disclosure and competing interests statement

JMS is an inventor on a patent related to chromosome engineering and the construction of aneuploid genomes. JMS is a co-founder of and equity holder in Meliora Therapeutics and KaryoVerse Therapeutics. J.M.S. has received consulting fees from Merck, Pfizer, Ono Pharmaceuticals, Highside Capital Management, and Meliora Therapeutics and is a member of the advisory boards of BioIO, Permanence Bio, KaryoVerse Therapeutics, and the Chemical Probes Portal.

# Expanded View Figures

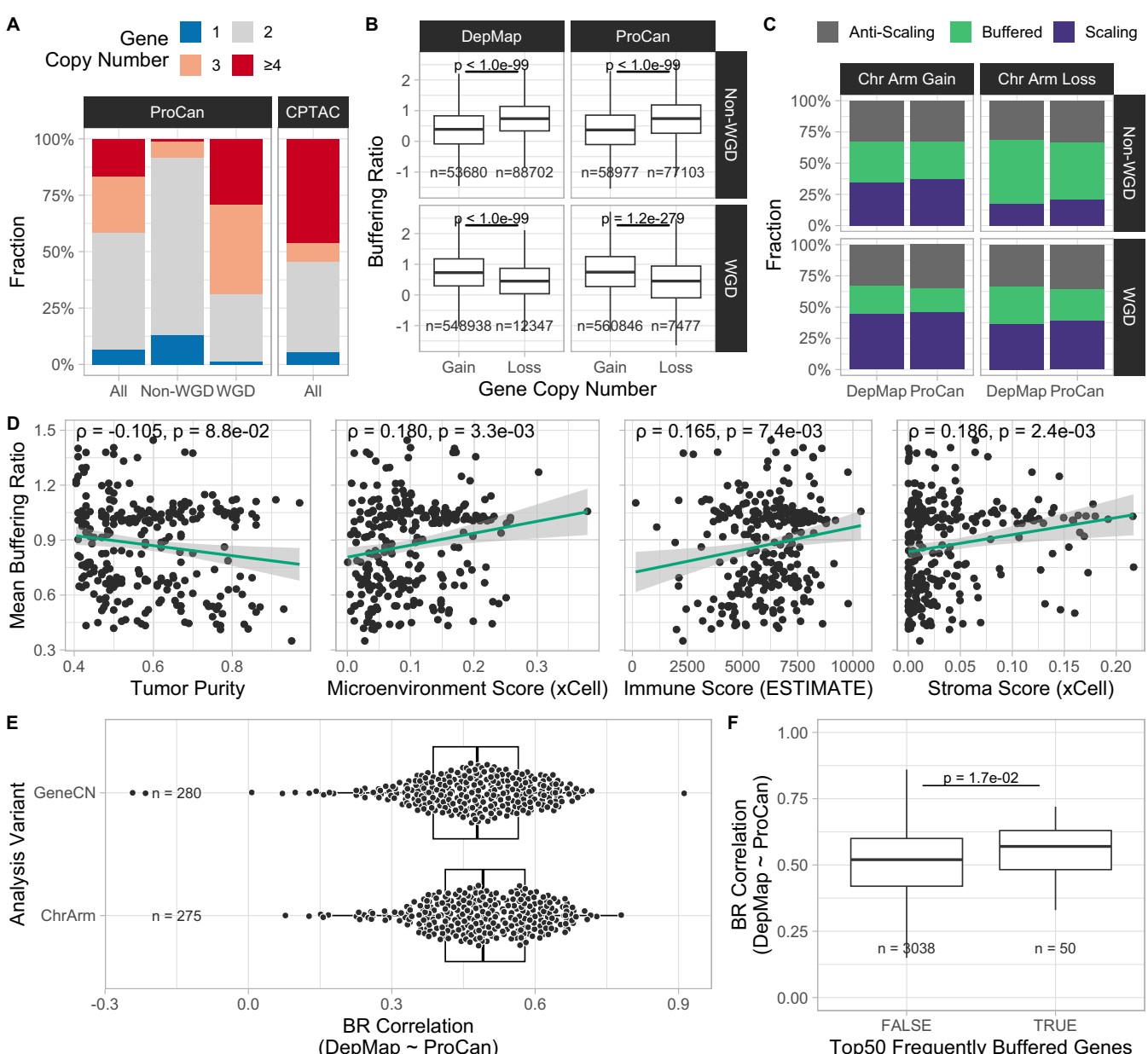

**Figure EV1. Controlling for whole-genome doubling status and tumor purity confirms observed protein buffering patterns.**

(A) Distribution of gene copy numbers among cell lines (ProCan) and tumor samples (CPTAC) separated by their whole-genome doubling (WGD) status. (B) Difference in buffering ratio distribution of all proteins between gene copy number gain and loss separated by the WGD status. (C) Categorical distribution of buffering classes across pan-cancer cell line datasets upon gene copy number gain and loss separated by WGD status. (D) Correlation between mean buffering ratio per tumor sample (CPTAC) and purity scores obtained from metadata provided by CPTAC (Spearman's ρ). (E) Correlation of buffering ratios per cell line between DepMap and ProCan (Spearman's ρ), separated by buffering ratios created using either gene (GeneCN) or chromosome arm copy numbers (ChrArm). (F) Gene-wise buffering ratio correlation between DepMap and ProCan (GeneCN, Spearman's ρ), separated by their presence in the list of frequently buffered genes with low BR variance. (B, E, F) Boxes represent the interquartile range (IQR) with the central line indicating the median. The whiskers extend to the data points within 1.5×IQR. *P* values were determined using a two-tailed Wilcoxon rank-sum (Mann–Whitney *U*) test.

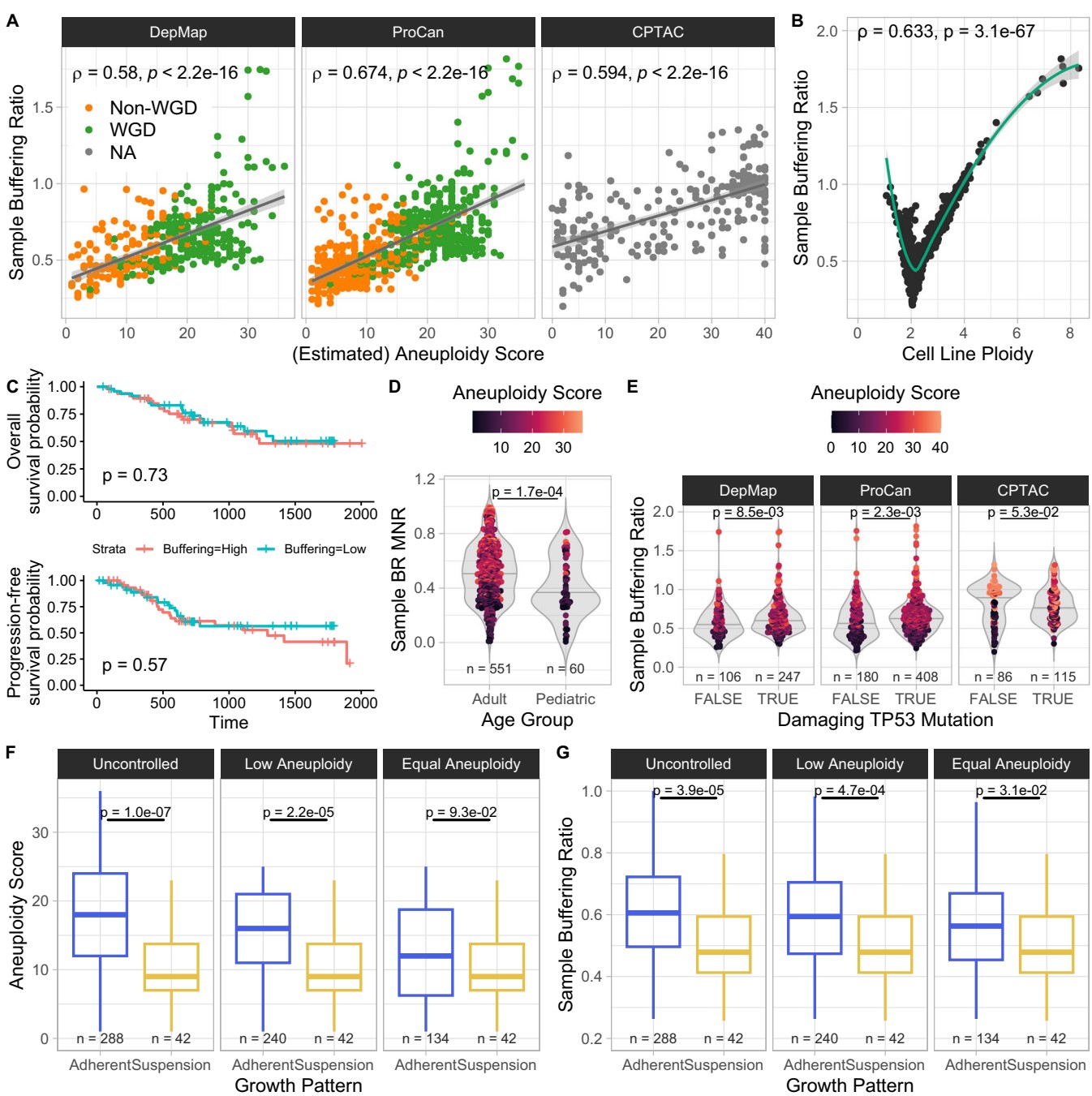

**Figure EV2.  Average buffering per sample shows relationship with cellular and clinical features of cancer.**

(A) Scatter plot showing a positive correlation between sample buffering ratio and aneuploidy score in all pan-cancer datasets (Spearman's $\rho$). (B) Scatter plot showing a LOESS-smoothed non-linear relationship between sample buffering ratio and cell line ploidy (ProCan; Spearman's $\rho$). (C) Kaplan–Meier plots showing a decreased progression-free and overall survival probability for tumor samples with high buffering (>80% quantile of sample buffering ratio) compared to low-buffering (<20% quantile, CPTAC). (D) Difference in sample buffering ratio mean normalized ranks between age categories shows decreased sample-wide buffering in pediatric cancers (Wilcoxon rank-sum test). (E) Difference in sample buffering ratio between samples with and without damaging TP53 mutations (Wilcoxon rank-sum test). (F) Difference in aneuploidy between cell lines grown in suspension and adherent cell culture types when applying aneuploidy control methods (DepMap). Controlled for aneuploidy by removing cell lines from the adherent cohort with aneuploidy scores above the maximum of the suspension cohort (middle), and by using stratified sampling to ensure an equal distribution of aneuploidy scores (right). (G) Difference in sample buffering ratio between cell lines grown in suspension and adherent cell culture types, controlled for aneuploidy (DepMap). (F, G) Boxes represent the interquartile range (IQR) with the central line indicating the median. The whiskers extend to the data points within 1.5×IQR. (D–G) P values were determined using a two-tailed Wilcoxon rank-sum (Mann–Whitney U) test.

                                                      

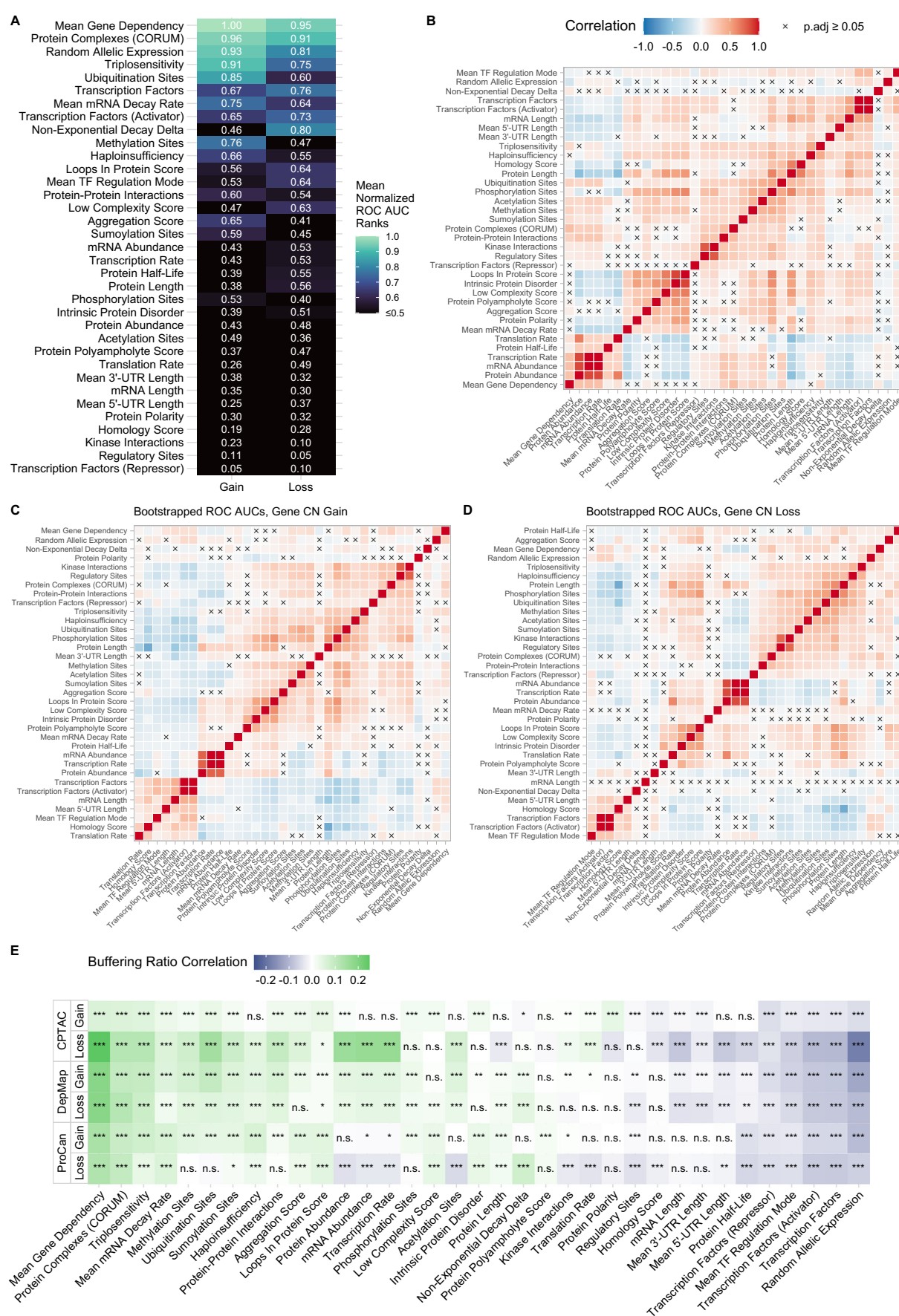

◀ **Figure EV3.   Correlation patterns of potential dosage compensation factors.**

(**A**) Mean normalized ranks (MNRs) of ROC AUCs of each factor used for classifying whether a gene is *Buffered* on protein level or *Scaling*, while excluding the *Anti-Scaling* class. MNRs were calculated across all analysis variants (GeneCN, ChrArm, ChrArmAvg) and pan-cancer datasets (DepMap, ProCan, CPTAC), grouped by gene or chromosome arm copy number gain and loss. WGD-controlled subsets were excluded. (**B–D**) Clustered heatmaps showing the correlation (Spearman's $\rho$) between factors. Correlations with insignificant Benjamini–Hochberg adjusted *P* values were crossed out ($p_{adj} \geq 0.05$). Correlations were calculated using factor values (**B**), bootstrapped ROC AUCs for predicting protein buffering upon gene copy number gain (**C**), and bootstrapped ROC AUCs upon gene copy number loss (**D**). (**E**) Spearman correlation between factors values and buffering ratios upon gene copy number loss and gain across all pan-cancer datasets (DepMap, ProCan, CPTAC).

                            

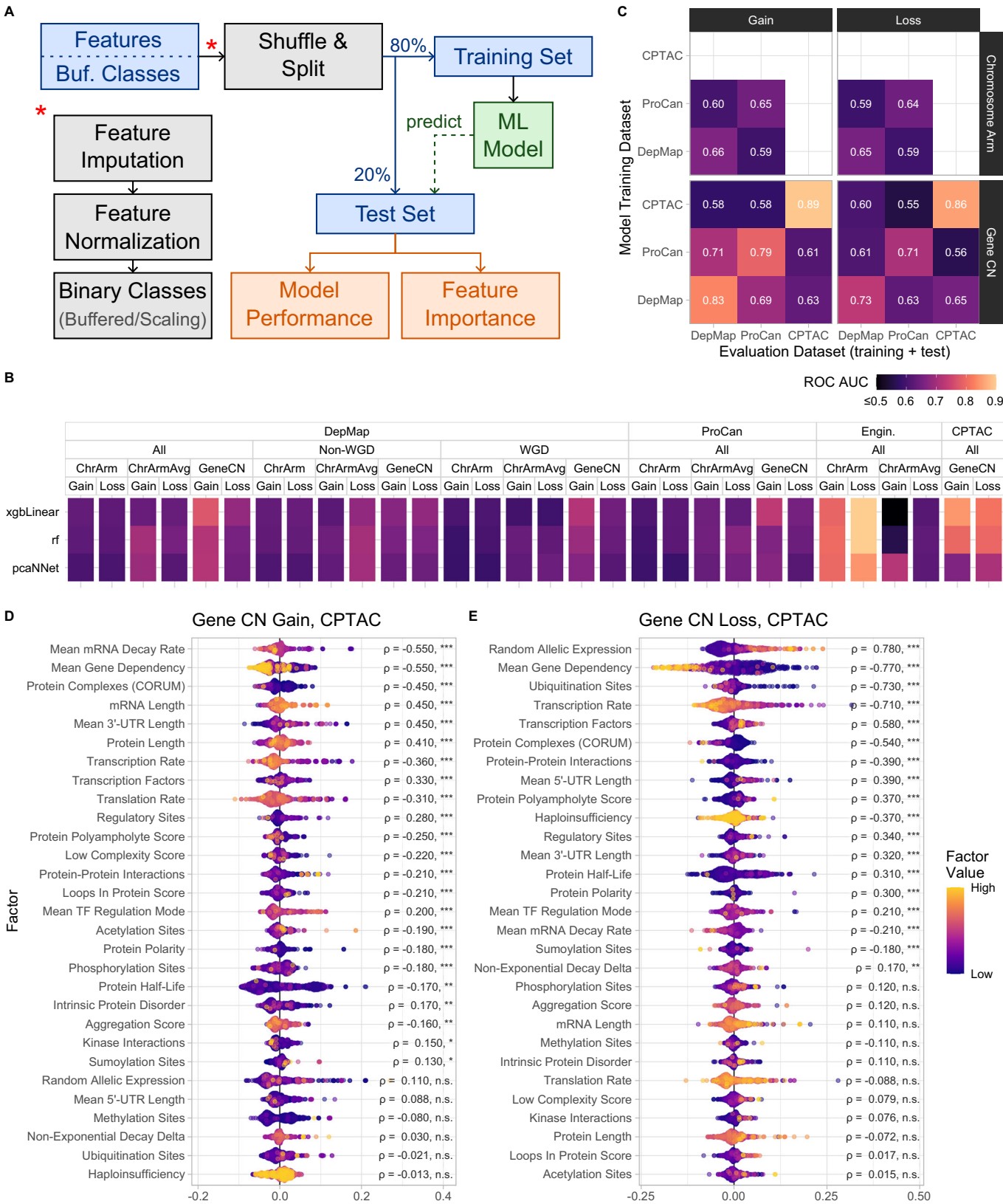

**Figure EV4.  Multifactorial models improve prediction performance of protein buffering.**

(A) Illustration of training and evaluation procedures of multifactorial models. (B) Model performance (ROC AUC) of trained models, separated by model architecture (xgbLinear, rf, pcaNNet), dataset (DepMap, ProCan, CPTAC, Engineered), subset (All, WGD, Non-WGD), analysis variant (GeneCN, ChrArm, ChrArmAvg), and copy number event (Gain, Loss). (C) Out-of-sample performance of multifactorial models trained on buffering classes derived from gene and chromosome arm copy number gain and loss data from different pan-cancer datasets (DepMap, ProCan, CPTAC). Model performance was evaluated using the ROC AUC of model predictions on merged training and test datasets. (D, E) SHAP values generated by evaluating the trained CPTAC gene copy number models on a random subset of the respective test set ($n = 300$). Negative SHAP values depict a higher contribution of a model's factor towards predicting an observation as *Buffered* relative to the baseline prediction. Each dot represents a gene in a sample. Color represents the min-max-scaled value of the factor used for prediction. Spearman's $\rho$ indicates the trend between SHAP value and feature value. SHAP values were mean-centered within each factor.

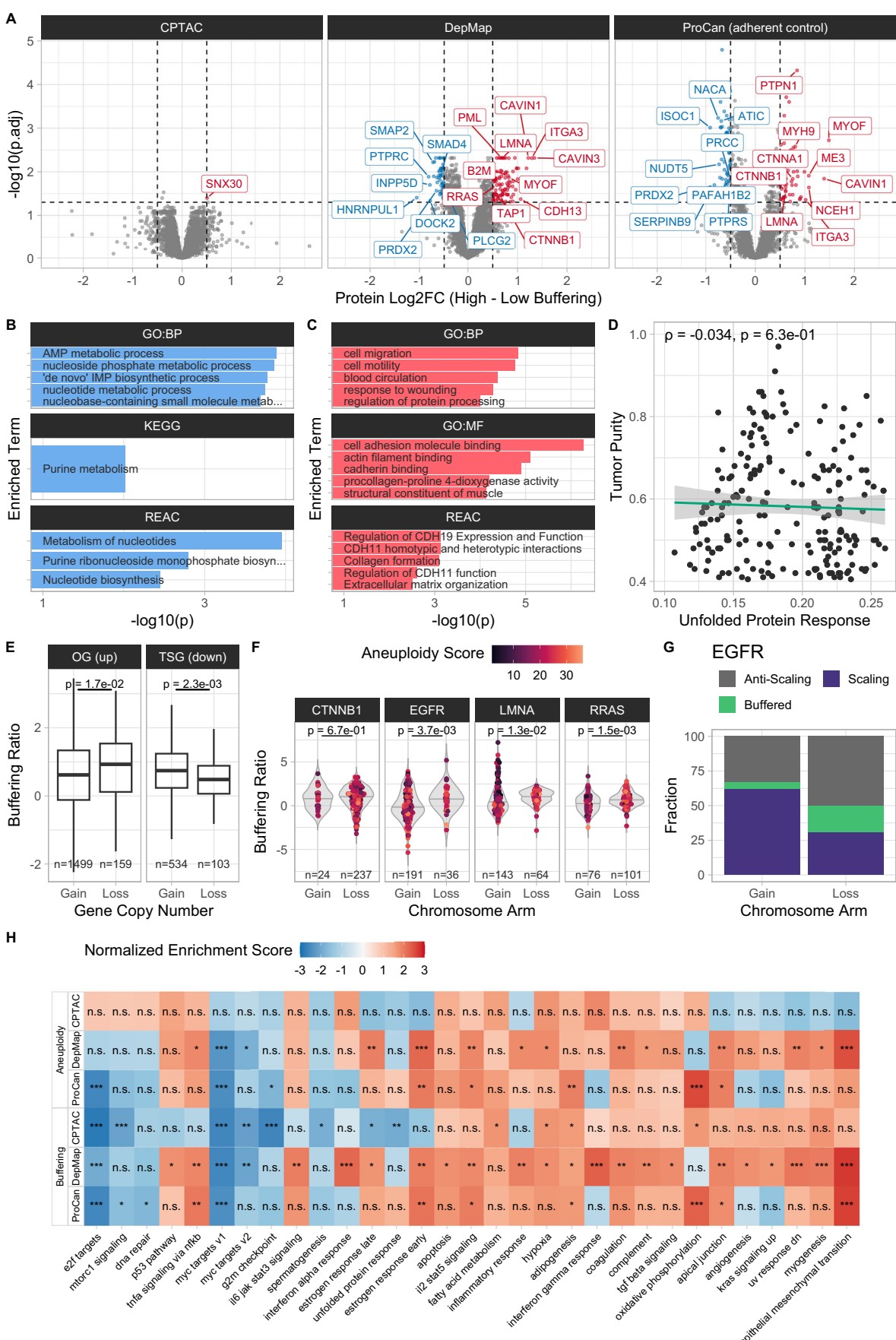

◀  **Figure EV5.  Enrichment patterns suggest oncogenic effects of protein buffering.**

(A) Volcano plots showing log2 fold change between high (>80% quantile of sample buffering ratios) and low (<20% quantile) buffering cell lines and tumor samples in DepMap ($n = 71$ per group), CPTAC ($n = 52$ per group), and adherent control datasets ($n = 82$ high buffering, $n = 51$ low buffering; Student's *t* test, Benjamini–Hochberg adjusted *P* values, significance thresholds: |Log2FC | > 0.5, $P_{adj} < 0.05$). (B, C) Overrepresentation analysis (ORA) of genes that are significantly down- (B) and upregulated (C) in highly buffering cell lines with adherent growth patterns (ProCan, adherent control; hypergeometric test using g:Profiler). (D) Correlation between tumor purity and single-sample enrichment scores for the *Unfolded Protein Response* HALLMARK gene set using CPTAC tumor sample proteome data (Spearman's ρ). (E) Buffering ratio distribution of oncogenes (OG) significantly upregulated in high buffering cell lines and of downregulated tumor suppressor genes (TSG) upon gene copy number gain and loss (ProCan, all cell lines). Boxes represent the interquartile range (IQR) with the central line indicating the median. The whiskers extend to the data points within 1.5×IQR. (F) Buffering ratio distribution of selected oncogenes upon gene copy number gain and loss (ProCan). (G) Categorical distribution of buffering classes for EGFR upon gain and loss of chromosome arm 7p (ProCan). (H) Normalized enrichment scores of MSigDB HALLMARK gene sets for pan-cancer datasets (DepMap, ProCan, CPTAC). Enrichment scores have been generated by comparing high against low-buffering samples and comparing high against low-aneuploid samples (>80% and <20% quantiles of sample buffering ratio and aneuploidy score estimates). Significance levels of Benjamini–Hochberg adjusted *P* values generated using fGSEA: *$P_{adj} < 0.01$, **$P_{adj} < 0.001$, ***$P_{adj} < 0.0001$. Removed gene sets with no significant enrichment scores. (E, F) *P* values were determined using a two-tailed Wilcoxon rank-sum (Mann–Whitney *U*) test.

                                               

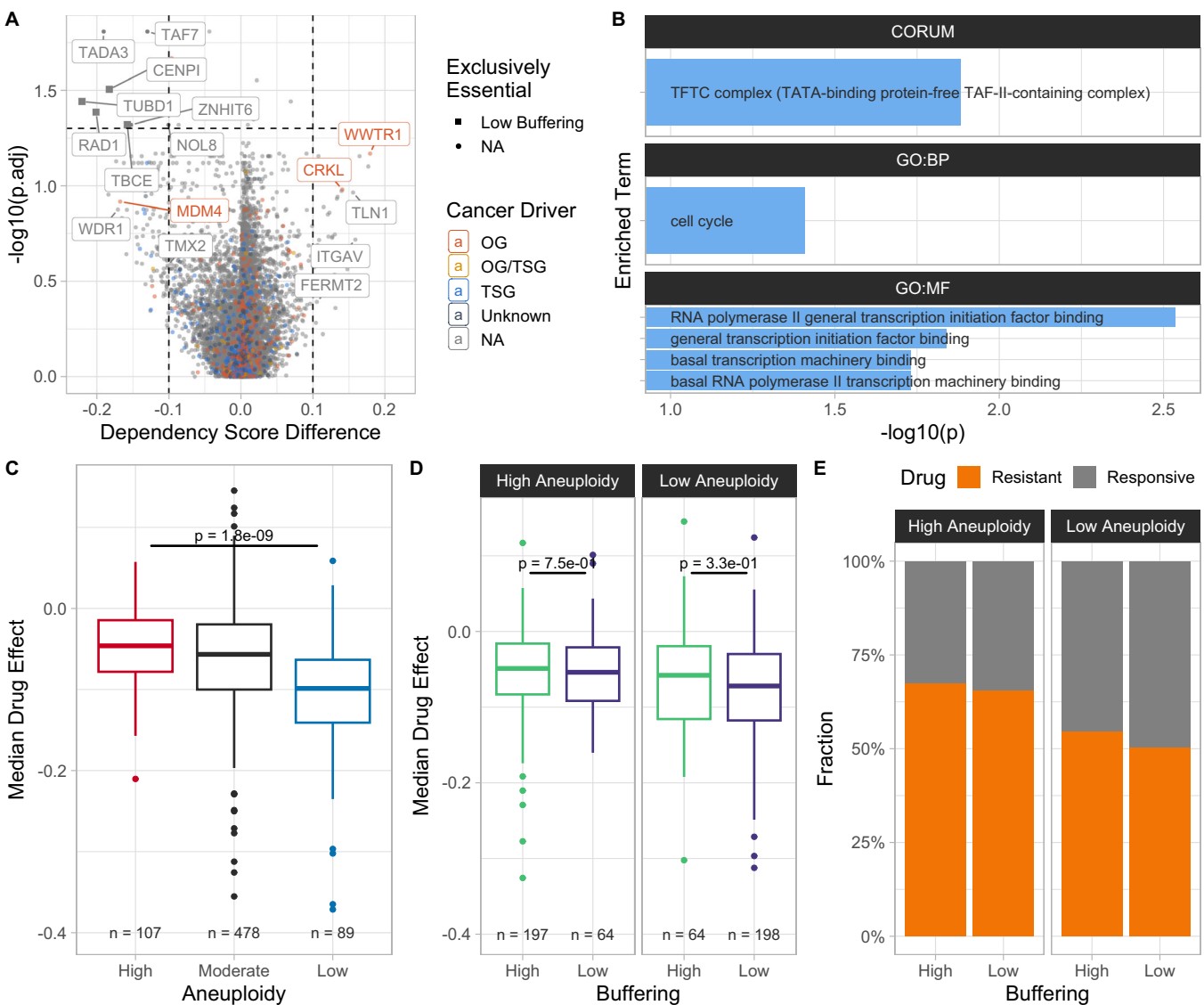

**Figure EV6. Adherent and aneuploid controls for CRISPR knock-out and drug sensitivity analyses.**

(**A**) Volcano plot showing the difference of the CRISPR-KO dependency score between high (>80% quantile of sample BR MNRs, $n = 90$) and low (<20% quantile, $n = 67$) buffering cell lines plotted against Benjamini–Hochberg adjusted Wilcoxon rank-sum test *P* values. Cell lines without adherent growth patterns were removed prior to the analysis. Genes were highlighted if they were significant in the adherent control or among the top 10 significant genes in Fig. 6A. (**B**) Over-representation analysis (ORA) of genes with significantly reduced CRISPR-KO dependency scores in highly buffering cell lines with adherent growth patterns (hypergeometric test using g:Profiler). (**C**) Median drug effect on cell viability across drugs from the PRISM Repurposing dataset for cell lines high (>80% quantile of aneuploidy score), moderate (20%-80%), and low aneuploidy (<20%). (**D**) Median drug effect for high (>50% sample BR MNR) and low-buffering cell lines, separated by median aneuploidy score. (**E**) Categorical distribution of drug-resistant (>50% median drug effect) and responsive cells between high (>50% sample BR MNR) and low-buffering cell lines, separated by median aneuploidy score. (**C**, **D**) Boxes represent the interquartile range (IQR) with the central line indicating the median. The whiskers extend to the data points within 1.5×IQR. *P* values were determined using a two-tailed Wilcoxon rank-sum (Mann–Whitney *U*) test.

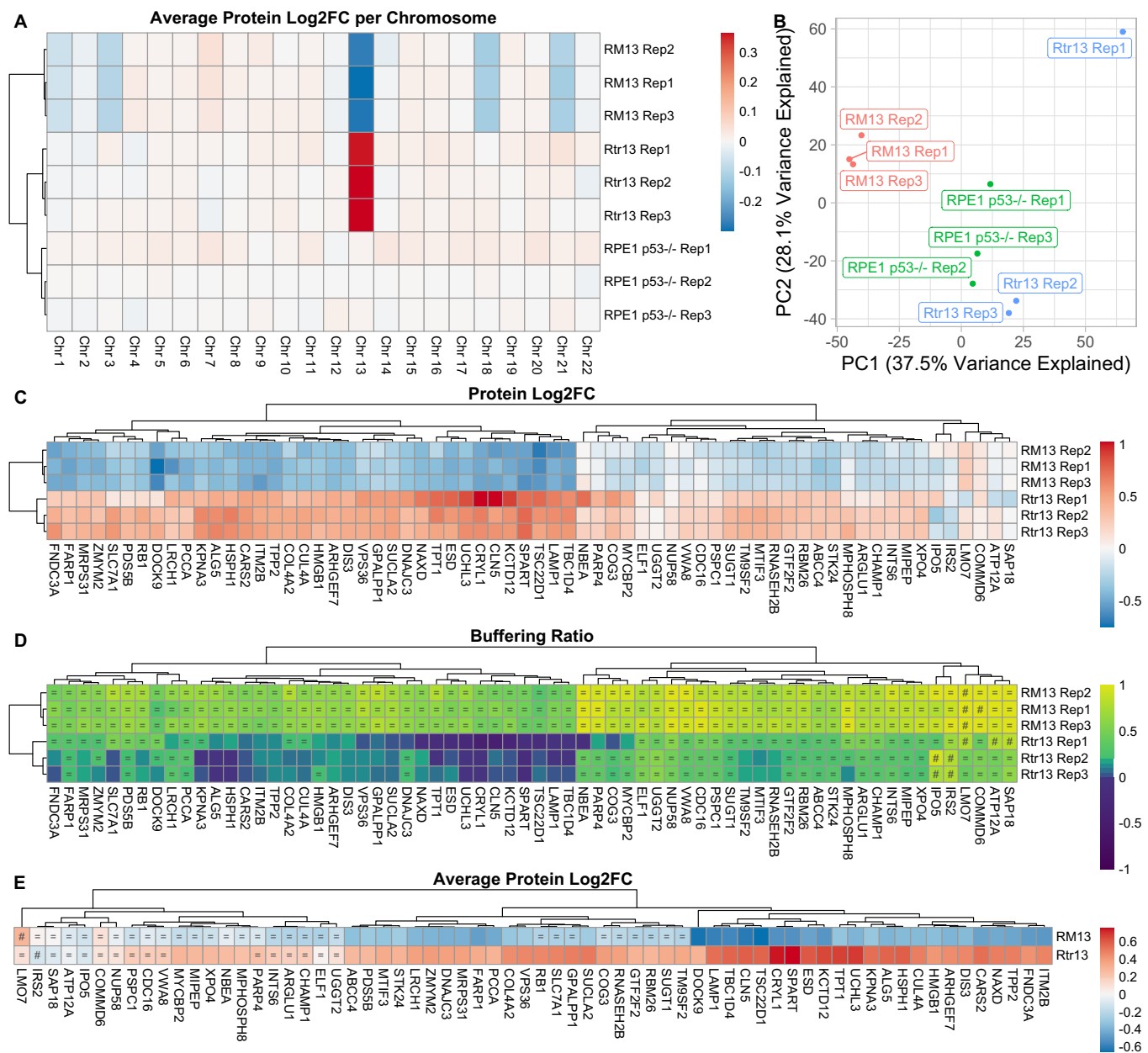

**Figure EV7. Proteome and protein buffering analysis of RPE-1 cell lines with engineered aneuploidies on chromosome 13 (PO211).**

(A) Mean log2 fold change of protein abundance per chromosome between cell line replicates and disomic baseline (median protein abundance of unaltered RPE-1 replicates per protein). (B) Principal components analysis (PCA) of PO211 replicates after normalization (first two principal components are displayed). (C) Protein abundance log2 fold change of proteins encoded on chromosome 13 between aneuploid replicates and disomic baseline. (D) Chromosome arm copy number-derived buffering ratios (BR) of proteins encoded on chromosome 13 in aneuploid replicates. Proteins are classified as *Buffered* (=) and *Anti-Scaling* (#) using BR-based thresholds (see "Methods"). (E) Mean protein abundance log2 fold-changes of proteins encoded on chromosome 13 between aneuploid and non-aneuploid cell lines. Proteins were classified as *Buffered* (=) and *Anti-Scaling* (#) using Log2FC-based thresholds (see "Methods").

