## [Peer Review File · Molecular Systems Biology]

Protein Buffering of Aneuploidy is Driven by Coordinated Factors Identified through Machine Learning

Erik Marcel Heller, Karen Barthel, Markus Räschle, Klaske Schukken, Jason Sheltzer, and Zuzana Storchova

Corresponding author(s): Zuzana Storchova (zuzana.storchova@rptu.de) , Jason Sheltzer (Jason.Sheltzer@yale.edu)

Review Timeline:

Submission Date:	9th May 25
Editorial Decision:	18th Jun 25
Revision Received:	28th Aug 25
Editorial Decision:	24th Nov 25
Revision Received:	19th Dec 25
Accepted:	3rd Jan 26

Editor: Poonam Bheda

Transaction Report:

18th Jun 2025

Manuscript Number: MSB-2025-13094

Title: Explainable Machine Learning Identifies Dosage Compensation Factors in Aneuploid Human Cancer Cells

Dear Prof Storchova,

Thank you for the submission of your manuscript to Molecular Systems Biology. We have now received feedback from the three reviewers who agreed to evaluate your manuscript. As you will see from the reports below, the referees acknowledge the interest of the study and are overall supporting publication of your work pending appropriate revisions.

I think that the recommendations of the reviewers are rather clear and I therefore do not see the need to repeat the comments listed below. All issues raised would need to be satisfactorily addressed. Please let me know in case you would like to discuss in further detail any of the any of the reviewer comments or your proposed revisions, I would be happy to schedule a call.

We require:

1) A .docx formatted version of the manuscript text (including legends for main figures, EV figures and tables). Please make sure that the changes are highlighted to be clearly visible. Alternatively you may choose to submit your manuscript as a LaTeX file.

4) A .docx formatted letter INCLUDING the reviewers' reports and your detailed point-by-point responses to their comments. As part of the EMBO Press transparent editorial process, the point-by-point response is part of the Peer Review File (PRF), which will be published alongside your paper.

5) A complete author checklist, which you can download from our author guidelines (<https://www.embopress.org/page/journal/17574684/authorguide#submissionofrevisions>). Please insert information in the checklist that is also reflected in the manuscript. The completed author checklist will also be part of the PRF.

6) Please note that all corresponding authors are required to supply an ORCID ID for their name upon submission of a revised manuscript.

7) It is mandatory to include a 'Data Availability' section after the Materials and Methods. Before submitting your revision, primary datasets produced in this study need to be deposited in an appropriate public database, and the accession numbers and database listed under 'Data Availability'. Please remember to provide a reviewer password if the datasets are not yet public (see <https://www.embopress.org/page/journal/17574684/authorguide#dataavailability>).

In case you have no data that requires deposition in a public database, please state so in this section as follows: "This study includes no data deposited in external repositories". Note that the Data Availability Section is restricted to new primary data that are part of this study.

8) All Materials and Methods need to be described in the main text using our 'Structured Methods' format, which is required for all research articles. According to this format, the Methods section includes a Reagents and Tools Table (listing key reagents, experimental models, software and relevant equipment and including their sources and relevant identifiers) followed by a Methods and Protocols section describing the methods using a step-by-step protocol format. The aim is to facilitate adoption of the methodologies across labs. Please upload the Reagents and Tools table as a separate document when submitting your revised manuscript. More information on how to adhere to this format as well as a downloadable template (.docx) for the Reagents and Tools Table can be found in our author guidelines:

<https://www.embopress.org/page/journal/17444292/authorguide#structuredmethods>

9) For data quantification: please specify the name of the statistical test used to generate error bars and p-values, the number (n) of independent experiments (specify technical or biological replicates) underlying each data point and the test used to calculate p-values in each figure legend. The figure legends should contain a basic description of n, p-values and the test applied. Graphs must include a description of the bars and the error bars (s.d., s.e.m.). Please provide exact p-values (in either the figure or figure legend).

10) Our journal encourages inclusion of *data citations in the reference list* to directly cite datasets that were re-used and obtained from public databases. Data citations in the article text are distinct from normal bibliographical citations and should directly link to the database records from which the data can be accessed. In the main text, data citations are formatted as follows: "Data ref: Smith et al, 2001" or "Data ref: NCBI Sequence Read Archive PRJNA342805, 2017". In the Reference list, data citations must be labeled with "[DATASET]". A data reference must provide the database name, accession number/identifiers and a resolvable link to the landing page from which the data can be accessed at the end of the reference. Further instructions are available at .

11) We replaced Supplementary Information with Expanded View (EV) Figures and Tables that are collapsible/expandable online. EV Figures should be cited as 'Figure EV1, Figure EV2' etc... in the text and their respective legends should be included in the main text after the legends of regular figures.

- Additional Tables/Datasets should be labeled and referred to as Table EV1, Dataset EV1, etc. Legends should be provided in a separate tab in case of .xls files. Alternatively, the legend can be supplied as a separate text file (README) and zipped together with the Table/Dataset file.

<https://www.embopress.org/page/journal/17574684/authorguide#expandedview>

12) Author contributions: CRediT has replaced the traditional author contributions section because it offers a systematic machine-readable author contributions format that allows for more effective research assessment. Please remove the Authors Contributions from the manuscript and use the free text boxes beneath each contributing author's name in our system to add specific details on the author's contribution. More information is available in our guide to authors.

13) Disclosure statement and competing interests: We updated our journal's competing interests policy in January 2022 and request authors to consider both actual and perceived competing interests. Please review the policy <https://www.embopress.org/competing-interests> and update your competing interests if necessary.

14) Every published paper now includes a 'Synopsis' to further enhance discoverability. Synopses are displayed on the journal webpage and are freely accessible to all readers. They include a short stand first (maximum of 300 characters, including space) as well as 2-5 one-sentences bullet points that summarizes the paper. Please write the bullet points to summarize the key NEW findings. They should be designed to be complementary to the abstract - i.e. not repeat the same text. We encourage inclusion of key acronyms and quantitative information (maximum of 30 words / bullet point). Please use the passive voice. Please attach these in a separate file or send them by email, we will incorporate them accordingly.

Please note that these would be the final versions and changes during proofing are usually not allowed.

15) As part of the EMBO Publications transparent editorial process initiative (see our policy here:

https://www.embopress.org/transparent-process#Review_Process), Molecular Systems Biology will publish online a Peer Review File (PRF) to accompany accepted manuscripts.

In the event of acceptance, this file will be published in conjunction with your paper and will include the anonymous referee reports, your point-by-point response and all pertinent correspondence relating to the manuscript. Let us know whether you agree with the publication of the PRF and as here, if you want to remove or not any figures from it prior to publication.

Please note that the Author checklist will be published at the end of the PRF.

Molecular Systems Biology has a "scooping protection" policy, whereby similar findings that are published by others during review or revision are not a criterion for rejection. Should you decide to submit a revised version, I do ask that you get in touch after three months if you have not completed it, to update us on the status.

Yours sincerely,

Poonam Bheda, PhD
Scientific Editor

Reviewer #1:

The paper by Heller et al is interesting, deals with a relevant biological question, but is also very complex and hard to follow. The thesis is that the relationship between gene copy number, gene dosage, and protein synthesis is not linear and in fact undergoes a buffering modality to maintain a certain degree of homeostasis in gene/chromosome gain or loss. The topic, seen from a distance, could be of clinical relevance, and the authors do try to make a connection between buffering capacity and drug resistance.

The paper is an in-depth analysis of the relationship between aberrant gene expression and protein quality control in cancer cells. At present, it is appreciated that aneuploidy also creates chaos at the protein quality control level. Here, Heller et al propose a compensatory mechanism through which the cell buffers protein synthetic output bidirectionally relative to gene gain or loss.

Because of its complex nature, the paper raises numerous concerns:

1. In Fig. 2C, the authors use a cumulative aneuploid score. However, since their starting point is a granular analysis of copy number gain/loss, in addition to a comprehensive aneuploid score. Results should also be represented as chromosome, arm and focal aneuploidy categories. This will provide an understanding of BR as a measure in a real context.
2. Fig. 2 shows that the argument for liquid (in suspension) tumors would be best served by analyzing multiple myeloma, as this cancer is liquid, has an active protein synthesis and secretion machinery, and has a different prognosis that varies based on aneuploidy status. How is BR in this case?
3. Fig. 3. Although mentioned in the manuscript, the definition of protein buffering remains ambiguous (see methodological comment below). If correctly understood, the mean dependency score (defined as $d_{high} - d_{low}$, where d_{high} and d_{low} represent the 80th and 20th percentiles of BR, respectively) is effectively a re-expression of the BR itself. As a result, it yields an AUROC of 1.0-but what does this actually imply? In both Figure 3A and 3B, triplosensitivity achieves a high AUROC (>0.8), which appears to contradict the concept of dosage compensation (or protein buffering). Triplosensitivity, by definition, refers to a phenotypic effect from an extra gene copy, suggesting that protein levels are not buffered. Therefore, if protein buffering were to occur, one would expect triplosensitivity to more directly predict the 'scaling' class, rather than the 'buffering' class.
4. Fig.4. How is the SHAP value ordered? If it means absolute mean importance, it should converge from the top (greater effect size) to the bottom. Also, if mean gene dependency has strong AUROC in Fig. 3, why is this feature importance in Fig.4D not considered as the strongest?

The legend at the bottom of Fig. 4D, E (left arrow Buffering, right arrow Scaling) is confusing and can be mistaken for Fig. 4F.

5. Fig. 5 G. The authors show that as the BR increases, the UPR demand decreases, particularly at lower ploidy score. However, in the same section, the authors conclude "This in combination with the downregulation of the CCT/TRiC complex suggests an increase in protein folding quality control in cell lines with high average protein buffering while demand for overall protein folding may be decreased", which seems to contradict the results.

The identification of the UPR-related gene set is intriguing. However, Figure 5F shows that the UPR signature is observed only in the CPTAC dataset, and in the opposite direction compared to expectations. This raises concerns, as the initial identification of the UPR gene set in Figure 5A was based on the ProCan dataset, where the UPR signal is not significant in Figure 5F ProCan dataset. Moreover, in the CPTAC dataset, the association becomes significantly negative (Fig. 5F). These inconsistencies between datasets warrant clarification.

The acronym eAS is not defined anywhere in the paper and it is difficult to know what one is looking at.

6. Fig. 6. Fig. 6D demonstrates a reasonable separation of drug response based on sample buffering. However, Figures S6C-D suggest that most of the variance is primarily explained by aneuploidy, which raises concerns about the findings in Figure 6D. Although the difference in drug response among highly buffered samples within highly aneuploid cells is statistically significant ($p = 0.021$), the odds ratio (OR = 1.088) indicates only a marginal effect, especially after extreme sample stratification using BR $>80\%$ versus the BR $< 20\%$. I recommend rephrasing this interpretation to avoid potentially misleading conclusions, as aneuploidy appears to account for the majority of the variance.

7. Fig. 6E shows data suggesting that in cell lines treated with different drugs, the BR is more relevant than aneuploidy in predicting drug response and cell viability. Whereas these experiments would be of immediate practical relevance, the are issues that make them unconvincing. (a) The Materials and Methods section does not provide methodologies and refers to <https://depmap.org/repurposing/>. This site shows 25 drug categories. The authors selected six and included the category Microtubule inhibitor which is not present in the site. This shows cherry picking and the introduction of a category that does not

exist in the primary data used in the analysis. (b) The authors state "The increased drug resistance was largely explained by increased aneuploidy, however, high buffering samples were still slightly more frequently drug resistant independent of aneuploidy status". However, Fig. S6 D does not show that accounting for buffering is significant. Therefore, the practical value of buffering seems to be lost in aneuploidy, and the present effort adds nothing new to what we already know.

8. Method are complicated and hard to understand. A few comments I have on the BR here:

In the Methods section, several manually defined thresholds lack sufficient justification. For example, the parameters *cb* and *pb* are defined inconsistently across datasets. On page 33, *cb* and *pb* are described as "we calculated the copy number and protein abundance baselines *cb* and *pb* for each gene as the median absolute gene CN and median protein abundance across all samples where the respective chromosome was disomic". This suggests that *cb* and *pb* represent the median values across a subset of the dataset. Is this interpretation correct? Are *cb* and *pb* meant to reflect the global medians under disomic conditions?

Yet, on page 34, a different approach is taken: "For the tumor sample dataset CPTAC, we applied the GeneCN analysis variant differently by assuming a copy number baseline of $cb = 2$ for all genes". What is the rationale behind setting $cb = 2$ for tumor samples? Why not apply the same median-based definition used on page 33? Is this due to the unavailability or unreliability of disomic reference samples in the tumor dataset? Additional clarification on this decision by the authors would help support the methodological consistency and transparency.

Finally, as a general comment, why use $> 80\%$ vs $< 20\%$ to define high buffering vs low buffering? Any sensitivity analysis to justify the cutoff point?

9. General comment. The authors present a multitude of data, and it is not always clear if the data refer to the tumor, tumor cells, or engineered cells. Nor is the demarcation between the effect of aneuploidy and buffering (BR) about these conditions always clear. Because this is relevant to the point the authors want to make and could have some practical utility, it is suggested that the authors develop a chart, table, heat map etc. in which the results of their analysis are reported in a synoptic way relative to the various experimental categories.

Reviewer #2:

The ms by Storchova, Sheltzer and colleagues have combined the use of ProteOMICs data of engineered cell lines, cancer cell lines and tumor samples with multifactorial machine learning methods to address the extent and potential role of protein buffering to aneuploidy. Authors unravel widespread protein buffering, but interesting differences between gene classes, gains vs losses and cellular contexts (engineered cells vs tumor cell lines/samples). They identify essential roles of gene essentiality, protein complex participation, gains vs losses and cellular context to the extent of protein buffering. The paper is well introduced, written and discussed, the figures are self-explanatory and data and conclusions are highly relevant in the field of aneuploidy and its consequences, and their impact in different types of cancer, either solid or liquid, either adults or pediatric is very high. Authors manage to synthesize complex data into specific questions and solid approaches to address them. I have nothing else to say but to support it for publication. It is indeed the first paper where this interesting aspect of aneuploidy (protein buffering) is thoroughly addressed between samples to reinforce its importance but also the differences between cellular contexts and gene types.

Reviewer #3:

Summary

In this paper, Heller et al. leverage genomic and proteomic datasets available for cancer cell line collections (DepMap, ProCan) and for a tumor sample collection (CPTAC), as well as proteomic data generated for RPE-1 cells engineered to carry monosomies or trisomies, to define the molecular characteristics of proteins buffered in aneuploid cancer cells. They calculate buffering ratios between observed and expected (copy number-dependent) proteomic load, considering both aneuploidy-driven and focal gene copy number variations. Using these buffering ratios, they then quantify the extent of buffering in different cell lines (DepMap, ProCan), detecting noticeable differences in buffering ability between cell lines, and demonstrating that protein buffering also occurs widespread in cancer samples. They continue to define predictive factors of protein buffering for chromosome gains and chromosome losses in unifactorial and multifactorial (machine-learning supported) contribution analyses. They find that gene essentiality, protein complex membership, and the number of ubiquitination sites are associated and predictive for buffering. In contrast, the frequency of random allelic expression, the number of regulatory sites on a protein, and the number of transcription factors interactions are predictive for scaling. Finally, they investigate whether the extent of buffering in cancer cell lines is related to drug sensitivity, showing that low-buffering cell lines are more sensitive to various drug classes, whereas high-buffering lines tend to be more drug-resistant.

General remarks

This is a thorough and interesting study on proteome buffering in cancer that quantifies attenuation across

cell lines and different tumors, and reinforces previously suggested associations between protein properties and buffering likelihood. The study also looks further to the association between buffering in aneuploidy and drug resistance in cancer, and I enjoyed reading this section in particular. Although the predictive buffering factors identified are not novel and have been suggested in prior studies, this study's main merit lies in the comprehensiveness in which it was done - leveraging data of two big cell line collections and a tumor sample collection to obtain generalizable results. It is also especially nice that the authors decided to not only look at aneuploidy-based copy number variations (so copy number variations that are based on the aneuploidy annotation of a chromosome arm), but also include focal gene copy number variants. The analyses they present are very detailed and rich, and mostly presented and described well in the text. The systematic analysis also raises interesting questions for future research, such as why adherent cells appear to buffer aneuploidy more effectively than suspension cells. Overall, this is a very well conducted, clearly presented study of interest to the field of aneuploidy and cancer research, and, due to its analytical nature, it will also appeal to a systems biology audience. I would therefore recommend the paper for publication, and suggest that the following points are addressed beforehand.

Major points

1. Besides increasing predictive power, it is not really clear what new biological insights the multifactorial machine learning approach added in comparison to the unifactorial correlation analysis (that has been conducted similarly for cancer cells in earlier work of the authors i.e. Schukken & Sheltzer, 2022, as well as recent yeast work done by others. The authors should highlight better what, if any, factors they newly identified as important for buffering using the ML approach. They also might want to consider whether the explainable machine learning should really be the emphasis of the paper as indicated by the title, or whether they could find a title that encompasses the other (equally important, and in my opinion more interesting) facets of their work better. A change of the title is of course up to the discretion of the authors and not mandatory for addressing this point.
2. The presentation and description of the SHAP value analysis is not clear enough. Why are the SHAP values for "mean gene dependency" > 0 for gene copy gains (4D) and therefore on the "scaling" side of the plot, yet associated with buffering in the text? Biologically, gene dependency being associated with buffering makes sense and is also picked up in the unifactorial analysis, but the SHAP analysis description seem incoherent with the figure to me. As another example, the concluding paragraph for this result section states "We showed that increased mean gene dependency and protein complex participation

contribute to a higher probability of protein buffering, whereas a higher number of activating TF interactions, protein regulatory sites, and higher RAE frequency contribute to a lower probability of protein buffering.", yet random allelic expression is on the same side (SHAP score > 0 , scaling) as protein complex participation and mean dependency in Fig. 4D? The representation of the SHAP results and the discussion of them is not yet intuitive enough for the reader, and improving them would greatly enhance the paper.

3. The authors discuss the down-, but not the upregulated pathways in the GSEA analysis using MSigDB hallmark gene sets in detail. For example, they analyze whether the enrichments they observe can mainly be attributed to aneuploidy or are a unique feature of high vs. low buffering cells/tumor samples, highlighting that the tumor samples (CPTAC) show different behavior compared to the cancer cell lines for many of the highly and significantly downregulated pathways (Fig. S5H, E2F targets, MTORC1 signaling, DNA repair). However, the authors do not discuss the potential relevance of the pathways which show the opposite pattern (epithelial mesenchymal transition, myogenesis, UV response, apical junction). The authors should expand on this. In addition, it would be great if Fig. S5H would be part of a main figure since it is discussed and analyzed extensively.

Minor points

1. Fig. 1B is partially confusing. What's with the BR > 0.2 white stripe in the middle? This figure should be improved, and the authors should here also refer to Table 2 in the text.
2. The authors should define how "relative abundance" in Fig. 1C was calculated, i.e. what is the "protein abundance baseline" here?
3. Please define the unit of the y axis in Fig. 1D.
4. It's surprising that the distributions of buffering ratios for DepMap is so different for gene copy number vs. chromosome arm (DepMap in Fig. 1H vs Fig. 1I). Could the authors explain this difference?
5. Transcription factors seem to play a strikingly different role for WGD vs non-WGD data; but only in chromosome gains. The authors should speculate why this could be the case (Fig. 3A, B).
6. Although it is probably a measure confounded by and contributing to many others, it would be interesting to test whether variability of protein expression influences buffering.
7. In addition to the major comment regarding the SHAP value analysis, maybe the factors in Fig. 4D, E could also be ordered in some way so that it is easier to assess the (predicted) contributions to buffering they have?
8. Could the authors speculate on why the number of protein complexes a protein participates in is predictive for buffering for gene CN gains, yet the number of protein-protein interactions is not (Fig. 4D)? Or maybe I am misunderstanding the SHAP value analysis, then please refer to major comment 2.

9. The authors state "While buffering in pan-cancer cell lines and tumor samples is comparable, chromosome-engineered aneuploid human cell lines are buffered to a greater extent. The underlying causes of these differences remain unclear". This finding is incongruent with work in yeast cells, where lab-engineered strains tend to buffer aneuploidy less efficiently - the authors should add this point to the discussion and speculate why this is the case. Also, for this comparison, the authors rely on ChrArm copy number changes (not gene CN changes) (Fig. 1F), could this skew the results?
10. For Fig. 6E, the authors should explain in the figure legend how "drug effect" is defined.
11. In the discussion, the authors mention "In particular, the BR is sensitive to noise from both proteomics and copy number data. To circumvent these issues, we calculated confidence scores for the BRs, and removed BR measurements with low confidence." - I thought this was a good idea, and therefore the authors could briefly mention this point in the results section and not just "hide" it in the methods.

Point-by-point response to the reviewers' comment.

The complete reviewers comments are in black, the responses in blue.

Reviewer #1:

The paper by Heller et al is interesting, deals with a relevant biological question, but is also very complex and hard to follow. The thesis is that the relationship between gene copy number, gene dosage, and protein synthesis is not linear and in fact undergoes a buffering modality to maintain a certain degree of homeostasis in gene/chromosome gain or loss. The topic, seen from a distance, could be of clinical relevance, and the authors do try to make a connection between buffering capacity and drug resistance.

The paper is an in-depth analysis of the relationship between aberrant gene expression and protein quality control in cancer cells. At present, it is appreciated that aneuploidy also creates chaos at the protein quality control level. Here, Heller et al propose a compensatory mechanism through which the cell buffers protein synthetic output bidirectionally relative to gene gain or loss.

Because of its complex nature, the paper raises numerous concerns:

1. In Fig. 2C, the authors use a cumulative aneuploid score. However, since their starting point is a granular analysis of copy number gain/loss, in addition to a comprehensive aneuploid score. Results should also be represented as chromosome, arm and focal aneuploidy categories. This will provide an understanding of BR as a measure in a real context.

>> Aneuploidy score provides a cumulative quantification of chromosome alterations present in a sample, such as a tumor or cell line, providing important context information about respective tumors. In Figure 2C, we present the mean BR per gene, but we also calculated the mean aneuploidy score across all samples of a cancer type and used this mean aneuploidy score solely to visualize the genomic context to the mean buffering ratios (BR). The intent of Figure 2C is to provide an overview of how the BR is distributed among different cancer types and contextualize this information by the average aneuploidy, as the degree of aneuploidy is a potential confounding factor. It is basically impossible to visualize the data as requested by the reviewer due to their complexity. We provided granular per-gene per-sample buffering ratios in Supplementary table 1. The table provides the BRs in a gain/loss dependent context using both chromosome arm and focal gene copy numbers. We have now clarified these points in the manuscript text.

2. Fig. 2 shows that the argument for liquid (in suspension) tumors would be best served by analyzing multiple myeloma, as this cancer is liquid, has an active protein synthesis and secretion machinery, and has a different prognosis that varies based on aneuploidy status. How is BR in this case?

>> We thank the reviewer for this interesting idea. We used the publication by Sarin et al. (Leukemia 2020, DOI: 10.1038/s41375-020-0785-1) and the OncoTree classification for Plasma Cell Myeloma (PCM) to identify 16 unique multiple myeloma cell lines. We observed similar patterns in these cell lines as described in Figure 2 (see Figure R1). An increased degree of aneuploidy correlated with increased buffering in multiple myeloma cell lines (Spearman's rho = 0.65). Furthermore, similar to MNM/LNM in Figure 2, multiple myeloma cell lines showed a lower sample BR than cell lines from other cancer types. The analysis is in agreement with the findings based on liquid tumors and does not provide additional insights. Therefore, we decided to keep the original figure which is additionally based on a larger number of samples.

Figure R1: Multiple Myeloma cell lines show insignificantly decreased average buffering compared to other cancer types (left; Wilcoxon rank-sum test). Multiple Myeloma cell lines show a positive association between the degree of aneuploidy and average buffering (right; Spearman's correlation coefficient).

3. Fig. 3. Although mentioned in the manuscript, the definition of protein buffering remains ambiguous (see methodological comment below). If correctly understood, the mean dependency score (defined as $d_{high} - d_{low}$, where d_{high} and d_{low} represent the 80th and 20th percentiles of BR, respectively) is effectively a re-expression of the BR itself. As a result, it yields an AUROC of 1.0- but what does this actually imply? In both Figure 3A and 3B, triplosensitivity achieves a high AUROC (>0.8), which appears to contradict the concept of dosage compensation (or protein buffering). Triplosensitivity, by definition, refers to a phenotypic effect from an extra gene copy, suggesting that protein levels are not buffered. Therefore, if protein buffering were to occur, one would expect triplosensitivity to more directly predict the 'scaling' class, rather than the 'buffering' class.

>> The mean dependency score is not a re-expression of the BR. It is a metric from large-scale functional genomics screens, e.g., the Cancer Dependency Map (DepMap) project, that measures how much a cancer cell line depends on a particular gene for survival – gene dependency score. The mean gene dependency used in the factor analysis (Figure 3 and 4) is defined as the mean gene dependency score for each gene across all cell lines. Thus, Figure 3 represents how well the average dependency on a particular gene can distinguish whether the protein is buffered or scaling. Furthermore, we emphasize that Figure 3A&B display normalized ranks of ROC AUCs, not plain ROC AUCs, as stated in the Figure Legend. In the bootstrapped analysis, the highest median ROC AUC observed was 0.63 (Figure 3C).

The confusion might have arisen due to the analysis of mean dependency score presented in Figure 6. Here, we analyze the differential dependency on certain genes when comparing cell lines with a high degree of average buffering against cell lines with a low degree of buffering, calculated as $d_{high} - d_{low}$ (x-axis of Figure 6A, "Dependency Score Difference").

We have now adjusted the methods section, the figure legend, color scales, and main text accordingly to increase clarity.

Triplosensitivity (i.e., duplication intolerance; R. Collins, et al., Cell 2022) was ranking high in the unifactorial analysis, indicating that the probability score for triplosensitivity was better suited to distinguish buffered and scaling proteins than other factors (Figure 3A&B). Yet, the median ROC AUC was relatively low (0.52-0.55). Correlation analysis suggests that an increased probability of triplosensitivity of a protein is associated with a stronger degree of buffering (Figure S3E). Indeed, an increased triplosensitivity may suggest that cells do not efficiently buffer the abundance of this particular protein and are therefore susceptible to alterations in the copy number. Alternatively, proteins with high triplosensitivity might need to be buffered to some extent to enable survival, whereas cells that harbor triplosensitive proteins with increased copy number without any buffering might not survive at all. This second hypothesis, which involves survivorship bias and posits that

proteins may be both triplosensitive and buffered, appears more likely given the observed positive association between triplosensitivity probability and the extent of buffering (Figure S3E).

4. Fig.4. How is the SHAP value ordered? If it means absolute mean importance, it should converge from the top (greater effect size) to the bottom. Also, if mean gene dependency has strong AUROC in Fig. 3, why is this feature importance in Fig.4D not considered as the strongest?

The legend at the bottom of Fig. 4D, E (left arrow Buffering, right arrow Scaling) is confusing and can be mistaken for Fig. 4F.

>> The SHAP values are ordered according to their contribution to prediction, but it was not intuitively visible from the plot. We thank the Reviewer for pointing this out. The raw SHAP values indicate how influential a feature is for an individual prediction relative to the baseline prediction (also called ϕ_0). In our case, the baseline prediction is the fraction of scaling samples in the training set of a given condition (e.g. CPTAC, Gene CN Loss). The SHAP values were calculated from the test set and are relative to ϕ_0 . Based on the baseline predictions, the exact genes in the test set, and the way the model uses a feature for prediction, the distribution of SHAP values can shift and thus their median might not be centered to 0. This can lead to SHAP values that are entirely located on the positive side implying that the inclusion of the feature supports the model in making decisions towards the Scaling class, which is, however, not the case (e.g., in the case of Mean Gene Dependency in Figure 4D). In such a case, the availability of the factor data alone shifted the prediction of the model independent of the individual values of the factor (e.g., high or low average dependency on a gene). We have now centered the SHAP values around 0, which makes the visualization more intuitive, and added the Spearman's ρ used for the ordering. Nevertheless, we noted that the raw SHAP values are more useful for understanding the model's technical behavior rather than offering direct biological interpretation and need to be viewed in the context of the underlying value of the factor. Additionally, the SHAP value plot presented data only for CPTAC. For these reasons, we moved the raw SHAP value plot to the Extended View figures (Figure EV4D&E). We have also adjusted the text to improve understanding.

Additionally, it should be noted that Fig. 3 shows results based on unifactorial model, while Figure 4 on multifactorial model, and the heat maps thus depict different results, as explained in the figure legend. We now emphasized this difference in the chapter titles and figure legends.

5. Fig. 5 G. The authors show that as the BR increases, the UPR demand decreases, particularly at lower ploidy score. However, in the same section, the authors conclude "This in combination with the downregulation of the CCT/TRiC complex suggests an increase in protein folding quality control in cell lines with high average protein buffering while demand for overall protein folding may be decreased", which seems to contradict the results.

The identification of the UPR-related gene set is intriguing. However, Figure 5F shows that the UPR signature is observed only in the CPTAC dataset, and in the opposite direction compared to expectations. This raises concerns, as the initial identification of the UPR gene set in Figure 5A was based on the ProCan dataset, where the UPR signal is not significant in Figure 5F ProCan dataset. Moreover, in the CPTAC dataset, the association becomes significantly negative (Fig. 5F). These inconsistencies between datasets warrant clarification.

>>To understand this perceived discrepancy, one has to look specifically how each of the categories is defined and what genes of each category are enriched. When comparing the gene sets "CCT complex" (CORUM), "unfolded protein binding" (GO:MF) identified in the over-representation analysis (ORA; Figure 5A-C) and the "Unfolded Protein Response" gene set (UPR; MSigDB HALLMARK) used in the gene set enrichment analysis (GSEA; Figure 5F&G), there was no overlap between the significantly enriched genes in the ORA gene sets and in the GSEA gene set. The genes significantly enriched in the "unfolded protein binding" (GO:MF) gene set were CCT complex genes (TCP1, CCT2-5, CCT6A, CCT7, CCT8), as well as NACA, NAP1L4, NUDC, NUDCD2, PPIA, PTGES3, and TUBB4B. There was also no overlap between the UPR gene set and genes highlighted in figure 5E (UGGT1, CANX, TAPBP).

This shows that these gene sets cover different aspects: while the UPR is directly concerned with the

response to proteotoxic stress relevant to cancer hallmarks, the gene sets enriched in the ORA are related to the binding to unfolded proteins in general. However, these results also may suggest that some of the genes deregulated in CPTAC are different to those in ProCan, as not all genes of the "unfolded protein binding" gene set are upregulated genes in high-buffering cell lines. To understand what parts of the UPR are deregulated differently between cell lines and tumor samples, we performed ORA on subsets of the UPR hallmark gene set. The subset of genes that were upregulated in both ProCan and CPTAC were related to protein maturation (DNAJC3, HYOU1, PDIA5, SPCS3). The subset of genes commonly downregulated were related to translation initiation factor activity and translation regulation in response to ER stress (EIF2S1, EIF4A1, EIF4G1). Genes downregulated in CPTAC and upregulated in ProCan were related to protein folding (CALR, HSP90B1, HSPA5) and rRNA metabolic processes (DDX10, EIF4A3, NOP56, RRP9). Genes upregulated in CPTAC and downregulated in ProCan were related to deadenylation-dependent mRNA decay (EIF4A2, LSM4). This shows that in cell lines with high buffering, the response to unfolded proteins is heterogeneous and not as clear as in tumor samples. While the CCT complex is downregulated (which is not considered in the UPR gene set), chaperones like CALR, HSP90B1, and HSPA5 were upregulated. These results might also suggest that these cell lines might be in a transition state, where many of the proteins are already buffered, while some of the proteotoxic stress is still being dealt with, especially as ER stress-mediated translation regulation is reduced in both tumor samples and cell lines. We updated the manuscript to better reflect the complexity of the unfolded protein response in highly buffering cell lines.

The acronym eAS is not defined anywhere in the paper and it is difficult to know what one is looking at.

>> We thank the reviewer for pointing this out. The acronym eAS stands for "estimated aneuploidy score", but in fact it is just the AS (aneuploidy score), calculated slightly differently for CPTAC due to the somewhat different type of data provided by this database. We now use the abbreviation AS throughout the manuscript to improve clarity and refer to the methods section containing a description of how the aneuploidy score was calculated for CPTAC.

6. Fig. 6. Fig. 6D demonstrates a reasonable separation of drug response based on sample buffering. However, Figures S6C-D suggest that most of the variance is primarily explained by aneuploidy, which raises concerns about the findings in Figure 6D. Although the difference in drug response among highly buffered samples within highly aneuploid cells is statistically significant ($p = 0.021$), the odds ratio (OR = 1.088) indicates only a marginal effect, especially after extreme sample stratification using BR>80% versus the BR < 20%. I recommend rephrasing this interpretation to avoid potentially misleading conclusions, as aneuploidy appears to account for the majority of the variance.

>> This is indeed an important point. We rephrased the statement to better highlight that in general there is only a marginal general effect of buffering on drug sensitivity, however, we point out that Heat Shock Protein and Checkpoint Kinase inhibitors show specifically increased resistance in highly buffering independently of aneuploidy.

7. Fig. 6E shows data suggesting that in cell lines treated with different drugs, the BR is more relevant than aneuploidy in predicting drug response and cell viability. Whereas these experiments would be of immediate practical relevance, they are issues that make them unconvincing. (a) The Materials and Methods section does not provide methodologies and refers to <https://depmap.org/repurposing/>. This site shows 25 drug categories. The authors selected six and included the category Microtubule inhibitor which is not present in the site. This shows cherry picking and the introduction of a category that does not exist in the primary data used in the analysis. (b) The authors state "The increased drug resistance was largely explained by increased aneuploidy, however, high buffering samples were still slightly more frequently drug resistant independent of aneuploidy status". However, Fig. S6 D does not show that accounting for buffering is significant. Therefore, the practical value of buffering seems to be lost in aneuploidy, and the present effort adds nothing new to what we already know.

>> Drug metadata, such as the mechanism of action (MOA), were derived from the table "primary-screen-replicate-collapsed-treatment-info.csv" obtainable from <https://depmap.org/repurposing/>. This table contains 852 different drug MOAs, including microtubule inhibitors such as colchicine. Figure 6E presents only the drug mechanisms that reached significance in both the Log2FC-based drug effect score analysis and the correlation-based analysis (sample BR MNR ~ median drug effect), as described in the legend of Figure 6E and in subsection "Drug Sensitivity Analysis" in Methods. We apologize for this confusion and clarified the source of the metadata in the manuscript. Separating the effect of aneuploidy effect from the buffering effect remains the most critical issue of our analysis, as buffering is linked to aneuploidy. While there is no significant difference in the strength of the median drug effect on cell line growth when comparing high and low buffering groups within aneuploidy groups, we observed, after stratifying cells into high-/low-buffering, high-/low-aneuploidy, and high-/low-resistance, a significant, albeit small, increase in high-resistance frequency in cells with high buffering compared to low buffering (Figure S6E, OR = 1.088, p = 0.021). Therefore, there is a small effect that can be attributed to buffering that is not explained by aneuploidy. This is also reflected by the drug mechanism analysis. The sensitivity to HSP and CHK inhibitors was increased when comparing high against low aneuploid cells (CHKi: 0.074 drug effect Log2FC, p_{adj} = 0.71; HSPi: 0.179 Log2FC, p_{adj} = 0.165), while being significantly decreased when comparing high against low buffering (Figure 6E, Supp. table 10).

8. Method are complicated and hard to understand. A few comments I have on the BR here: In the Methods section, several manually defined thresholds lack sufficient justification. For example, the parameters cb and pb are defined inconsistently across datasets. On page 33, cb and pb are described as "we calculated the copy number and protein abundance baselines cb and pb for each gene as the median absolute gene CN and median protein abundance across all samples where the respective chromosome was disomic". This suggests that cb and pb represent the median values across a subset of the dataset. Is this interpretation correct? Are cb and pb meant to reflect the global medians under disomic conditions? Yet, on page 34, a different approach is taken: "For the tumor sample dataset CPTAC, we applied the GeneCN analysis variant differently by assuming a copy number baseline of cb = 2 for all genes". What is the rationale behind setting cb = 2 for tumor samples? Why not apply the same median-based definition used on page 33? Is this due to the unavailability or unreliability of disomic reference samples in the tumor dataset? Additional clarification on this decision by the authors would help support the methodological consistency and transparency.

>> We thank the reviewer for these comments and adjusted the Methods section to improve clarity. Indeed, c_b and p_b represent the median copy number and protein abundance values of a protein under disomic conditions across cell lines in a dataset. However, this approach could not be used for CPTAC. This dataset provides data on matched normal tissue samples (assumed to be disomic) to tumor tissue samples. To quantify the absolute copy number changes, the AscatNGS algorithm was used, which compares tumor samples against normal tissue samples. For proteins, we directly calculated the protein abundance baseline using normal tissues.

Finally, as a general comment, why use > 80% vs < 20% to define high buffering vs low buffering? Any sensitivity analysis to justify the cutoff point?

>> In fact, we tried multiple cutoffs and selected those that optimize the following criteria:

1. Number of cancer types – we want to cover as many cancer types as possible to avoid type-specific effects;
2. Number of significant hits for each dataset;
3. Number of observations (cell lines / tumor samples);
4. Robustness of results across datasets. We defined robustness as the Jaccard index of significantly deregulated genes across datasets, i.e., the number of significant genes that are common in all datasets divided by the number of significant genes identified in total. This inter-dataset robustness was calculated across cell line datasets. We performed this sensitivity analysis across multiple cutoff combinations in increments of 5% (Figure R2).

We found that the 80/20 cutoff provides the best balance for our optimization goals. An alternative cutoff would have been 75/25. This cutoff would have increased the number of hits for CPTAC but

decreased the number of hits in the cell line datasets and decreased the inter-dataset robustness. Accordingly, we adopted the 80/20 cutoff, applying it uniformly to all analyses and datasets to enable fair comparisons.

Figure R2: Results of the sensitivity analysis to determine sample buffering ratio cutoffs for dataset stratification. Values were calculated by performing differential expression analyses for each pair of cutoffs. A gene was considered significant if $|\text{Log}_2\text{FC}| > 1$ and $p.\text{adj} < 0.05$ (Benjamini-Hochberg

procedure). Inter-dataset robustness describes the Jaccard index of significantly enriched genes across cell line datasets.

9. General comment. The authors present a multitude of data, and it is not always clear if the data refer to the tumor, tumor cells, or engineered cells. Nor is the demarcation between the effect of aneuploidy and buffering (BR) about these conditions always clear. Because this is relevant to the point the authors want to make and could have some practical utility, it is suggested that the authors develop a chart, table, heat map etc. in which the results of their analysis are reported in a synoptic way relative to the various experimental categories.

>> This is an interesting idea. As EMBO Molecular Systems Biology requires a separate synopsis for every published paper, we generated a figure summarizing the procedure and results and provided it according to the submission guidelines.

Reviewer #2:

The ms by Storchova, Sheltzer and colleagues have combined the use of ProteOMICs data of engineered cell lines, cancer cell lines and tumor samples with multifactorial machine learning methods to address the extent and potential role of protein buffering to aneuploidy. Authors unravel widespread protein buffering, but interesting differences between gene classes, gains vs losses and cellular contexts (engineered cells vs tumor cell lines/samples). They identify essential roles of gene essentiality, protein complex participation, gains vs losses and cellular context to the extent of protein buffering. The paper is well introduced, written and discussed, the figures are self-explanatory and data and conclusions are highly relevant in the field of aneuploidy and its consequences, and their impact in different types of cancer, either solid or liquid, either adults or pediatric is very high. Authors manage to synthesize complex data into specific questions and solid approaches to address them. I have nothing else to say but to support it for publication. It is indeed the first paper where this interesting aspect of aneuploidy (protein buffering) is thoroughly addressed between samples to reinforce its importance but also the differences between cellular contexts and gene types.

>> We thank the reviewer for their positive evaluation.

Reviewer #3:

Summary

In this paper, Heller et al. leverage genomic and proteomic datasets available for cancer cell line collections (DepMap, ProCan) and for a tumor sample collection (CPTAC), as well as proteomic data generated for RPE-1 cells engineered to carry monosomies or trisomies, to define the molecular characteristics of proteins buffered in aneuploid cancer cells. They calculate buffering ratios between observed and expected (copy number-dependent) proteomic load, considering both aneuploidy-driven and focal gene copy number variations. Using these buffering ratios, they then quantify the extent of buffering in different cell lines (DepMap, ProCan), detecting noticeable differences in buffering ability between cell lines, and demonstrating that protein buffering also occurs widespread in cancer samples. They continue to define predictive factors of protein buffering for chromosome gains and chromosome losses in unifactorial and multifactorial (machine-learning supported) contribution analyses. They find that gene essentiality, protein complex membership, and the number of ubiquitination sites are associated and predictive for buffering. In contrast, the frequency of random allelic expression, the number of regulatory sites on a protein, and the number of transcription factors interactions are predictive for scaling. Finally, they investigate whether the extent of buffering in cancer cell lines is related to drug sensitivity, showing that low-buffering cell lines are more sensitive to various drug classes, whereas high-buffering lines tend to be more drug-resistant.

General remarks

This is a thorough and interesting study on proteome buffering in cancer that quantifies attenuation across cell lines and different tumors, and reinforces previously suggested associations between protein properties and buffering likelihood. The study also looks further to the association between buffering in aneuploidy and drug resistance in cancer, and I enjoyed reading this section in particular. Although the predictive buffering factors identified are not novel and have been suggested in prior studies, this study's main merit lies in the comprehensiveness in which it was done - leveraging data of two big cell line collections and a tumor sample collection to obtain generalizable results. It is also especially nice that the authors decided to not only look at aneuploidy-based copy number variations (so copy number variations that are based on the aneuploidy annotation of a chromosome arm), but also include focal gene copy number variants. The analyses they present are very detailed and rich, and mostly presented and described well in the text. The systematic analysis also raises interesting questions for future research, such as why adherent cells appear to buffer aneuploidy more effectively than suspension cells. Overall, this is a very well conducted, clearly presented study of interest to the field of aneuploidy and cancer research, and, due to its analytical nature, it will also appeal to a systems biology audience. I would therefore recommend the paper for publication, and suggest that the following points are addressed beforehand.

Major points

1. Besides increasing predictive power, it is not really clear what new biological insights the multifactorial machine learning approach added in comparison to the unifactorial correlation analysis (that has been conducted similarly for cancer cells in earlier work of the authors i.e. Schukken & Sheltzer, 2022, as well as recent yeast work done by others. The authors should highlight better what, if any, factors they newly identified as important for buffering using the ML approach. They also might want to consider whether the explainable machine learning should really be the emphasis of the paper as indicated by the title, or whether they could find a title that encompasses the other (equally important, and in my opinion more interesting) facets of their work better. A change of the title is of course up to the discretion of the authors and not mandatory for addressing this point.

>> Judging by the ranking of the unifactorial correlation plot, where we correlated the Buffering Ratio and feature values directly (Figure 3E), we obtain a similar set of features that generally are predictive for buffering (gene dependency, complex participation, ubiquitination & SUMOylation sites, haploinsufficiency, mRNA decay, protein loops, protein-protein interactions) and for scaling (random allelic expression, transcription factors, protein regulatory sites). While this confirms that the factors we identified via machine learning can be also identified when looking at each factor separately, it is impossible to confidently state a trend based on the unifactorial analysis since the correlation is weak (absolute correlation below 0.27). The machine learning-based approach not only allowed for stronger prediction, but showed that a combination of factors and their non-linear relationships is required for confidently predicting buffering. Only with this combined model were we able to decipher the strongest contributors and demonstrate that their contribution to predicting buffering was much stronger than when viewed individually.

To show how these factors work together and identify functionally coordinated groups of factors, we performed a hierarchical clustering and dimensionality reduction of the raw SHAP values in different conditions. We saw that the mean gene dependency, protein complex participation, and ubiquitination sites consistently cluster together across multiple conditions, which confirms that they predict buffering similarly, and suggests that they might be functionally connected in their contribution to protein buffering (figure R3). Besides showing stronger trends and possible functional coordination, our multifactorial approach also revealed more nuanced relationships compared to the unifactorial correlation analysis. In cell lines, non-exponential decay predicts protein buffering more strongly upon gene copy number loss than upon gain. Furthermore, are mRNA decay and translation rate more predictive towards buffering upon gene copy number gain. In total, the multifactorial analysis allowed us to move beyond single-factor associations (as done by Schukken & Sheltzer 2022) to identify functional groups of predictors, highlight context-specific effects, and determine directional trends between buffering and factors that would have remained hidden otherwise. We

adapted the manuscript text to point out these differences and clarify the role of our multifactorial analysis. Moreover, we highlight the difference between the unifactorial ROC AUC analysis (which only tells how well a feature can separate Buffered and Scaling groups, but does not provide any direction of the prediction) and the multifactorial correlation analysis (which provides a directional trend between the feature value and the chance of buffering).

Therefore, we are still convinced that the use of machine learning was fundamental to our project and opt for maintaining the previous title.

Figure R3: Hierarchical clustering analysis of raw SHAP values shows functional coordination of factors. Left: Clustering based on SHAP-values generated from all pan-cancer datasets (DepMap, ProCan, CPTAC; GeneCN-derived BR; Gain & Loss). Right: Clustering based on SHAP-values generated from CPTAC (GeneCN-derived BR; Gain & Loss).

2. The presentation and description of the SHAP value analysis is not clear enough. Why are the SHAP values for "mean gene dependency" > 0 for gene copy gains (4D) and therefore on the "scaling" side of the plot, yet associated with buffering in the text? Biologically, gene dependency being associated with buffering makes sense and is also picked up in the unifactorial analysis, but the SHAP analysis description seem incoherent with the figure to me. As another example, the concluding paragraph for this result section states "We showed that increased mean gene dependency and protein complex participation contribute to a higher probability of protein buffering, whereas a higher number of activating TF interactions, protein regulatory sites, and higher RAE frequency contribute to a lower probability of protein buffering.", yet random allelic expression is on the same side (SHAP score > 0, scaling) as protein complex participation and mean dependency in Fig. 4D? The representation of the SHAP results and the discussion of them is not yet intuitive enough for the reader, and improving them would greatly enhance the paper.

>> We thank the reviewer for pointing out that the raw SHAP values in Figure 4D and E are difficult to interpret and seem incoherent with following analyses. It should be noted that raw SHAP values cannot be directly interpreted in terms of biological contribution of each analyzed factor towards buffering or scaling. The raw SHAP values indicate how influential a feature is for an individual prediction relative to the baseline prediction (also called ϕ_0). In our case, the baseline prediction is the fraction of scaling samples in the training set of a given condition (e.g. CPTAC, Gene CN Loss). The SHAP values were calculated from the test set and are relative to ϕ_0 . Based on the baseline predictions, the exact genes in the test set, and the way the model uses a feature for prediction, the distribution of SHAP values can shift and thus their median might not be centered to 0. This can lead to SHAP values that are entirely located on the positive side indicating that the inclusion of the feature always supports the model in making decisions towards the Scaling class (e.g., in the case of

Mean Gene Dependency in Figure 4D). As a consequence, although many of the proteins are buffered (as 50% of the test set are buffered proteins), all the SHAP values for Mean Gene Dependency are on the positive “Scaling” side even though the entire range of different dependency values (from 0 to 1) is covered. Therefore, the raw SHAP value is not indicative for how strong the prediction is towards buffering depending on the actual value of our factor (i.e., depending on whether there is a high or low mean gene dependence).

To avoid these technical artifacts that depend on how the model calculates buffering, we correlated the SHAP values with their respective feature values to uncover trends with biologically interpretable meaning. Here, for example, we observed that if the mean gene dependency increases, the SHAP values decrease. This gives the interpretation that highly essential genes are more likely to be buffered than non-essential genes. We plotted the relationship between SHAP value and feature value in a scatter plot (figure R4) which revealed that this relationship is non-linear. As the raw SHAP values are more helpful in understanding how the model operates and are of more technical than biologically interpretable nature, we moved the raw SHAP values to the supplement and improved the text (Figure EV4D&E). To confirm our results from a different analysis perspective, we performed a hierarchical clustering and dimensionality reduction of the raw SHAP values in different conditions and identified mean gene dependency, protein complex participation, and ubiquitination sites as a functionally coordinated group of factors (see answer to major point 1).

Figure R4: Scatterplot showing the non-linear relationship between normalized mean gene dependency and associated SHAP values (CPTAC, gene copy-number gain, test-test). The trend line is fitted using local polynomial regression (LOESS).

To increase clarity, we have now centered the SHAP values around 0, which makes the visualization more intuitive, and added the Spearman’s ρ used for their ordering. Nevertheless, we note that the raw SHAP values are more useful for understanding the model’s technical behavior rather than offering direct biological interpretation and need to be viewed in the context of the underlying value of the factor. Additionally, the SHAP value plot presented data only for CPTAC. For these reasons, we moved the raw SHAP value plot to the Extended View figures. We have also adjusted the text to improve understanding.

3. The authors discuss the down-, but not the upregulated pathways in the GSEA analysis using MSigDB hallmark gene sets in detail. For example, they analyze whether the enrichments they observe can mainly be attributed to aneuploidy or are a unique feature of high vs. low buffering cells/tumor samples, highlighting that the tumor samples (CPTAC) show different behavior compared to the cancer cell lines for many of the highly and significantly downregulated pathways (Fig. S5H, E2F targets, MTORC1 signaling, DNA repair). However, the authors do not discuss the potential relevance of the pathways which show the opposite pattern (epithelial mesenchymal transition,

myogenesis, UV response, apical junction). The authors should expand on this. In addition, it would be great if Fig. S5H would be part of a main figure since it is discussed and analyzed extensively.

>> We thank the reviewer for this suggestion. We did not discuss the gene sets myogenesis, UV response, and apical junction in the manuscript as their change in enrichment in high buffering groups was confounded by the change in enrichment induced by aneuploidy (Fig. S5H). We changed the text to highlight the EMT pathway and state that the DNA repair pathway is only significantly enriched in high-buffering cell lines.

Minor points

1. Fig. 1B is partially confusing. What's with the BR > 0.2 white stripe in the middle? This figure should be improved, and the authors should here also refer to Table 2 in the text.

>> We thank the reviewer for pointing out a lack of clarity in Figure 1B. The intent of the white stripe in the range between -0.3 and 0.3 DNA copy number Log2FC was to communicate that values in this range have an insufficient change in copy number in order to be confidently grouped into Scaling, Buffered, or Anti-Scaling. We originally set this cutoff when using relative copy numbers and removed BR values in this range. We changed our analysis pipeline to use absolute copy numbers generated by ABSOLUTE and AscatNGS and we note that the copy number Log2FC is either outside of the [-0.3, 0.3] range or zero when using absolute copy numbers; for the latter case no BR values were generated. To improve the clarity of figure 1B we moved the annotation of the "BR > 0.2" cutoff used for separating Buffered and Scaling groups to the individual Buffered groups, annotated the white areas with "Insufficient CN Change", and changed the phrasing in the figure legend.

2. The authors should define how "relative abundance" in Fig. 1C was calculated, i.e. what is the "protein abundance baseline" here?

>> The strategy to determine the protein abundance baseline is described thoroughly in Materials and Methods; we have now added a brief explanation to the figure legend.

3. Please define the unit of the y axis in Fig. 1D.

>> These values (labeled as "Protein Abundance") represent log2-transformed protein intensities reported by the DepMap CCLE dataset. These were quantified using Spectronaut from aggregated relative peptide intensities obtained from TMT 10-plex labeled experiments, and normalized using a custom pipeline (see Nusinow, et al., Cell 2020, <https://doi.org/10.1016/j.cell.2019.12.023>). We therefore added the indication "[a.u.]" for "arbitrary units" to the y-axis label of figure 1D.

4. It's surprising that the distributions of buffering ratios for DepMap is so different for gene copy number vs. chromosome arm (DepMap in Fig. 1H vs Fig. 1I). Could the authors explain this difference?

>> We thank the reviewer for pointing this out. This is largely a technical effect caused by the different strategies that have to be used to calculate the copy number difference depending on whether we used chromosome arm copy number data or gene copy number data. The gene copy number-derived buffering ratio (BR) describes the buffering against a disomic baseline, meaning the observed copy number of a gene is compared against a copy number of 2. The chromosome arm-derived BR describes the buffering against the rounded basal ploidy of the cell, meaning the observed chromosome copy number is the rounded ploidy (i.e., the chromosome copy number baseline) plus the chromosome copy number alteration (gain/loss/neutral, obtained from Cohen-Sharir et al., 2021). These two approaches provide different perspectives on buffering (absolute gene copy number against disomy, aneuploidy against euploidy), which explains the quantitative difference. Therefore, we always compared only the buffering ratio within each approach.

5. Transcription factors seem to play a strikingly different role for WGD vs non-WGD data; but only in chromosome gains. The authors should speculate why this could be the case (Fig. 3A, B).

>> We thank the reviewer for their comment. In addition to the stated observation that Transcription Factors seem to be more predictive upon chromosome arm / gene copy number gain especially in

WGD-positive cells, we observe that transcription factors are predictive towards scaling (Figure 4D, previously Figure 4F). Again, this trend is stronger upon chromosome arm / gene copy number gain in WGD-positive cells (Figure R5). We speculate that this is related to an increased number of promoters that transcription factors can interact within WGD positive cells. Increasing the copies of a gene offers more promoters for transcription factors to bind to, allowing for increased protein production. If the relationship between DNA copies, transcription factor interactions, and protein abundance was linear, a relative gain of a single copy in WGD cells would have less impact on protein scaling than in non-WGD cells as the ratio of increased transcription factor interactions would be lower (5/4 versus 3/2 assuming a ploidy of 4 for WGD and 2 for non-WGD). However, the observed scaling effect is stronger in WGD cells leading to the assumption that either the total number of transcription factors has a stronger influence on scaling, or that there are non-linear effects where increasing the number of transcription factor interactions does not increase protein expression by much, unless a certain threshold is reached. Thus, we cannot clearly entangle this question.

Figure R5: SHAP-Feature correlation heatmap, divided by WGD and non-WGD subsets of the DepMap dataset.

6. Although it is probably a measure confounded by and contributing to many others, it would be interesting to test whether variability of protein expression influences buffering.

>> We previously performed this analysis by calculating the coefficient of variation (CV) of protein abundance for each protein in samples where the protein was encoded on neutral chromosomes and correlating this Protein Neutral CV with the gene copy number-derived Buffering Ratio (Spearman's correlation coefficient). Based on the reviewer's suggestion, we further investigated and expanded the analysis by correlating the protein abundance CV across all samples with the Buffering Ratio. We observed no strong effect of protein abundance variation on Buffering in tumor samples (Fig. R6). In cell lines, we observed no strong correlation when using the raw Buffering Ratios, however, there was a negative correlation between the degree of protein variation and the average Buffering Ratio of a protein across samples (Fig. R7). This suggests that proteins with lower expression variability are more likely to be stronger buffered in cell lines. However, due to the difference between cell lines and tumor samples we cannot state that there is a clear influence of protein expression variability on buffering in general.

Figure R6: No strong correlation in tumor samples (CPTAC) between (1) protein abundance coefficient of variance (CV) and buffering ratio (BR), (2) protein abundance CV and gene-wise mean BR, (3) protein abundance CV on neutral chromosomes and BR, (4) neutral protein abundance CV and mean BR.

Figure R7: Correlation in cell lines (DepMap) between (1) protein abundance coefficient of variance (CV) and buffering ratio (BR), (2) protein abundance CV and gene-wise mean BR, (3) protein abundance CV on neutral chromosomes and BR, (4) neutral protein abundance CV and mean BR.

7. In addition to the major comment regarding the SHAP value analysis, maybe the factors in Fig. 4D, E could also be ordered in some way so that it is easier to assess the (predicted) contributions to buffering they have?

>> We thank the reviewer for their comment. The raw SHAP plot (now in Expanded View) is sorted by the absolute correlation coefficient between SHAP value and feature value to show features on the top of the plot that have the strongest trend. We added the correlation coefficient to the plot to improve clarity.

8. Could the authors speculate on why the number of protein complexes a protein participates in is predictive for buffering for gene CN gains, yet the number of protein-protein interactions is not (Fig. 4D)? Or maybe I am misunderstanding the SHAP value analysis, then please refer to major comment 2.

>> We thank the reviewer for the comment. Figure 4F (now 4D) suggests a similar trend of an increased number of protein-protein interactions (PPIs) being predictive towards buffering p compared to protein complex participation. The strength of this trend is weaker, however, which might stem from the transient nature these interactions have. Furthermore, the HIPPIE dataset contains a lot more PPIs than CORUM has protein complexes and covers multiple types of PPIs. Therefore, the decreased trend strength could be due to noise as some individual or types of PPIs may contribute less to buffering than others.

9. The authors state "While buffering in pan-cancer cell lines and tumor samples is comparable, chromosome-engineered aneuploid human cell lines are buffered to a greater extent. The underlying causes of these differences remain unclear". This finding is incongruent with work in yeast cells, where lab-engineered strains tend to buffer aneuploidy less efficiently - the authors should add this point to the discussion and speculate why this is the case. Also, for this comparison, the authors rely on ChrArm copy number changes (not gene CN changes) (Fig. 1F), could this skew the results?

>> This is a very interesting point. A recently published paper showed that lab-engineered yeast strains buffer aneuploidy less efficiently and have more transient changes in the karyotype compared to natural yeast strains (Muenzner et al., Nature 2024). The authors argue that aneuploidy may provide a selective advantage, but a lack of dosage compensation makes cells more vulnerable to the fitness costs associated with aneuploidy, making them transient in environmental stress conditions. Furthermore, they state that if environment conditions favor aneuploidy, selective pressure may select for cells with dosage compensation to stabilize the aneuploid state. However, in cell lines dosage compensation does not seem to contribute much to the adaptive process in human cell lines when comparing later passages to early passages, as we showed recently (Boekenkamp et al., EMBO J 2025). Our assumption is that the fitness costs induced by aneuploidy is stronger and the associated selective pressure much stricter in human cancer cells than in yeast cells. This may require early dosage compensation, otherwise the cells would not survive the aneuploidy induced stress. We therefore speculate that dosage compensation as an adaptive process occurs earlier than in yeast cells.

We verified that these differences are not skewed due to technical artifacts and choice of method by applying the buffering ratio to the natural yeast strain proteomics dataset provided by Muenzner et al. Using our BR-cutoff and their regression cutoff for determining buffering and attenuation, we confirmed 80% of genes they reported as attenuated also as buffered and were able to replicate the positive association between buffering and protein turnover (Spearman's $\rho = 0.22$, median BR \sim median protein turnover).

10. For Fig. 6E, the authors should explain in the figure legend how "drug effect" is defined.

>> We added the explanation to the figure legend.

11. In the discussion, the authors mention "In particular, the BR is sensitive to noise from both proteomics and copy number data. To circumvent these issues, we calculated confidence scores for the BRs, and removed BR measurements with low confidence." - I thought this was a good idea, and therefore the authors could briefly mention this point in the results section and not just "hide" it in the methods.

>> We thank the reviewer for the suggestion and added a description of the BR confidence values to the main text.

24th Nov 2025

Manuscript Number: MSB-2025-13094R

Title: Explainable Machine Learning Identifies Dosage Compensation Factors in Aneuploid Human Cancer Cells

Dear Prof Storchova,

Thank you for the submission of your revised manuscript to Molecular Systems Biology. We have now received the enclosed reports from the two referees that were asked to re-assess it and a new referee, Reviewer #4. As you will see the reviewers are now globally supportive and I am pleased to inform you that we will be able to accept your manuscript pending the following final amendments:

- 1) Please now remove all tracked changes/colored text from the manuscript.
- 2) Please correct the reference citation in the reference list such that when there are more than 10 authors on a paper, only the first 10 should be listed, followed by "et al.". Please check "Author Guidelines" for more information.
<https://www.embopress.org/page/journal/17574684/authorguide#referencesformat>
- 3) Our journal encourages inclusion of *data citations in the reference list* to directly cite datasets that were re-used and obtained from public databases. Data citations in the article text are distinct from normal bibliographical citations and should directly link to the database records from which the data can be accessed. In the main text, data citations are formatted as follows: "Data ref: Smith et al, 2001" or "Data ref: NCBI Sequence Read Archive PRJNA342805, 2017". In the Reference list, data citations must be labeled with "[DATASET]". A data reference must provide the database name, accession number/identifiers and a resolvable link to the landing page from which the data can be accessed at the end of the reference. Further instructions are available at .
 - the data callouts in the text for (Hausser, J. (2019, August 3); DepMap, Broad Institute. (2023)) are currently missing "Data ref:" as a prefix.
- 4) In the Methods, please take care of the following:
 - The Materials and Methods section should be renamed to "Methods".
 - Please ensure that a statement on whether or not blinding was done is included in the Methods even if no blinding was done. Please also be sure to update the Author Checklist with this information and where it can be found in the manuscript.
- 5) Please rename the individual sections of the manuscript and place them in the following order: Title page - Abstract & Keywords - Introduction - Results - Discussion - Methods - Data Availability - Acknowledgements - Disclosure and Competing Interests Statement - References - Figure Legends - Expanded View Figure Legends.
- 6) For the figures and figure legends, please take care of the following:
 - Please note that the exact p values are not provided in the legends of figures 1D, G, I; EV1 B, EV2 A
 - Please indicate the statistical test used for data analysis in the legends of figures 1D, G, H, I, K; 2E, 4D, 5B, C; 6B, C, D; EV1 B, EV2 B, F, G; EV5 B-F, H; EV6 C, D.
 - Please note that the box plots need to be defined in terms of minima, maxima, centre, bounds of box and whiskers, and percentile in the legends of figures 1D, G, H, I; 2E, 6D, EV1 B, E, F; EV2 D-G; EV5 E, F; EV6 C, D
 - Please note that information related to n is missing in the legends of figures 5A, 6A, EV1 E, EV5 A, EV6 A
- 7) Please rename the source file names, titles, legends and manuscript callouts for Table EV1-EV7 to Dataset EV4-EV10 (as there are already Dataset EV1-EV3 in zip folders). The legends should remain in a separate tab/sheet in each Excel file, just labeled correctly.
- 8) Please note that funding information should be given in the "Acknowledgements" section of the manuscript as well as our manuscript submission system. Currently Deutsche Forschungsgemeinschaft (DFG) STO 918/7-2 is missing in the manuscript. The following are missing in our manuscript submission system: an American Cancer Society Research Scholar Grant, a grant from Project 8p, grant #2023236 from the US-Israel Binational Science Foundation, a grant from the Andrew McDonough B+ Foundation, and a sponsored research agreement from Ono Pharmaceutical. There is also a discrepancy in a grant number: in the manuscript it is listed as R01CA276666 whereas in the manuscript submission system the same is listed as RO1CA2766666 (0 versus O). Please rectify this.
- 9) Synopsis:
 - Synopsis image: We would suggest simplifying the synopsis image. Currently most of the text will be difficult to read as it appears quite small. Moreover, we would advise a graphic that summarises the main findings of the manuscript on a glance.
 - Synopsis text: Please provide a separate word document including a short standfirst (maximum of 300 characters, including spaces) and up to 5 bullet points to summarise the key NEW findings. They should be designed to be complementary to the abstract - i.e. not repeat the same text. We encourage inclusion of key acronyms and quantitative information (maximum of 30 words / bullet point). Please use the passive voice.
 - Please check your synopsis text and image before submission with your revised manuscript. Please be aware that in the proof stage minor corrections only are allowed (e.g., typos).
- 10) Please ensure that a completed Source Data checklist is uploaded as a Related Manuscript File. You should have received instructions on how to prepare the checklist and files that should be uploaded in an email after the previous decision letter was sent to you. Source Data should be organized as a single source data file (zipped) per figure for main figures (all EV and/or

Appendix figure Source Data can be included in a single folder), with the panels clearly visible in the folder structure instead of a single excel file for all Source Data. e.g. all the Source data files for figure 1 need to be saved in a single folder and this needs to be zipped and then uploaded as "SD figure 1.zip" file.

11) As part of the EMBO Publications transparent editorial process initiative (see our policy here: https://www.embopress.org/transparent-process#Review_Process), Molecular Systems Biology will publish online a Peer Review File (PRF) to accompany accepted manuscripts. This file will be published in conjunction with your paper and will include the anonymous referee reports, your point-by-point response and all pertinent correspondence relating to the manuscript. Let us know whether you agree with the publication of the PRF and as here, if you want to remove or not any figures from it prior to publication. Please note that the Authors checklist will be published at the end of the PRF.

12) After your paper is published, we may promote it on social media. If you have any handles or hashtags for Bluesky you would like included, please let us know.

13) Please provide a point-by-point letter INCLUDING my comments and your detailed responses (as Word file).

I look forward to reading a new revised version of your manuscript as soon as possible.

Yours sincerely,

Poonam Bheda, PhD
Senior Scientific Editor
Molecular Systems Biology

Reviewer #1:

The authors have put considerable efforts in addressing each point raised in my review. It remains that as stated previously the paper is difficult to follow. This is mitigated by the exhaustive replies to the many concerns raised but again these did not make it into the revised manuscript to the extent others (the readers) will not raise concerns on their own.

Overall the paper is an elaborate analysis with a purpose and a biological question with "potential" clinical relevance worth publication.

I was unable to locate the new general Figure I suggested the authors prepare and they said they did, but I could not find.

Reviewer #2:

As I already said in the first round of revision, I support publication of this ms in MSB.

Reviewer #4:

In this paper, Heller et al analyze proteomic and genomic datasets from thousands of cancer cell lines, primary tumors, and engineered aneuploid cells, to show that protein-level buffering is widespread, context-dependent, and stronger in tumors than in vitro models. Using machine-learning models they find that buffering is most strongly predicted by gene essentiality, participation in protein complexes, mRNA decay properties, and dosage sensitivity, whereas transcriptional regulation favors proportional scaling. Highly buffered cancers display reduced proteotoxic stress, altered adhesion and ECM programs, and increased drug resistance, while poorly buffered cancers are vulnerable to chaperone, checkpoint kinase, and microtubule inhibitors. These results indicate that dosage compensation is a pervasive feature of aneuploid cancers.

This is a well-executed and conceptually important study that significantly advances understanding of how aneuploid cancers maintain proteostasis. Some of the findings have been observed previously in aneuploid yeast cells, but demonstrating that these hold true in the context of cancer is an important advance. The analytical innovations are solid and the biological interpretations thoughtful. The authors have done a very good job in responding to each of the reviewer comments. The remaining limitations of the study relate to causality and functional validation, but these do not detract from the manuscript's value.

I would suggest that the authors consider a title that conveys the biological finding rather than focuses on the methodology. For example, "Protein Buffering Is a Common and Predictable Outcome of Aneuploidy in Cancer"

Response to editor's comments

1) Please now remove all tracked changes/colored text from the manuscript.

>> Done

2) Please correct the reference citation in the reference list such that when there are more than 10 authors on a paper, only the first 10 should be listed, followed by "et al.". Please check "Author Guidelines" for more information.

<https://www.embopress.org/page/journal/17574684/authorguide#referencesformat>

>> Done

3) Our journal encourages inclusion of *data citations in the reference list* to directly cite datasets that were re-used and obtained from public databases. Data citations in the article text are distinct from normal bibliographical citations and should directly link to the database records from which the data can be accessed. In the main text, data citations are formatted as follows: "Data ref: Smith et al, 2001" or "Data ref: NCBI Sequence Read Archive PRJNA342805, 2017". In the Reference list, data citations must be labeled with "[DATASET]". A data reference must provide the database name, accession number/identifiers and a resolvable link to the landing page from which the data can be accessed at the end of the reference. Further instructions are available at

<https://www.embopress.org/page/journal/17574684/authorguide#referencesformat>.

- the data callouts in the text for (Hausser, J. (2019, August 3); DepMap, Broad Institute. (2023)) are currently missing "Data ref:" as a prefix.

>> Done

4) In the Methods, please take care of the following:

- The Materials and Methods section should be renamed to "Methods".

- Please ensure that a statement on whether or not blinding was done is included in the Methods even if no blinding was done. Please also be sure to update the Author Checklist with this information and where it can be found in the manuscript.

>> Done

5) Please rename the individual sections of the manuscript and place them in the following order: Title page - Abstract & Keywords - Introduction - Results - Discussion - Methods - Data Availability - Acknowledgements - Disclosure and Competing Interests Statement - References - Figure Legends - Expanded View Figure Legends.

>> Done

6) For the figures and figure legends, please take care of the following:

- Please note that the exact p values are not provided in the legends of figures 1D, G, I; EV1 B, EV2 A

>> The algorithm used for the analysis does not calculate precise p values below 10^{-99} in all cases, therefore we cannot provide the exact value.

- Please indicate the statistical test used for data analysis in the legends of figures 1D, G, H, I, K; 2E, 4D, 5B, C; 6B, C, D; EV1 B, EV2 B, F, G; EV5 B-F, H; EV6 C, D.

>> This has been done for all figures except 4D, where the statistical method is already denoted in the figure legend.

- Please note that the box plots need to be defined in terms of minima, maxima, centre, bounds of box and whiskers, and percentile in the legends of figures 1D, G, H, I; 2E, 6D, EV1 B, E, F; EV2 D-G; EV5 E, F; EV6 C, D

>> Done

- Please note that information related to n is missing in the legends of figures 5A, 6A, EV1 E, EV5 A, EV6 A

>> Done

7) Please rename the source file names, titles, legends and manuscript callouts for Table EV1-EV7 to Dataset EV4-EV10 (as there are already Dataset EV1-EV3 in zip folders). The legends should remain in a separate tab/sheet in each Excel file, just labeled correctly.

>> Done

8) Please note that funding information should be given in the "Acknowledgements" section of the manuscript as well as our manuscript submission system. Currently Deutsche Forschungsgemeinschaft (DFG) STO 918/7-2 is missing in the manuscript. The following are missing in our manuscript submission system: an American Cancer Society Research Scholar Grant, a grant from Project 8p, grant #2023236 from the US-Israel Binational Science Foundation, a grant from the Andrew McDonough B+ Foundation, and a sponsored research agreement from Ono Pharmaceutical. There is also a discrepancy in a grant number: in the manuscript it is listed as R01CA276666 whereas in the manuscript submission system the same is listed as R01CA2766666 (0 versus O). Please rectify this.

>> We corrected the information as requested; we now listed only the grants relevant for the project discussed in the manuscript, which are NIH grants R01CA237652 and R01CA276666, and a sponsored research agreement from Ono Pharmaceutical.

9) Synopsis:

- Synopsis image: We would suggest simplifying the synopsis image. Currently most of the

text will be difficult to read as it appears quite small. Moreover, we would advise a graphic that summarises the main findings of the manuscript on a glance.

- Synopsis text: Please provide a separate word document including a short standfirst (maximum of 300 characters, including spaces) and up to 5 bullet points to summarise the key NEW findings. They should be designed to be complementary to the abstract - i.e. not repeat the same text. We encourage inclusion of key acronyms and quantitative information (maximum of 30 words / bullet point). Please use the passive voice.

>> We have simplified the synopsis image and minimized the amount of the text.

10) Please ensure that a completed Source Data checklist is uploaded as a Related Manuscript File. You should have received instructions on how to prepare the checklist and files that should be uploaded in an email after the previous decision letter was sent to you. Source Data should be organized as a single source data file (zipped) per figure for main figures (all EV and/or Appendix figure Source Data can be included in a single folder), with the panels clearly visible in the folder structure instead of a single excel file for all Source Data. e.g. all the Source data files for figure 1 need to be saved in a single folder and this needs to be zipped and then uploaded as "SD figure 1.zip" file.

>> Done.

11) As part of the EMBO Publications transparent editorial process initiative (see our policy here: https://www.embopress.org/transparent-process#Review_Process), Molecular Systems Biology will publish online a Peer Review File (PRF) to accompany accepted manuscripts. This file will be published in conjunction with your paper and will include the anonymous referee reports, your point-by-point response and all pertinent correspondence relating to the manuscript. Let us know whether you agree with the publication of the PRF and as here, if you want to remove or not any figures from it prior to publication. Please note that the Authors checklist will be published at the end of the PRF.

>> We fully support the transparent editorial process initiative and agree with the publication.

12) After your paper is published, we may promote it on social media. If you have any handles or hashtags for Bluesky you would like included, please let us know.

>>

13) Please provide a point-by-point letter INCLUDING my comments and your detailed responses (as Word file).

Response to reviewers' comments

Reviewer #1:

The authors have put considerable efforts in addressing each point raised in my review. It remains that as stated previously the paper is difficult to follow. This is mitigated by the exhaustive replies to the many concerns raised but again these did not make it into the revised manuscript to the extent others (the readers) will not raise concerns on their own.

Overall the paper is an elaborate analysis with a purpose and a biological question with "potential" clinical relevance worth publication.

>> We thank the reviewer for the positive evaluation, considering the manuscript "worth publication". We indeed worked extensively to simplify the manuscript's text. While in principle it would be possible to add all additional extensive explanations, this would necessarily lead to expanding the text, which is already now very long. Through feedback from our colleagues and other reviewers we believe that the current version will be understandable to readers of MSB.

I was unable to locate the new general Figure I suggested the authors prepare and they said they did, but I could not find.

>> We prepared this as a part of the synopsis figure.

Reviewer #2:

As I already said in the first round of revision, I support publication of this ms in MSB.

>> We thank the reviewer for the support.

Reviewer #4:

In this paper, Heller et al analyze proteomic and genomic datasets from thousands of cancer cell lines, primary tumors, and engineered aneuploid cells, to show that protein-level buffering is widespread, context-dependent, and stronger in tumors than in vitro models. Using machine-learning models they find that buffering is most strongly predicted by gene essentiality, participation in protein complexes, mRNA decay properties, and dosage sensitivity, whereas transcriptional regulation favors proportional scaling. Highly buffered cancers display reduced proteotoxic stress, altered adhesion and ECM programs, and increased drug resistance, while poorly buffered cancers are vulnerable to chaperone, checkpoint kinase, and microtubule inhibitors. These results indicate that dosage compensation is a pervasive feature of aneuploid cancers.

This is a well-executed and conceptually important study that significantly advances understanding of how aneuploid cancers maintain proteostasis. Some of the findings have been observed previously in aneuploid yeast cells, but demonstrating that these hold true in the context of cancer is an important advance. The analytical innovations are solid and the biological interpretations thoughtful. The authors have done a very good job in responding to each of the reviewer comments. The remaining limitations of the study relate to causality and functional validation, but these do not detract from the manuscript's value.

I would suggest that the authors consider a title that conveys the biological finding rather than focuses on the methodology. For example, "Protein Buffering Is a Common and Predictable Outcome of Aneuploidy in Cancer"

>> We thank the reviewer for the support as well as for the suggestion of the improved title. We have now improved the title as follows:

Protein Buffering of Aneuploidy is Driven by Coordinated Factors Identified through Machine Learning

3rd Jan 2026

Manuscript number: MSB-2025-13094RR

Title: Protein Buffering of Aneuploidy is Driven by Coordinated Factors Identified through Machine Learning

Dear Prof Storchova,

Congratulations on an excellent manuscript, I am pleased to inform you that your manuscript has been accepted for publication in Molecular Systems Biology. Thank you for your comprehensive response to referee concerns. It has been a pleasure to work with you to get this to the acceptance stage.

You may qualify for financial assistance for your publication charges - either via a Springer Nature fully open access agreement or an EMBO initiative. Check your eligibility: <https://link.springer.com/journal/44320/how-to-publish-with-us>

Yours sincerely,

Sincerely,

Poonam Bheda, PhD
Scientific Editor
Molecular Systems Biology

>>> Please note that it is Molecular Systems Biology policy for the transcript of the editorial process (containing referee reports and your response letter) to be published as an online supplement to each paper. If you do NOT want this, you will need to inform the Editorial Office via email immediately. More information is available here: <https://link.springer.com/partners/embo-press/editorial-policies#Peer%20review>